# Intrinsic brain dynamics in the Default Mode Network predict involuntary fluctuations of visual awareness

Dian Lyu[1,2], Shruti Naik [3], David K. Menon [1,4] & Emmanuel A. Stamatakis [1,2] ✉

Brain activity is intrinsically organised into spatiotemporal patterns, but it is still not clear whether these intrinsic patterns are functional or epiphenomenal. Using a simultaneous fMRI-EEG implementation of a well-known bistable visual task, we showed that the latent transient states in the intrinsic EEG oscillations can predict upcoming involuntarily perceptual transitions. The critical state predicting a dominant perceptual transition was characterised by the phase coupling between the precuneus (PCU), a key node of the Default Mode Network (DMN), and the primary visual cortex (V1). The interaction between the lifetime of this state and the PCU- > V1 Granger-causal effect is correlated with the perceptual fluctuation rate. Our study suggests that the brain's endogenous dynamics are phenomenologically relevant, as they can elicit a diversion between potential visual processing pathways, while external stimuli remain the same. In this sense, the intrinsic DMN dynamics pre-empt the content of consciousness.

Being aware is believed to be a globally "illuminated" inner state when locally encoded information gets propagated through subsequent pathways and becomes accessible to other processing streams in the brain[1]. However, the mechanism of the propagation process is largely unknown. Recent theoretical developments in brain dynamics suggest that the spontaneous information propagation may be empowered by the intrinsic ignition of neural activity[2,3]. Empirically, intrinsic brain activity has been extensively studied recently during resting state when no external stimuli are presented to participants. At rest, neural activity from certain distant regions is correlated forming what are known as large-scale networks. We hypothesise that the information propagation during the state of "being aware" can be influenced by intrinsic perturbations from the endogenous dynamics of large-scale brain networks, by which the brain's intrinsic activity might cause a "butterfly effect" to the downstream perceptual or cognitive events. In this study, we used the well-known Binocular Rivalry (BR) paradigm, with a simultaneous functional magnetic resonance imaging (fMRI)-electroencephalogram (EEG) implementation, to investigate whether

and how intrinsic brain activity of a large-scale brain network influences the involuntary perceptual fluctuations during bistable visual awareness.

BR is a robust visual phenomenon where the participant perceives continuously alternating images even though the external visual stimuli remain the same. To successfully trigger this phenomenon, dissimilar images need to be presented to the two eyes (each eye's view is blocked from the other); and to ensure balanced representations of the two images (or "percepts" as we shall call them in this paper), the two images should be distinguishable but of comparable features such as image contrast, cognitive load of the content etc.[4]. BR is a popular paradigm for the studies of visual awareness, and by extension the neural correlates of consciousness (NCCs)[4–6]. A classical model for explaining the BR mechanism is the mutual inhibition model (MIM), which generally takes the form of a non-linear dynamic system[7]. Such models require mutually inhibitory neurons encoding the representations of the two images (forming attractors in the model) and possible sources of perturbations, i.e., global and/or local adaptation

[1]University Division of Anaesthesia, University of Cambridge, Addenbrooke's Hospital, CB2 0QQ Cambridge, UK. [2]Department of Clinical Neuroscience, University of Cambridge, Addenbrooke's Hospital, CB2 0SP Cambridge, UK. [3]Cognitive Neuroimaging Unit, INSERM, CEA, Universite´ Paris-Saclay, 91191 Gif/ Yvette, France. [4]Wolfson Brain Imaging Centre, University of Cambridge, Cambridge Biomedical Campus (Box 65), CB2 0QQ Cambridge, UK. ✉e-mail: eas46@cam.ac.uk

of the neuronal populations as well as neuronal noise[8–10]. A dominant percept during BR can be considered as a stable state/equilibrium achieved by the neuronal activity settling in one of the two possible attractors, while the state can escape from the current attractor under the influence of noise or habituation processes (such as spike frequency adaptation and synaptic fatigue)[7–10]. The MIM has been successful in fitting behavioural data measured during experiments, but empirical neural evidence to support the model is limited[11]. In fact, this explanatory framework only focuses on the dynamics of the local neural circuitry, but totally ignores the background neural dynamics. However, the intrinsic dynamics of brain networks may have a lot of interactions with primary sensory regions, and as a result modulate local sensory processing[1,12–14].

Intrinsic brain activity is not random, instead, it is spatiotemporally organised into reproducible, topologically meaningful patterns[15–17]. Although the spatial patterns of large-scale networks are mostly derived from mentally-unconstrained resting state, they can also be identified in tasks across various cognitive domains, hence have been suggested to serve as a topological scaffold supporting various brain processes[18,19]. A large-scale network that has achieved a prominent position in recent literature is the default mode network (DMN)[20]. The DMN encompasses a wide range of associative regions across the prefrontal, temporal and parietal lobes, and consumes a major part of the brain's energy budget[20,21]. A functional gradient analysis demonstrated that the DMN is situated at the top of the brain's information processing hierarchy, where multi-modal sensory information is integrated and highly abstract information (such as a concept of "self") is formed[22]. Dynamic neural state studies using Hidden Markov models have shown that DMN activity dominates the latent brain states which are characterised by the synchronisation of high-order cognitive networks; where latent states are defined as recurring transient states that are assumed to drive the observed global neural dynamics[15,17]. From a perspective of control theory, it was also suggested DMN regions have the highest capability for steering the whole brain from one state to another[23].

Clinical studies have already highlighted the significance of the DMN in consciousness[24–26], but little work has been done to investigate the role of the DMN in visual awareness, even though the bistable visual phenomenon is a classic experimental paradigm for investigating the NCCs[5,6,27]. There has been some suggestion in the literature that the DMN may play a role in BR. In many fMRI/EEG studies utilising the BR paradigm, frontal and parietal regions have been reported to be engaged[28–31], some of which may overlap with DMN subregions, especially in the parietal cortex where high-level cognitive networks mostly converge[32]. A recent fMRI study using ambiguous images as stimuli to study pattern disambiguation established a role for the DMN in prior-guided visual perception[33], which suggests that the DMN is involved in the online modulation of visual processing and its function may perhaps be associated with perceptual disambiguation. Furthermore, there is accumulating evidence demonstrating the DMN's engagement in tasks, contrasting to the well-established belief that the DMN is a "resting state network"[34–36]. These suggest that the contribution to cognitive function from areas that make up the DMN is still not well understood. Given the DMN's role in global cortical dynamics and conscious representation, we hypothesised that intrinsic DMN dynamics may influence the perceptual fluctuations during BR, possibly by causing perturbation to the equilibrium of the current dominant percept.

## Results
### Behavioural analyses
This experiment has been previously validated, analysed and published in the recent literature[37,38]. For maximally eliciting BR perception, the experiment used stimuli of rotating green and red checkerboard images, which were presented to each eye simultaneously. The order of green and red visual stimuli was counterbalanced between the left and right eyes across participants and their multiple experimental sessions. In a perceptually matched Replay (RPL) condition where no BR is elicited, the same rotating images of the red or green checkerboard were presented to both eyes at the same time, but they alternated in time (Fig. 1a, d). Participants were asked to instantaneously report their percept by using three different buttons, respectively for the (dominant) red or green percept, and the mixed percept i.e., a transitional phase between the red and green percept. Therefore, this paradigm had a 2×2 factorial design of dominant (red or green percept) and mixed percept types for the BR and RPL conditions.

Individual differences of the percept duration during BR were significant (Fig. 1b) (Residual Sum of Squares [RSS] was reduced by 1490, $p = 0.00$ according to a $\chi^2$ test, and the Akaike Information Criterion [AIC] was decreased by 1057.5. Both measures suggest that there was stronger model evidence for a model that considers individual differences compared to the null model). The duration of (dominant/mixed) percept formed a heavy-tailed distribution (skewness = 1.29, kurtosis = 2.89 for dominant percept; 3.15 and 17.82 for mixed percept), and the dominant percepts (median = 2.22 seconds for the red and 2.04 for the green) were significantly longer (decreased $AIC/dAIC$ = 7985) than the mixed percepts (median = 0.33) (Fig. 1c).

### Multimodal neuroimaging analyses
Different than the goal of the original study using this dataset to reveal fMRI activation associated with BR-induced perceptual transitions[38], our goal was to test the hypothesis that the endogenous neural dynamics in high-level cortical areas (i.e., DMN) can influence low-level (i.e., primary visual cortex) information processing. Given the controversies regarding the DMN's role during tasks[34,36,39], we first confirmed the involvement of the DMN regions in this task with the fMRI dataset. Then using the EEG dataset, we conducted evoked response analyses to (1) narrow down a time window preceding the subjective report of a perceptual transition and (2) constrain source spatial localisation within the DMN, for facilitating our further modelling of the dynamic neural process. Finally, with the DMN source signals of the specified time window, we adopted time-delay embedded Hidden Markov Model to search for the transient spatiotemporal patterns which serve as endogenous neural triggers of the upcoming, involuntary perceptual changes during BR. The analysis pipeline is presented in the supplementary information (Supplementary Fig. 1)

In agreement with the previous literature[30,31], we found significantly activated regions in the cuneus, the intraparietal sulcus/inferior parietal lobule (BA40), angular gyrus (BA39) and inferior frontal gyrus (BA47), for the contrast of BR (dominant) > RPL (dominant). As suggested by the meta-analysis driven BrainMap network atlas, a large portion of the significantly activated regions overlapped with the DMN, supporting our hypothesis about the DMN's involvement in the task (view the full report of significant clusters by using this link: https://htmlpreview.github.io/?https://github.com/Aubrey-Lyu/BR-project/blob/master/Analysis-2_fMRI/results/fMRI_activation_results_formated.html). Here we highlight an $F$-contrast for the 2-by-2 interaction effect: 2 perception types (dominant vs. mixed) × 2 ways of perception generation (BR vs. RPL), which revealed the DMN and visual cortex having the most variable activation patterns among the four conditions (Fig. 2).

The EEG recording with its excellent temporal resolution allowed us to investigate a short time interval right before manual indication of a perceptual change. Global Field Power (GFP) of the perceptual conditions revealed that the most eventful epoch was between [−400, −200] ms, which was when the EEG sensor voltages differed the most between the BR (dominant) and RPL (dominant) conditions (Mean GFP was 0.99 μV for BR and 1.59 μV for RPL, with a difference of $t = -2.23$, $p = 0.037$). Further, a cluster-based permutation $t$-test over the [−500, 0] ms time-window confirmed significant time-clusters between [−388,

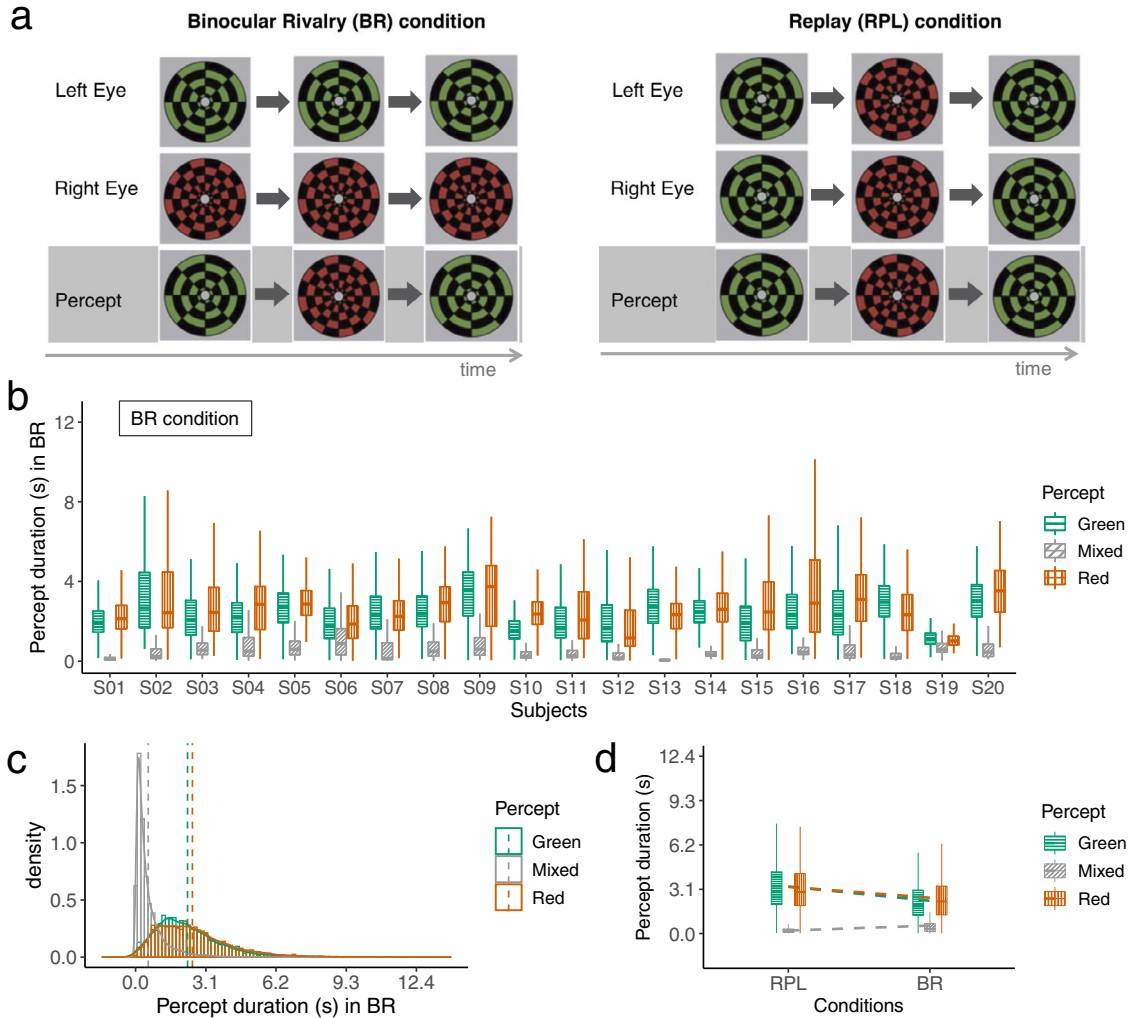

**Fig. 1 | Experimental design with 2-by-2 conditions of different percepts and ways of percept generation. a** Experimental schema adapted from the original study from which the data was shared[38]. During the binocular rivalry (BR) blocks (left), the participants were presented with different images, each for one eye, in the centre of the visual field. During the replay (RPL) blocks (right), the participants were presented with the same image for both eyes, and this image alternates over time between red, green via a short superimposition. The purpose of the RPL condition is to simulate the perceptual experience in the BR condition but without evoking binocular rivalry: both conditions can generate alternating red/green (dominant) and mixed (transitional) percepts, but the alternation was endogenously generated in the BR condition, while it was exogenously elicited in the RPL condition. **b** Individual difference in percept duration (s) for different percept types in the BR condition ($N_{trial} \approx 327$ per participant per condition). **c** Distributions of percept duration in the BR condition. **d** Percept duration for different percepts in the RPL and BR condition ($N_{participant} = 20$).

−350] ms, $p$-corr < 0.05) (Fig. 3a). Though based on response-locked analyses, this result is comparable with previous literature dominated by stimulus-locked analyses, which suggests that a ERP component (roughly happening 300 ms before a key press) from parietal activity indicates a conscious recognition of a perceptual disambiguation[40].

We observed no significant difference for mixed percept between the BR and RPL conditions during the same temporal window (Fig. 3b, mean GFP during [−400, −200] ms was 1.11 μV for BR and 1.34 μV for RPL, with a difference of $t = -1.45$, $p = 0.175$; Fig. 3b). Further 2-way Analysis of Variance (ANOVA) in this time-window revealed a significant main effect of the source of perceptual transitions ($F_{1,19} = 4.70$, $p$-corr = 0.04), but no significant main effect of perceptual types (mixed or dominant) was found ($F_{1,19} = 0.29$; $p$-corr = 0.59). Although there was a trend in the interaction effect between the source of perceptual transitions and the perceptual type, the test was not significant ($F_{1,19} = 2.63$; $p$-corr = 0.12; Fig. 3c). The event-related potential (ERP) topographies during the [−400 −200] ms time window for the dominant percept in both conditions are shown in the Fig. 3d.

To discover the brain location of the signals that generated this difference in scalp topography, we then conducted source reconstruction for the ERPs during this time window. The source reconstruction indicated that the toporagphy was driven by a deactivation in the posterior cingulate cortex (PCC; Brodmann area (BA) 23/24), precuneus (PCU; BA 31/7 m), thalamus, insula, caudate, claustrum and the fusiform gyrus (BA 20) during BR (dominant) vs. RPL (dominant) condition. An $F$ contrast during this time window for the interaction effect of the four conditions revealed the following regions: insula (BA 13), postcentral gyrus (BA 43), thalamus, parahippocampal gyrus (BA 34), PCC (BA 30), inferior frontal gyrus (BA 47) and anterior cingulate cortex (ACC) (Fig. 3e). The activation profiles for finer grained frequency bands and time windows were also explored for a sanity check, with no further attention paid to their differences. The full results are presented in the online repository: https://htmlpreview.github.io/? https://github.com/Aubrey-Lyu/BR-project/blob/master/Analysis-3_EEG/results/evokeResponse_result_table_permutationtest.html.

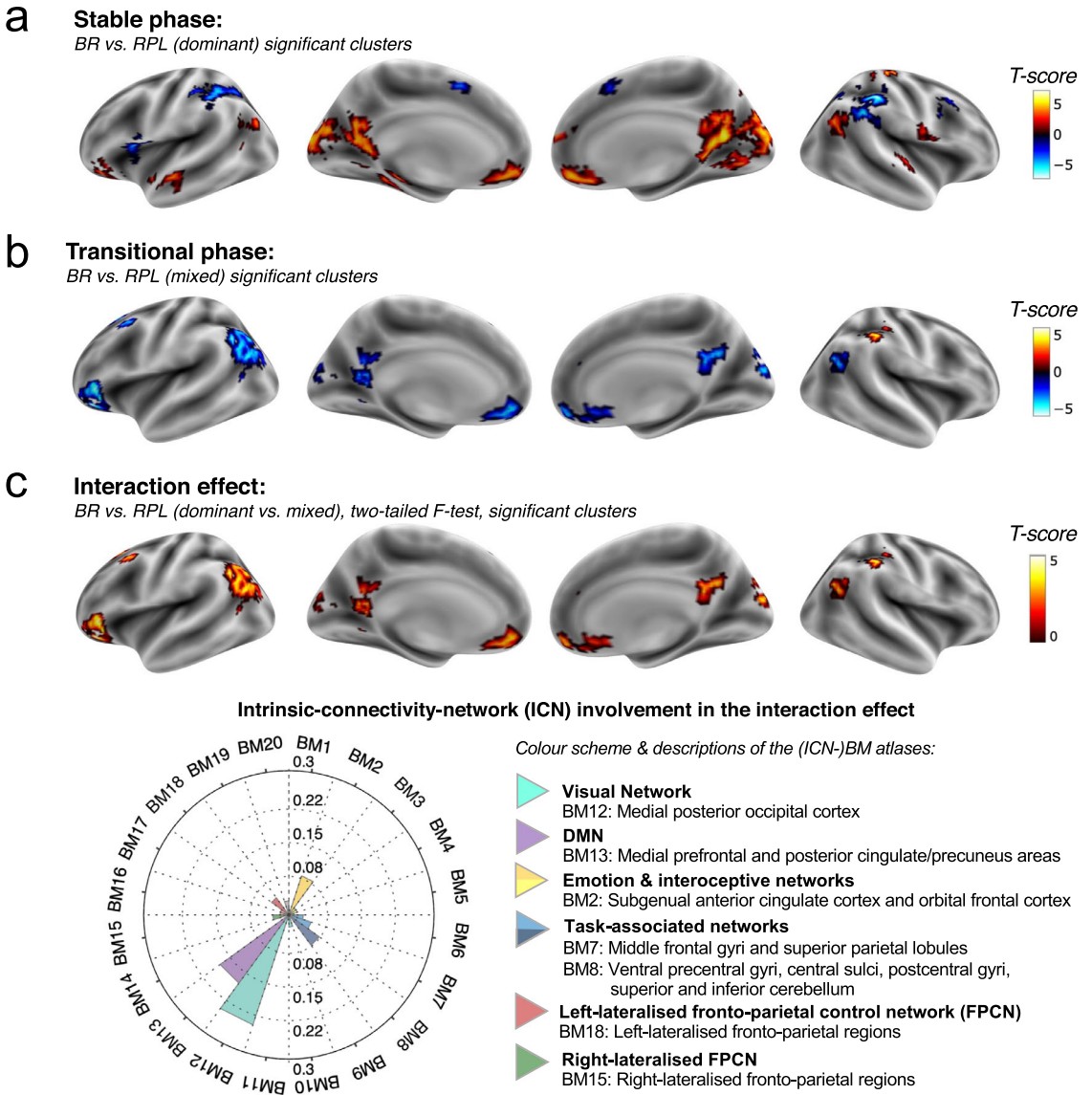

**a  Stable phase:**
*BR vs. RPL (dominant) significant clusters*

**b  Transitional phase:**
*BR vs. RPL (mixed) significant clusters*

**c  Interaction effect:**
*BR vs. RPL (dominant vs. mixed), two-tailed F-test, significant clusters*

**Intrinsic-connectivity-network (ICN) involvement in the interaction effect**

*Colour scheme & descriptions of the (ICN-)BM atlases:*

**Visual Network**
BM12: Medial posterior occipital cortex

**DMN**
BM13: Medial prefrontal and posterior cingulate/precuneus areas

**Emotion & interoceptive networks**
BM2: Subgenual anterior cingulate cortex and orbital frontal cortex

**Task-associated networks**
BM7: Middle frontal gyri and superior parietal lobules
BM8: Ventral precentral gyri, central sulci, postcentral gyri,
        superior and inferior cerebellum

**Left-lateralised fronto-parietal control network (FPCN)**
BM18: Left-lateralised fronto-parietal regions

**Right-lateralised FPCN**
BM15: Right-lateralised fronto-parietal regions

**Fig. 2 | FMRI activation revealing the DMN's involvement in the current task.** Subplots **a**–**c** show the significant clusters [$p_{voxel} < 0.001$ (uncorrected) & $p_{cluster} < 0.05$ (family-wise error corrected)] respectively for the contrasts "BR (dominant) - RPL (dominant)", "BR (mixed) - RPL (mixed)" and the interaction effect between perceptual generating (BR vs. RPL) and perceptual (dominant vs. mixed) conditions. The interaction analysis (**c**) shows the regions mostly sensitive to the condition differences. For statistical testing of (**a**–**c**), we adopted linear mixed-effects models, where the paired *t*-tests and *F*-test were carried out at the individual level, while statistical inferences were made at the population level with group-level one-sample *t*-tests (input being the individual-level estimators). Hence, the colour bars of the subplots (**a**–**c**) indicate the *t* scores from the group-level testing ($N_{participant} = 20$). Boxplots in **b** and all other cases in this paper present the median, lower quartile and upper quartile of the data respectively at the middle, the lower and upper bound of the boxes. The data range is indicated by the whisker vertically centred at the box. The circular plot in **c** shows the Intrinsic Connectivity Network

(ICN) affiliation of the significant clusters for the interaction effect. "ICN involvement" is a measure of correspondence between an activation map and large-scale networks with well-established cognitive function, as provided by the BrainMap (BM) meta-analysis database[89]. The BM number on the circular plot indicates the number of the ICN-BM network atlas. Cognitive domains of the networks that have negligible involvement in this task are not listed in the figure. These are BM1: Limbic and medial-temporal areas; BM3: Bilateral BG and thalamus; BM4: Bilateral anterior insula/frontal opercula and anterior cingulate gyrus; BM5: Midbrain; Cerebellum; BM6: Superior and middle frontal gyri; Sensorimotor; BM9: Superior parietal lobule; Frontoparietal (perception-somesthesis-pain); BM10: Middle and inferior temporal gyri; Frontoparietal (cognition-language); BM11: Lateral posterior occipital cortex; BM14: Cerebellum; BM16: Transverse temporal gyri; BM17: Dorsal precentral gyri, central sulci, postcentral gyri, superior and inferior cerebellum; BM19: Artefactual component; BM20: Artefactual component.

## Dynamic neural pattern analyses

To uncover the endogenous neural activities that trigger the upcoming perceptual change, we applied a dynamic neural pattern analysis to the one second (s) time window before every subjective indication of a perceptual change. We identified the dynamic neural patterns/states by using the time-delay embedded Hidden Markov Model (TDE-HMM) toolbox (https://github.com/OHBA-analysis/HMM-MAR) which provides a series of encoding and decoding methods to (1) discover the transient spatiotemporal patterns inherent in the source signals of all

trials, (2) extract spectral information (i.e., the power and phase coherence among ROIs in canonical frequency bands) from the states for interpretation. According to our hypothesis: if the intrinsic fluctuation of the DMN regional activity influences the perceptual transitions during BR, we should expect to find a correspondence between the neural dynamics of DMN regions and the upcoming perceptual transitions.

We set the HMM algorithm to extract four states among the regional signals of all trials from all conditions and subjects (Fig. 4).

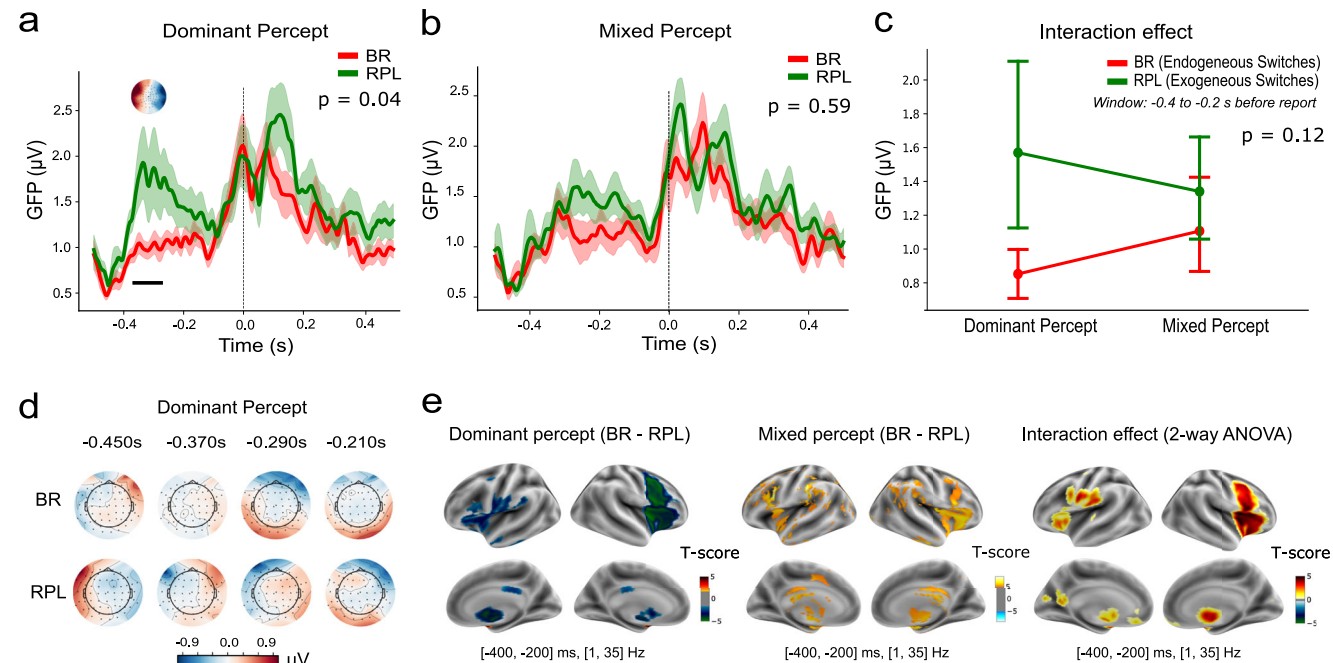

**Fig. 3 | Event-Related Potentials for different perceptual conditions. a** Event-Related Potential contrast between BR and RPL for dominant percept at the sensor level. The global field power (GFP) in the RPL condition was significantly higher than that in BR condition from −400 to −200 ms before the reported change in percept at 0 s. This is indicated in the plot with the black horizontal line ($p$-corr <0.05 significant clusters, by one-sample permutation cluster $t$-test). The ERP topography for the "BR (dominant) vs. RPL (dominant)" contrast is displayed at the top of the significant time window. **b** No significant difference was found in the GFP between the BR (mixed) and RPL (mixed) conditions. **c** GFP difference across sources of perceptual transitions and types of percept. The interaction effect was not significant ($p$-corr = 0.12, $F_{1,19}$ = 2.63). The plot shows the averaged GFP across the time window between −400 and −200 ms and its 95% confidence intervals for each group. **d** ERP topographies in both of the BR and RPL conditions, dynamically changing during the critical time window (e.g., [−450, −210] ms) before a response to dominant percept. **e** Source localisation of the evoked responses contrasted between BR and RPL conditions, and for the interaction effect of the two variables, during the [−400, −200] ms before a subjective report. Plotted brain regions/voxels survived the significance test with FWE-corrected non-parametric $p$-values <0.05.

Four components were specified as we expect the states to be interpretable in terms of the experimental conditions, supposing that the visiting time of the four states in the four conditions would be significantly different. In addition, four components have often been specified for discovering the EEG microstates in existing literature[41], and were shown to have correspondence with the Intrinsic Connectivity Network (ICN) dynamics measured from the BOLD signals[42]. The resulting auto-covariance patterns of the four states are presented in the SI (Supplementary Fig. 4). All of the comparisons were carried out within subjects and the difference was then grouped together for the population-level inference.

To establish the cognitive relevance of the data-driven neural states, we firstly compared the states' dynamics features, such as the switching rate (SR), i.e., the rate of state switches which can be understood as a measure of state stability, and the fractional occupancy (FO), i.e., the proportion of dwelling time on a certain state given a period of time.

The SRs were not different across conditions, except that the RPL (mixed) had significantly higher SR than the rest ($t$ = 3.57, $p$-adj = 0.00 vs. BR (dominant); $t$ = 3.30, $p$-adj = 0.00 vs. BR (mixed), and $t$ = 3.00, $p$-adj = 0.01 vs. RPL (dominant)). The fact that the SRs were about a quarter each suggested that there was a good mix of the presence of the four states, indicating that the states had been separated well for capturing the multi-dimensional variance in the data.

Importantly, we found that the four states' FO was significantly different among the conditions ($F_{9,304}$ = 22.86, Cohen's $F^2$ = 0.68, $p$ = 0.00, $dAIC$ = 147.45). To establish exactly how the states' FO differed (Fig. 5a), permutation tests were conducted within each experimental condition for state-wise comparisons (with 2000 permutations for simulating a null distribution of the state-wise differences). It

turned out that the states can discriminate conditions, in a way that a unique state was always visited mostly during one specific condition (Fig. 5b). This pattern was most noticeable in the BR (dominant) and RPL (mixed) conditions, where respectively State 4 and State 1 clearly stood out, with significant median differences of 0.44 ($p$-adj = 0.00), and 0.36 ($p$-adj = 0.00) to the second most prominent state in their respective conditions. Similarly, in the RPL (dominant) and the BR (mixed) condition, State 2 and State 3 were respectively visited the most, nevertheless they were not visited significantly more often than the second most prominent state.

To validate that the states captured the endogenous neural trigger of perceptual transitions in the BR, we further conducted a cross validation (CV), using the trial-by-trial FOs of the hidden neural states to predict the type (mixed or dominant) of the upcoming transition. This was done respectively for the BR and RPL conditions. As a result, the accuracy score for the BR condition was 91.38% ($SD$ = 7.98%), which was significantly higher ($t$ = 2.92, $p$ = 0.006) than the score of 82.40% ($SD$ = 10.77%) for the RPL condition (see the SI Supplementary Fig. 6 for more details). This suggests that these neural states have registered the endogenous neural activities that could predict the upcoming perceptual change during BR.

To unpack the neuronal signatures encapsulated in the states, we then extracted the spectral information (power and phase coherence) from the multivariate auto-covariance matrix of each state. Once we had estimated the power and coherence for each state, we factorised the frequencies into a few dominating components so as to facilitate interpretation with conventionally defined frequency bands. For that, we applied a non-negative matrix factorisation (NNMF) algorithm on the coherence matrix, concatenated across all states and ROIs and confined it to four components. The four components turned out to

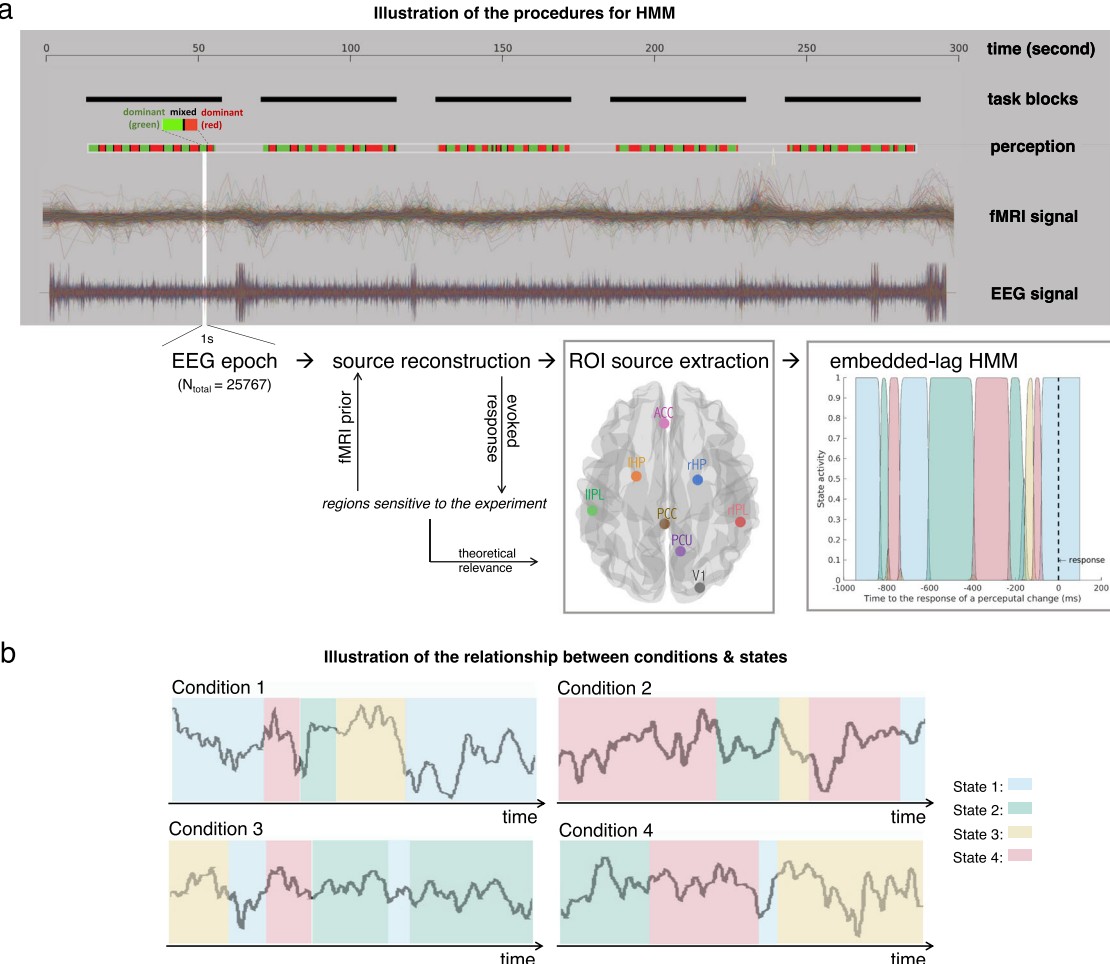

**Fig. 4 | Schematic illustration of the procedures for HMM. a** Illustration of the HMM analysis pipeline. The one-second EEG epochs/trials were taken right before a response of every perceptual change ("dominant" in red or green, "mixed" in black). To ensure experimental sensitivity, we select our regions of interest (ROIs) from the significant results of the source-level evoked response in our previous analyses. Also because of our theoretical interest, we further constrained our selection of ROIs by choosing only the V1 and DMN regions (among all the significant voxels) to construct the HMM. In the last subplot we presented an example of how the states change across time in a random trial. The motivation and procedures for selecting

the particular time window and ROIs were detailed in the corresponding method section in the SI. **b** Illustration of the relationship between conditions and states. The surrogate timeseries indicate the EEG source signals of a single trial. The TDE-HMM algorithm clusters the timeseries into four states based on the signals' inherent spectral features. The inherent states are assumed to recur across time and to be replicable in all trials, conditions and participants, as the endogenous neural activity have been showed to have a robust spatiotemporal structure. However, given the phenomenological differences among the conditions, the compositions of the states during the trials are hypothesised to vary.

match traditional frequency bands: theta (peak at 4 Hz, half-maximum at 1, 12 Hz), alpha (peak at 11 Hz, half-maximum at 7, 15 Hz), beta (peak at 20 Hz, half-maximum at 15, 34 Hz) and gamma (peak at 34 Hz, half-maximum at 26 Hz; Supplementary Fig. 8 in SI). Therefore, we next focused on state-wise frequency-specific spectral information. To identify the neural features that are representative to each state, we performed permutation tests for each power and coherence values, comparing across all four states (5000 permutations for each null distribution simulation; significance level is $p < 0.01$; Fig. 5c).

Within the alpha band, State 4 was characterised by an increased phase coherence between PCU and V1, and between PCU and rIPL, and State 1 was characterised by an overall increased power in the DMN and V1 regions, accompanied by a general decreased phase coherence among the posterior DMN nodes (PCC, bilateral HP and IPL). These two states were often present during the BR (dominant) trials [FO = 0.62 ± 0.27, 0.29 ± 0.22 (mean ± standard deviation), respectively for State 4 and 1]. Comparing the phase coherence among the ROIs across the four states (Fig. 5c), the one between PCU and V1 clearly stood out, as the PCU-V1 phase coherence was highest in State 4, an indicative state for the BR (dominant); and it was the lowest in State 2, an indicative

state for the RPL (dominant) condition. This made us hypothesise that the PCU-V1 coherence may underlie the spontaneous transition to a stable percept.

For the state features in other frequencies (Supplementary Figs. 11–13 in SI), we would like to highlight the general increase of the theta coherences among DMN regions before the mixed-percept transition in BR, especially the increased theta coherence between ACC and PCC, which is unseen in other frequencies. This corresponds to previous MEG literature where intrahemispheric and interhemispheric theta coherences were found to be increased during perceptual dominance periods in BR[43]. The strong ACC-PCC coherence that we observed only in the theta oscillations also echoes the result of a recent study focused on the endogenous neural dynamics during resting state, using the same HMM technique, which showed that the mPFC/ACC-PCC coherence in the delta/theta frequencies characterises the anterior higher-order cognitive state[17].

We then investigated the behavioural relevance of the critical state: State 4. State 4 was focused on because it was visited distinctively the most in the BR (dominant) condition, and the least in the RPL (dominant) condition. State lifetime was used as an intuitive index

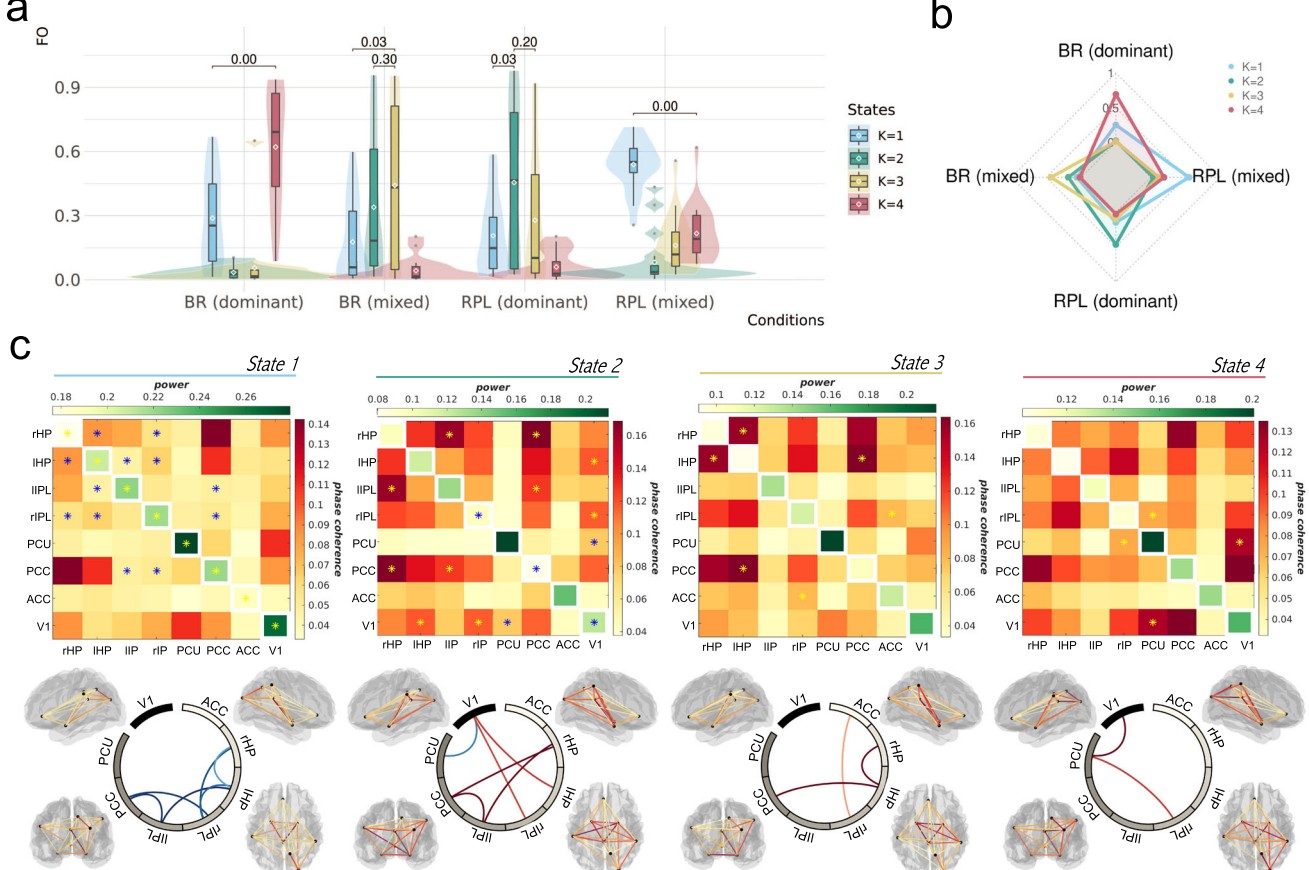

**Fig. 5 | Correspondence between states and conditions. a** Different fractional occupancy (FO) of the four states across conditions. Plotted data are the mean FO values averaged across trials for each individual, grouped by the two categorical variables: conditions and states. FDR-adjusted *p*-values of pairwise permutation tests (two-tailed) are presented on the significance bar on top of the compared groups (*N* = 20 per group). The distributions of non-averaged FO for all trials/ epochs of all participants, and an individual's averaged State FO in the four experimental conditions are presented in SI (Supplementary Figs. 9 and 10). For the box plots, the white dot/circle on each box indicates the mean, while the length of the box indicates quantiles with the middle bar showing the median. **b** Radar chart showing the correspondence between conditions and states. Scales on the radar chart indicate the median values of the state FO in the corresponding condition. **c** Spectral information of the states in the alpha band. Diagonal and off-diagonal

values of the upper-triangle heatmaps respectively indicate the power and phase coherence of the ROIs. The asterisk on the heatmap indicates that the phase coherence (or power) in this state is significantly higher (yellow) or lower (blue) compared with the other states (one-tailed test with the 95% confidence interval by permutation). The circular bundle plots below highlight the significant connectivities, with red and blue respectively signifying higher and lower significance for the connectivity. All of the connectivities (whether significant or not) are also mapped onto standardised brain anatomy from the left, right, back and top views (clockwise), where the colour intensity of the connectivity has been normalised across all states, thus being suitable for visual state-wise comparisons. Coordinates of the nodes were selected based on significant peaks from the previous evoked-response analyses (see the Supplementary Table 2 for all of the peak coordinates).

for the state presence, estimated as the time that elapsed between entering and exiting a state according to the HMM[44]. We correlated it with the reaction time (RT) which was defined as the interval of the participant's adjacent reports to perceptual changes in both BR (dominant) and BR (mixed) conditions. When the correlation is examined on the individual level (i.e., when the variance is provided by different performance across trials), the RT was the perceptual duration of the last percept; but when examined at the group level (i.e., variance provided by different average performance of all individuals), the individual-averaged RT reflected the alternation rate for that participant. We found that longer State 4's lifetime could predict faster perceptual transitions at the group level (*b* = −0.38, *t* = −2.28, df = 37, Cohen's $F^2$ = 0.12, *p* = 0.03; *dAIC* = 35.03; Fig. 6a).

We further investigated the cognitive implication of the "top-down" connectivity from the PCU to V1, since we are interested in how the DMN exerts an influence on the V1, and PCU was the only DMN region shown to have a functional coupling with V1. The functional coupling between the PCU and V1 stood out as a significant neural feature of State 4; and it also demonstrated opposite patterns between BR (dominant) and RPL (dominant), the key contrast that we are

interested in. As we were agnostic to the nature of this hypothetical causal relationship, we adopted a simple form of causality to perform this analysis, namely, Granger Causality (GC), which basically searches for a time-lagged linear relationship between $x_1$ and $x_2$ that can predict $x_1$ better than what $x_1$'s own auto-regression can predict (i.e., $x_2$ Granger-causes $x_1$).

We did not find a trial-by-trial relationship between the GC (PCU → V1) and the perceptual duration, or an interaction between the GC (PCU → V1) and State 4 lifetime to the perceptual duration. This might be caused by a ceiling effect of the State 4 lifetime from a within-subject analysis: there are six participants who showed a dominant presence of State 4 (> 0.9 s) in most of the trials in the BR (dominant) condition, thus providing little variability/statistical power for examining the aforementioned interaction effect. However, a relationship between the GC (PCU → V1) and the perceptual duration was established at the group level. Specifically, we found an interaction effect between the individual-averaged state lifetime (K = 4) and GC (PCU → V1) for predicting the individual-averaged perceptual duration. In other words, the GC becomes more relevant to quicker perceptual transitions when the trial is visited by State 4 more (*b* = −0.39, *t* = −3.01,

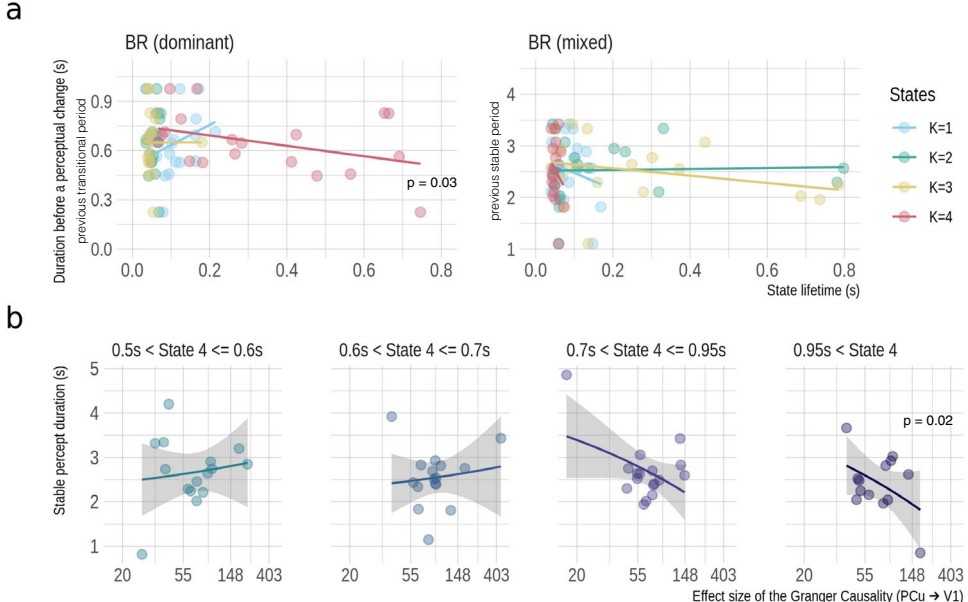

**Fig. 6 | Behavioural relevance of State 4. a** Lifetime of State 4 can predict the length of intervals between perceptual changes during BR (for both dominant and mixed transitions). The longer State 4 persists, the shorter the transitional period lasts, i.e., the quicker the rivalry is resolved. **b** shows that the duration of the last stable percept is correlated with the Granger causality (GC) effect from PCU to V1 at the group level ($N = 20$, shaded area indicating the model residual of the linear regression). Their relationship is also dependent on the lifetime of State 4. The GC effect size was approximated by the improvement of model evidence (para-meterised by decreased Bayesian Information Criterion [dBIC]).

df = 76, Cohen's $F^2 = 0.13$, $p = 0.00$ fitted with a generalised linear model; $dAIC = 4.54$ compared to a multiple regression model with no interaction). When State 4 dwells for the majority (> 95%) of the trial time (i.e., >0.95 s), the negative correlation between the GC and the perceptual duration was significant ($b = −0.931$, $t = −2.821$, df = 11, Cohen's $F^2 = 0.41$, $p = 0.017$; $dAIC = 29.56$; Fig. 6b). This suggests that the top-down modulation of the PCU to the V1 can influence the perceptual transition under the condition of State 4 being present.

## Discussion

Awareness is suggested to emerge through the interactions between local processing triggered by external stimuli, and global broadcasting afforded by the brain's intrinsic spatiotemporal states[1]. Resting-state whole-brain signals are shown to be spatiotemporally organised[15], but it is still not clear whether this intrinsic architecture is epiphenomenal or functional. By using simultaneously recorded EEG-fMRI data, we showed that the brain's spatiotemporal patterns or states found in this study, characterised by EEG oscillation patterns in the DMN regions, are phenomenologically relevant. We found that the four data-driven states are associated with the four perceptual experiences corresponding to the experimental conditions. The state indictive of a dominant perceptual transition during the BR task was characterised by phase coupling between the precuneus (PCU) and the primary visual cortex (V1). We further found that the causal effect from the PCU to the V1 is a temporally cumulative effect along with the state lifetime of State 4. Taken together, our results suggest that the brain's intrinsic dynamics can influence visual awareness, possibly by triggering a diversion of the whole visual processing pathway.

There are a lot of controversies in existing literature regarding the sources of perceptual fluctuations during BR. Many advocate that the rivalry is resolved at the early visual stage, such as in the lateral geni-culate nucleus or the primary visual cortex, probably due to sensory bottlenecks[45,46]. But it is also well-known that the BR alternation rate can be modulated by object features, emotional or semantic loading embedded in the stimuli as well as higher-level cognition such as working memory and attentional control[47]. Therefore, many suggest that the observed activity in the early visual pathway is but an outcome of the feedback projections from the higher-level areas (such as the fronto-parietal regions), which is where the NCCs actually lie[48,49]. According to them, the higher-level regions suppress unfavourable stimuli representation in the early visual pathway in order to maintain a congruent conscious experience[11]. These high-level regions have been localised to frontoparietal regions by previous functional and structural MRI studies, including the frontal eye field, superior/middle/inferior frontal gyrus and superior/inferior parietal lobule (SPL/IPL)[50–52]. Especially the right-lateralised SPL was suggested to play a causal role in the BR transition by a Transcranial magnetic stimulation (TMS) study[50]. SPL is not investigated in our study, but parts of our results hinted at its relevance. Our critical state, State 4, was characterised not only by an increased coupling between PCU and V1, but also between PCU and rIPL. Although the IPL is usually considered to be part of the DMN, it is anatomically adjacent and functionally coupled to the SPL which is a core of the posterior attentional control system[53,54]. Despite its plausibility, the high-level control hypothesis cannot explain why the perceptual transition keeps occurring when a stable percept has been reached. In addition, it is difficult to dissociate the cause and effect of the BR perception in practice. Given the poor temporal resolution of fMRI, the frontoparietal regions or more specifically the right-lateralised SPL, as often reported in the BR literature[13,50,51], might get involved by its association with an attention effect which inevitably intertwines with an involuntary perceptual change. It has been argued that the frontal regions' involvement during BR may also reflect introspection and action to perceptual transitions, rather than being a real cause[55,56].

Our study provides a novel perspective as to why regular perceptual transitions may occur during BR. We highlight the possible influence of the temporal unfolding of the intrinsic large-scale cortical dynamics, which could bias the binocular signalling gains, perturb the current equilibrium, and eventually lead to a diversion of visual stream. Vidaurre et al. (2018) found that the intrinsic transient states exhibited in the posterior subdivisions of the DMN were characterised by high power and total coherence in the alpha (8–12 Hz) range[17]. It was also suggested that pre-stimulus posterior alpha power can modulate primary visual processing, likely by re-shaping the functional architecture

of the brain network when preparing for upcoming processing[57]. These findings support our hypothesis that the DMN's activity may serve as a source of "perturbation" to a bistable visual system during BR, thus introducing a bias towards what external information gets released from the overloaded visual channel.

Although the DMN and V1 are situated at the two opposite ends of the brain's functional gradient[22], our result suggests that they may achieve signal coupling via the PCU. To further understand this signal coupling, we derived a Granger-causal effect of the PCU to V1 from their EEG signals at the same time windows. We found that, during the critical state, the stronger the top-down effect from the PCU to V1, the quicker the individual's perception alternated. However, this relationship only existed at the group level, but not for trial-by-trial prediction; therefore, it only establishes the behavioural relevance of the Granger-causal effect of PCU to V1, but did not provide enough evidence for us to claim that the effect from PCU to V1 means either signal adaptation or suppression, as both could potentially lead to a trial-by-trial prediction. That neither state properties nor Granger-causality from PCU to V1 could predict trial-by-trial performance may just be due to a lack of statistical power; however, it could also be a true negative. Namely, the intrinsic brain dynamics may reflect more of individual-specific biological regularities, rather than "on-line" cognitive control. In fact, the influence of voluntary top-down control over BR is known to be limited[58,59]; and BR has been reported to have notable inter-individual differences, depending on the individual's gender, psychological predispositions, neurotransmitter levels and even handedness[47,60]. Twin and genome-wide association studies have showed that the individual differences in BR can be attributed to genetic factors[61,62]. These suggest that BR may be bound to certain biological constraints of the brain's general state, while participants' involuntary top-down control over the BR rate (limited but possible; such as paying more intrinsic attention could help by strengthening perceived contrast of the image[58]) might have exerted its influence rather indirectly by altering the intrinsic state of the brain. Although the PCU has been associated with visuospatial integration and imagery[63], and its role in BR transition could be seen as a sign of voluntary cognitive processes, either attentional control or memory retrieval[64,65], we rather propose that the PCU-V1 coupling might be an underlying pre-conscious process, an indication of how intrinsic dynamics exert influence onto information encoding in the primary visual cortex. Previous studies have suggested the PCU's role in consciousness. For example, recent literature using graph theoretic metrics and information theory has provided a new perspective for how we understand consciousness; it suggests that the posteromedial cortex (PMC; PCC/PCU), standing out as a hub for the whole-brain information exchange, affords the complexity that consciousness emerges from[66–68]. There have also been theoretical arguments hinting at the PCU's role in information integration between the internal and external worlds[69,70], a fundamental component of conscious processing[71,72].

Finally, we would also like to comment on the correspondence between the brain's spatiotemporal dynamics and the phenomenological mental dynamics. With abundant empirical evidence and theoretical arguments, it has been suggested that the spatiotemporal patterns of neural dynamics is the same thing as the instantiation of mental dynamics[16,73]. However, we would like to point out the differences between the neural substrates representing the content of the streams of consciousness, and the neural dynamics underlying consciousness which themselves are not necessarily represented in the conscious domain[74]. Although abundant evidence has suggested the DMN's significance to consciousness[20,24–26], it may not necessarily be the case that the neural dynamics of the DMN correspond to the individual's stream of consciousness in a simultaneous and linear fashion. However, the simultaneous and linear correspondence is always assumed in neuroscience studies for mapping between brain and mind. As the controversy about the DMN's function is accumulating, we probably need to rethink the legitimacy of such

correspondence we are tempted to make between the physical and mental levels of existence.

## Methods
### Subjects and experiment paradigm
The simultaneous recording of fMRI-EEG dataset for 20 young health participants is publicly available (https://doi.org/10.5061/dryad. bf1b1) along with a published paper[37,38]. During the experiment, 20 participants were presented with dichoptic stimuli, which were different images for the two eyes. To ensure separated eye views, a vertical divider was placed between the head coil mirror and the centre of the stimulus presentation screen located in the back of the Magnetic Resonance Imaging (MRI) scanner bore. In a Binocular Rivalry (BR) condition, the stimuli were rotating green and red checkerboard images respectively. In a perceptually matched control condition (i.e., the Replay condition), same rotating images, either the red or green checkerboard alternating in time, were presented to both eyes at any time. Participants reported their current perceptual state using buttons to indicate red, green or mixed percepts. The mixed percept is a transitional phase between the two stable (red and green) percepts. The order of green and red visual stimuli for two eyes in the rivalry condition was counterbalanced between the left and right eyes across participants and their multiple experimental sessions.

For each participant, each experimental session/condition of rivalry or replay consisted of 5 consecutive 42-second blocks of continuous stimulus presentation followed by 12 s of rest, and 5 total blocks for each stimulus type. Each participant repeated the BR and RPL sessions 5–7 times to increase replicability. Importantly, two different replay conditions were employed: smooth replay and instantaneous replay. The smooth replay condition was recommended to use for analyses as opposed to the instantaneous replay condition according to the previous literature[56]. In the smooth replay condition, a smooth, expanding wedge was presented to approximate the gradual perceptual transitions during the real rivalry condition, between the switches from green to red (or from red to green) percepts. Specifically, a small wedge of the target checkerboard would smoothly expand to cover the old one during the course of one second. More detailed description of the experimental design was presented in the original article[38].

### Simultaneous EEG-fMRI data acquisition
The EEG-fMRI data were acquired and shared by Jamison et al., (2015) and Roy et al. (2017)[37,38]. We will present here the key parameters, while full descriptions of data acquisition have been presented in the above original papers. Electroencephalogram (EEG) and electrocardiogram (ECG) were recorded using a 64-channel MRI-compatible amplifier (BrainAmp MRplus, Brain Products). All signals were referenced to an electrode at the FCz position, and sampled at 5000 Hz. Electrode impedances were made sure to be all below 20 kOhm.

Structural and functional Magnetic Resonance Imaging (MRI) were acquired using a Siemens Skyra 3 $T$ scanner with a custom, high slew-rate gradient insert developed for use in the Human Connectome Project. The whole-brain blood-oxygen level dependent (BOLD) functional data were acquired using a typical gradient-echo (GE)-Echo Planar Imaging (EPI) pulse sequence (flip angle/FA = 90°, repetition time/TR = 2200 ms, echo time/TE = 30 ms, 3 mm isotropic voxels, 36 axial slices, with fat saturation pulse). 129 volumes were acquired for most of the participants, which covered the whole experiment and aligned temporally with the EEG recording. There were 7 participants who had longer scanning sessions (with 130–141 volumes), but no participant's data were discarded. During the scanning session, the timing of each volume acquisition was recorded and used for event-related activation study. As a convention, the first 5 volumes were considered unreliable due to the initiation of the scanner and have been excluded for the following processing.

## EEG data preprocessing

Data was primarily preprocessed by the data distributors with standard pipelines which have been detailed in published articles[37,38]. To recapitulate, the raw EEG data were re-reference to the average of all channels, downsampled to 250 Hertz (Hz), band-pass filtered between 0.5 and 30 Hz. Gradient artefacts were removed using a PCA-based optimal basis set (OBS) algorithm, and cardioballistic artefacts were removed based on a combination of ICA, OBS, and an information-theoretic rejection criterion[37].

## FMRI data processing

Apart from the preprocessed steps already carried out by the data distributors, which included slice-timing correction, motion correction, co-registration, normalisation and spatial smoothing, we extracted the signals (the first eigenvectors) from the white matter (WM) and cerebrospinal fluid (CSF) by using standard template masks with a threshold of 0.7, and treated them as non-neuronal confounds, along with 6 movement parameters. The nature of the BR condition resembles a resting state where the participant typically stares at an invariant fixation point, while being more susceptible to movement since continuous responses are required here, therefore we paid great attention to the motion confound. Fast motions were inspected for each scan of every subject. Volumes with >3 millimetres (mm) displacement (i.e., the voxel displacement resulting from the combined effect of individual translations and rotations) compared to the previous volume are considered to be contaminated by fast motion and were deweighted in the GLM design using the ArtRepair toolbox[75]. Overall, bad volumes comprised on average 5.12% (STD = 7.05%), and in the worst case 34% (44 volumes out of 129) of a person's data in one scanning session.

The procedure of ArtRepair motion scrub is described below. For the following fMRI data processing, the main GLM for statistical inferences has been estimated twice, once with the ArtRepair and once without. The logic of the ArtRepair treatment is to deweight the contribution of the bad-motion volumes in a general linear model (GLM) estimation. To determine if the treatment was successful, a global quality metric of the range of contrast estimates over all the voxels within the standard brain mask was provided. For good contrast estimates, the mean of the contrast image should be near zero and the mean of the residual-sum-of-squares/variance (ResMS) image should be small (https://cibsr.stanford.edu/tools/human-brain-project/artrepair-software.html). If the post-hoc global quality metrics suggested that the original contrast estimate was actually better, no ArtRepair treatment would be applied to that session's data. As a result, there were 51.84% contrasts having their standard deviations (STDs) reduced (by 8.08% on average) after the ArtRepair procedure.

All fMRI statistics were carried out with the SPM12 toolbox (https://www.fil.ion.ucl.ac.uk/spm/) in MATLAB. We used the moment of subjective reports as the critical timepoint and created the event-related BOLD response, which was modelled by convolving the canonical haemodynamic response function with a spike function (1 at the event moment, 0 otherwise).

To fit the fMRI timeseries of each scanning session, the modelled event-related BOLD responses associated with the three main effects (dominant green percept, dominant red percept and mixed percept) were taken as the main regressors, along with the non-neuronal confounds (the first six principal components extracted from WM/CSF and six movement parameters) and the block effects (the 5 task blocks). To increase reliability, each participant was scanned in multiple sessions, with about half (~6) of the sessions under the BR condition and another half under the RPL. Therefore, the first-level GLM for each participant was modelled with a mixed-level factorial design, with percept-generation conditions (BR/RPL) and perceptual types (dominant/mixed) as two factors. The global session effects were specified as nuisance covariates (i.e., separate columns of identity vectors indicating different scanning sessions). With the GLM, the coefficient

(beta) values associated with the main regressors were then contrasted with each other to reveal the effects that different experiment conditions induced on brain activity. Contrasted conditions were 1. BR vs. RPL for dominant percepts (red or green), 2. BR vs. RPL for the mixed percept, 3. Dominant vs. mixed percepts in BR, 4. Dominant vs. mixed percepts in RPL, 5. Difference of the contrast dominant vs. mixed percepts, comparing between the BR and RPL condition (F contrast), 6. The red vs. green percept (for sanity check). One sample $t$-tests against zero for finding baseline activation for each condition were also conducted in order to provide priors for the EEG source reconstruction during the corresponding trials.

For the group-level GLM, a random effect (RFX) design was used, with random effect being the intercept of within-subject GLM fitting, and fixed effect being the main effects/contrasts of interest. Specifically, weighted beta coefficient maps from individuals were fed into a one-sample $t$-test to make group-level inferences. To address the multiple comparison problem, cluster-level inferences were used and corrected for the family-wise error rate within the framework of Gaussian random field implemented in SPM (version 12)[76]. Clusters were defined by a default voxel-level threshold of 0.001 (uncorrected). The family-wise error rate was controlled at the cluster level, and a threshold of $P$-corrected < 0.05 was used to determine significance among clusters.

## EEG data processing

**Evoked Responses at the sensor level.** With the preprocessed data, we then conducted the evoked EEG response analysis as a sanity check for the data. It was conducted on the platform of MNE-python (version 0.23.4)[77] (https://mne.tools/stable/index.html). To get the event-related potentials, we epoched the pre-processed data using a 1 s time-window around the perceptual report starting at −0.5 s (before report) and ending at 0.5 s (after report). While epoching, a baseline correction was applied to remove the average activity in the range of [−0.5, −0.4] (i.e., 0.1 s before the critical period) from the whole time-series. We acknowledge the limitation of the response-locked analyses (as constrained by the experimental design), as the epochs incorporate a mixture of perceptual and motor processes. If we accept RPL as a perceptually matched condition, any contrasted differences between the BR and RPL should elucidate the endogenously driven neural features that are distinct in the BR condition, rather than the noise or motor-related features that are assumed to be common to both conditions.

For each subject, evoked activity was calculated by averaging these epochs across all sessions for that subject. Global Field Power (GFP) for each condition, was calculated as the standard deviation of the voltages across electrodes at each time-point (after removing the ECG electrode and bad electrodes identified for a particular subject). Statistical analyses were performed using one-sampled cluster $t$-tests to compare the conditions for each subject, as detailed in MNE-python statistics toolbox.

**Evoked Responses at the source level.** The source-level EEG data processing was conducted with the SPM12 toolbox (https://www.fil.ion.ucl.ac.uk/spm/). After converting the data to the M/EEG SPM format, the continuous data spanning across the whole session were first chopped to many short epochs, which corresponded to the 1-second period before a perceptual change, as reported by the subject. Structural MRI images of each participant were segmented and were used to co-register with the electrodes. As a result, the EEG electrode positions were projected to each participant's MRI space with a rigid body co-registration by minimising the differences between the landmark head points (along with other electrode positions) and the scalp mesh reference. We used the 3-shell spherical method for the forward model. Scalp, skull and brain tissues were segmented. Their conductivity ratios were specified to be 1, 1/80, 1 (conductivities being 0.3300, 0.0042 and 0.3300 Siemens/metre, respectively). To do the inverse computing which is to estimate the signals of the dipoles (or sources) from the

observed EEG data on the scalp, we used different model configurations including different forward model assumptions i.e., using the minimum norm estimation that assumes independently and identically distributed (IID) sources or the Standardised Low-Resolution Electromagnetic Tomography (sLOR); using signal hanning or not; and using fMRI priors or not. These models were all estimated and compared against each other. The best model configuration with the highest model evidence was chosen for further analyses[78] (see Supplementary Fig. 2 and Supplementary Table 1 for the result of the model comparisons for the inverse modelling configurations).

The usage of fMRI priors in the source reconstruction is as follows. According to the condition type, i.e., stimuli types (BR vs. RPL) × perceptual types (dominant vs. mixed) that the reported perceptual change belongs to, the corresponding group-level fMRI (significant) activation clusters were applied to constrain the sources' covariance for the model inversion. The usefulness of fMRI priors was then determined by variational free energy (model evidence)[78,79], in the comparison with the model inversion without the fMRI priors.

The model inversion was conducted respectively for the following frequency bands: delta (0.5–3.5 Hz), theta (4–7 Hz), alpha-1 (7.5–9.5 Hz), alpha-2 (10–12 Hz), beta-1 (13–23 Hz), and beta-2 (24–30 Hz), according to the conventional International Federation of Clinical Neurophysiology guideline[80].

Evoked (epoch-averaged) responses on the source level were analysed for each frequency band, and for finer time windows. According to previous sensor-level evoked responses which had been compared between conditions, we knew that the time window from 200 ms to 400 ms prior to a subjective report marked the biggest difference between an endogenously (BR) and exogenously (RPL) induced perceptual change; therefore, we divided the 1-s epoch into 5 shorter windows with an equal length of 200 ms and looked into the evoked responses for these time windows. The images for evoked response were saved as nifti files and in standard MNI space. They were then contrasted between conditions, and non-parametric permutation test were conducted for the group-level inferences with the SNPM toolbox (https://warwick.ac.uk/fac/sci/statistics/staff/academic-research/nichols/software/snpm).

### Selection of time window and regions of interest for the TDE-HMM dynamic process modelling

Instead of modelling the dynamic process for the whole brain during the whole experiment, we narrowed down the time window and regions of interest (ROIs), in order to increase the method's sensitivity and avoid overfitting[17]. Our previous evoked-response analysis helped us to narrow down the time of interest to a window between [−400, −200] ms before a subjective report of perceptual change, which is when the pre-response ERPs showed the biggest GFP difference between BR and RPL. However, the endogenous neural activity triggering an upcoming transition might take effect even earlier than that[40]. Therefore, we modelled the dynamic process in the 1-s (with 100-ms post-response padding) window before every report of a perceptual change. The 1-s window was chosen as a trade-off between having enough sample points and having enough specificity to the targeted events. It has been suggested in the literature that endogenous perceptual disambiguation is associated with posterior parietal activity 50 ms before the onset of a bistable stimulus;[81] and the upper limit of reaction time to it is about 600 ms[40,82], while the upper limit of pure motor execution is about 150 ms[40,83]. Therefore, the 1-s window should be able to cover the whole dynamic neural process towards a perceptual transition and a little further before. 25767 trials were generated in total, considering all 4 conditions: BR/RPL (dominant) (i.e., transitions from a mixed to a dominant percept in the BR or RPL setting; $n = 7795/7148$), and BR/RPL (mixed) (i.e., transitions from a dominant to a mixed percept in the BR or RPL setting; $n = 5786/5038$). As the algorithm computes the latent states recurring across all

timepoints of all trials, the state delineation is assumed to be driven by the endogenous neural states which have a robust spatiotemporal structure recurring through time[9,17,19,22,84].

Given our research interest in the default mode network (DMN) we constrained the subsequent modelling within the 8 regions: bilateral parahippocampal gyri (HP), bilateral inferior parietal lobules (IPL), anterior cingulate cortex (ACC), posterior cingulate cortex (PCC), precuneus (PCU), and the primary visual cortex (V1). To ensure experimental sensitivity, we extracted signals from the significant peak voxels (confined within the DMN) from the previous source-level evoked-response analysis. The contrasts used for determining the regional involvement in the task were BR (dominant) vs. RPL (dominant), BR (mixed) vs. RPL (mixed) and the interaction effects between the two factors; both contrast directions were considered. Hence, 1-second (+0.1 s post-response padding) source signals from these coordinates were extracted for all trials (trials being the 1-s epoch before a subjective report of a percept change). When multiple coordinates were identified within a same ROI (identified by automated anatomical labelling), the average of their signals was used as the representative signal of that ROI. The anatomical labels for the peak coordinates were identified using the Talairach Atlas (http://www.talairach.org/), upon a conversion to the Talairach space. The list of the peak voxel coordinates used for this purpose are presented in Supplementary material Table 2. The full results of all significant voxels/clusters upon all contrasts are available from an online repository: https://htmlpreview.github.io/?https://github.com/Aubrey-Lyu/BR-project/blob/master/Analysis-3_EEG/results/evokeResponse_result_table_permutationtest.html.

### Time-delay embedded Hidden Markov Model

The Hidden Markov Model (HMM) as a general framework assumes a hidden sequence of a finite number of states which drives the observed time series[85]. In practice, the algorithm adopts a probabilistic model which infers the probability of each state being active at each time point (order = 0). In the present study, the states were estimated at the group level, but the information about the state probability is specific to each subject. We used the Time-delay embedded HMM (TDE-HMM) approach to search for the hidden states[17].

In this approach, the observation model is described by the auto-covariance (or lagged cross-covariance) of our specified regions within a sliding time-window of 60 ms. For the 8 specified ROIs and the 250 Hz sampling rate of our data, the observed neural activity over a window of 15 time points centred at t, is described by an auto-covariance matrix of 15 × 8 by 15 × 8. We chose auto-covariance over Gaussian distribution as our observation model, because for EEG data, the frequencies and phases bring richer information than the amplitude alone. The auto-covariance matrix can effectively capture patterns of linear synchronisation in the oscillatory activities (i.e., "state-wise phase-locking"). To avoid overfitting, the HMM was trained on a principle-component-analysis (PCA) decomposition of the auto-covariance matrices. As recommended, we used twice the number of ROIs for the number of PCs (i.e., 16 PCs) (https://github.com/OHBA-analysis/HMM-MAR/wiki/User-Guide). The PCs explained 96.1% of the data variance for this dataset on average (lowest and highest across subjects are 94.5% and 97.0%). For model inference, stochastic inference was used to alleviate the computation time as opposed to the standard variational Bayes. After finding the states based on the data's transient oscillatory patterns, we then obtained the state-specific spectral properties, i.e., the power and coherence, from the multivariate auto-covariance matrix in each state's observation model. Having estimated the power and spectral coherence for each state, we then factorised them into different frequency modes for finer analyses by using a non-negative matrix factorisation (NNMF) algorithm. The whole pipeline of HMM analyses was facilitated by the HMMMAR toolbox[17]: https://github.com/OHBA-analysis/HMM-MAR.

We provide validation of the HMM states by reproducing the states on randomly half-split data. The procedure is as follows: we randomly selected half of the subjects' data, ran the HMM analyses on each half, matched the states and measured the states' similarity between the two runs. These procedures were repeated 5 times. To match the states across runs, we ordered the states in a way to ensure the maximum similarity between the states in different runs. The state similarity is measured with Pearson correlation and presented in Supplementary Fig. 14a. We then performed statistical testing on the consistency of states across runs. This is to ensure the matched states reliably represent the same process. For example, if the similarity between the matched states is significantly higher than any two non-matched pairs, it means we correctly identified the state in different runs (Supplementary Fig. 14b).

### Cross validation

The cross validation (CV) was conducted with the Scikit-learn toolbox[86] (https://scikit-learn.org/stable/) on Python. The fitting data were the Fractional Occupancies of the 4 hidden neural states for every trial. The target variable was the binary variable indicating whether the trial is transiting to a dominant or mixed percept. The model was trained separately for BR and RPL conditions.

We used the linear support vector machine as the estimator, and the default 5-fold CV. The regularisation parameter was selected from the range $[10^{-5}\ 10^{0}]$ with a step of 0.2 in the $\log_{10}$ scale. The result was reported in the regularisation parameter range where the CV score improved and converged, suggesting that this is the best prediction that this estimator could achieve. The relationship between the CV score and the regularisation parameter was depicted in the Supplementary Fig. 15.

### Granger causality between the PCU and V1

Granger causality is used in this study to establish the causal influence from the PCU to V1 in the BR dominant condition. Supposing $X_1$ and $X_2$ are two timeseries, $X_2$ is called to "Granger" cause (GC) $X_1$ if it can help us predict $X_1$ better than just using the past knowledge of $X_1$ itself. From the perspective of model comparison, we compared the following two models: the null model, which is an autoregressive model for the V1 timeseries (Eq. 1), and the augmented model, which also included the backshift datapoints of the PCU timeseries as additional predictors (Eq. 2). For model selection, we used Bayesian Information Criterion (BIC) to decide which model is better in terms of the balance between data fitting and model complexity. The GC effect size was approximated by the difference of the BIC (dBIC), i.e., the improvement of model evidence after additionally considering the cross-lagged effect of the PCU during every 20 preceding time points (in 80 ms). The GC was calculated for the original signals of PCU and V1 for every trial/epoch, by using the granger_cause_1 toolbox[87,88] released from the MATLAB Central File Exchange (https://www.mathworks.com/matlabcentral/fileexchange/59390-granger_cause_1).

Mathematical equations:

$$\mathbf{X_1} = \sum_{j=1}^{p} b_{11,j}\mathbf{X_1}(t-j) + \mathbf{E_1}(t) \tag{1}$$

$$\mathbf{X_1} = \sum_{j=1}^{p} b_{11,j}\mathbf{X_1}(t-j) + \sum_{j=0}^{p} b_{12,j}\mathbf{X_2}(t-j) + \mathbf{E_1}(t) \tag{2}$$

where $\mathbf{X_1}$, $\mathbf{X_2}$ are two timeseries for $t = 1...T$, $\mathbf{E_1}(t)$ is a white Gaussian random vector, $b_{11,j}$ and $b_{12,j}$ are the correlation coefficients respectively for the autoregressive model of $\mathbf{X_1}$, and the multivariate-autoregressive model between $\mathbf{X_1}$ and $\mathbf{X_2}$, for every backshift of $j$ within the maximal time-lag $p$. For model 1, we used the "best" $p$ adaptive to the V1 timeseries, for maximising the variance that can be explained by the null, i.e., autoregressive, model. Since we are agnostic to the time lag that the PCU (i.e., $\mathbf{X_2}$) is supposed to lead V1 (i.e., $\mathbf{X_1}$), for model 2, we estimated the models with all possible values for $p$ from 1 to 20 (leaving at least 20 time points for the coefficient to be robustly estimated). The improvement of model 2 relative to model 1 was parameterised by dBIC. Averaged dBIC across all $p$ from 1 to 20 was calculated as an unbiased evaluation of the GC effect.

### Statistical testing for the behavioural relevance of the neural states

The lifetime of each state was used to correlate with the pre-switch perceptual duration with an Analysis of Covariance (ANCOVA) at the group level. For the ANCOVA model, the dependent variable was the perceptual duration, and the independent variables are the lifetime of the state and the categorical variable condition. State lifetime can be obtained from the Viterbi path, which is an estimate of the most probable hidden state path given the observed data (See detailed instruction from the online user guide: https://github.com/OHBA-analysis/HMM-MAR/wiki/User-Guide). An equivalent measure to the lifetime is the Fractional Occupancy (FO), which is the proportion of dwelling time on a certain state given a period of time.

We also investigated the relationship between the GC (PCU-V1) and the stable percept duration. The GC effect, which is indicated by the GC effect size, approximated by dBIC (as recommended by the toolbox granger_cause_1: https://www.mathworks.com/matlabcentral/fileexchange/59390-granger_cause_1), was then used to correlate with the stable percept duration at the group level. During this investigation, we found that the relationship is dependent on the presence of a particular hidden state. Therefore, we improved the model by adding the interaction effect between the lifetime of the state and the GC effect (size). Considering the non-normal distribution of the data, we used the generalised linear model (Gamma distribution for error fitting) as well as the Akaike Information Criterion (AIC) for model selections throughout the paper. The whole process of statistical testing was implemented in R, with the lme4 and rcompanion packages. We have uploaded all the code and data used in this project to Github (https://github.com/Aubrey-Lyu/BR-project/tree/master) for public review.

### Reporting summary

Further information on research design is available in the Nature Research Reporting Summary linked to this article.

## Data availability

All data needed to evaluate the conclusions in the paper are previously published[38] and made publicly available in the repository: (https://doi.org/10.5061/dryad.bf1b1). Source data displayed in the figures of the present study is provided. Source data are provided with this paper.

## Code availability

The code to replicate all the analyses can be obtained from the GitHub repository: https://github.com/Aubrey-Lyu/BR-project/.

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

## Acknowledgements

Acknowledgements are warmly given to the research team led by Bin He from the Department of Biomedical Engineering and Department of Psychology in the University of Minnesota for data collection and deposition; and Abhrajeet V. Roy for the E-mail correspondence regarding data clarification. This work was supported by the Canadian Institute for Advanced Research (CIFAR) (RCZB/072 RG93193; to D.K.M. and E.A.S.); the National Institute for Health Research (NIHR), Cambridge Biomedical Research Centre and NIHR Senior Investigator Awards (to D.K.M.); the British Oxygen Professorship of the Royal College of Anaesthetists (to D.K.M.); the China Scholarship Council and Cambridge Commonwealth, European & International Trust (to D.L.); the Stephen Erskine Fellowship, Queens' College, University of Cambridge (to E.A.S.). Computing infrastructure at the WBIC High Performance Hub for Clinical Informatics was funded by Medical Research Council research infrastructure Award No. MR/M009041/1.

## Author contributions

D.L. conceived the presented project. D.L. acquired and sorted the data. D.L. and S.N. conducted the EEG evoked-potential analysis. D.L.

conducted the fMRI, HMM and behavioural analysis. D.L and E.A.S. initiated the paper drafting. D.L. wrote the manuscript with input from S.N, in consultation with E.A.S. and D.K.M. D.L. and E.A.S. took the lead in paper editing. All authors provided critical feedback and helped shape the research, analysis and manuscript.

## Competing interests

The authors declare no competing interests.
