## [Peer Review File · Nature Communications]

Intrinsic brain dynamics in the Default Mode Network predict involuntary fluctuations of visual awarenessREVIEWER COMMENTS

Reviewer #1 (Remarks to the Author):

Summary:

This study aims to explore the relationship between (conscious) perception/visual awareness and the intrinsic dynamics of large-scale networks (esp. the DMN). More specifically, the objective is to determine whether such intrinsic dynamics serve a role in perception/awareness. To do this, the authors utilize data from an EEG-fMRI experiment that had two main conditions: 1) binocular rivalry (BR) and 2) perceptually matched replay (RPL). First, fMRI contrasts resulted in the observed activation within cortical regions that are considered to be part of the DMN. These results constrained, or were used to guide, subsequent EEG analysis. EEG results indicated a particular time period where endogenous vs. exogenous information could be distinguished (-400 - -200 ms), as well as the cortical sources associated with this distinction (including several regions of the DMN). Within these regions, HMM (with 4 states) was used to probe intrinsic dynamics (e.g., switching rate and fractional occupancy), whereby state 4 was found to be associated with the BR (dominant) condition and suggested to be important for the prediction/production of perceptual fluctuations. Further spectral considerations of the states (with emphasis here on the alpha band) indicated a precuneus-V1 phase coherence for state 4 which was also associated with faster perceptual transitions. The authors conclude that these results point to the idea that intrinsic dynamics may not be epiphenomenal, and rather predictors of perception (and its fluctuations), as shown within the DMN.

General Comments:

Overall, given the missing information about the endogenous features/sources of perception, this study is important in helping to address that very point. This paper is manifold in its objectives, which include: 1) investigating intrinsic dynamics and their relationship to perception, and 2) establishing a role for the DMN (specific cortical regions) in endogenously-driven perception/awareness. The findings suggest that the DMN might be involved in task-related activity, as is shown by fMRI activation for endogenously- vs. exogenously-driven perception. Moreover, they help to identify a time period that is important for such a distinction, as well as the implicated neural correlates. These considerations will help to address gaps in the literature related to (conscious) perception/awareness.

'Major' Comments:

These may be some points to address:

There are methodological details that, although were pertinent to the previously published articles, are also important here:

1. The results of the power analysis that led to 20 participants were not included or described (even if previously published) (Is there sufficient power for this study?)
2. Presumably, gaze was tracked throughout the entirety of the experiment, but this was not mentioned or described
3. The number of predictors in the creation of GLMs for fMRI analysis was not clear, nor what the predictors were used to model
4. Several contrasts were conducted, though thresholds and multiple comparison details should be further emphasized in the corresponding sections
5. The analyses that were conducted at the individual vs group level should be more clearly indicated / separated in the text
6. One methodological consideration revolves around ROI analysis and why the authors did not first identify key DMN regions and extract data from these regions / use only these regions to guide their analysis (given the importance of the DMN here)
7. A brief Methods section included in the main text would be useful in guiding the reader, in addition to have a full-length version of the Methods in the supplementary section

There are also some theoretical points to address:

1. Although the fMRI analysis suggests that DMN regions were active in distinguishing between endogenously- vs. exogenously-driven perception, additional analysis might be considered to determine the direction of the effect in order to determine the effect that is driving the activation (for example, is it the increased endogenous-related or suppressed exogenous-related activity that is driving the activation in those regions?)
2. Given the involvement of the DMN and the link found between PCU and V1, it would be good to explore how the other DMN regions affect (if at all) the PCU > V1 relationship for perceptual transitions (e.g., through dynamic causal modeling?)
3. Examining cortical modulations stemming (esp. temporally) from V1 in the endogenous vs. exogenous conditions would help to provide further clarification or insight into how the results lead to a 'diversion of the visual processing pathway', as well as how this leads to visual awareness
4. Although the DMN is the large-scale network of focus here, a short mention of whether it is the only source of perturbations to result in perceptual fluctuations and/or how the other large-scale networks mentioned might be involved in producing visual awareness

Minor Comments:

1. Abstract: "causal effect" could be further clarified in terms of Granger causality
2. fMRI activation study (this subheading might suggest that an entirely separate study was conducted): "over- lapped" should be changed to 'overlapped'
3. Fig. 2/legend: Acronyms should be defined in full when first mentioned (e.g., ICN, BM, FCPN)
4. Hidden neural states within DMN ... subsection: the HMM acronym should be defined (this acronym is only defined in the supplementary section)
5. Fig. 5 legend: "pair-wise" should be 'pairwise', "(clock-wise)" should be '(clockwise)'
6. Fig. 5 legend: font size of cortical region names on the circular bundle plots is quite small
7. PCU-V1 phase ... subsection: "Therefore, we next focused into.." – the 'into' should be changed to 'on' or similar
8. There are two "PCU-V1 phase coherence and Granger Causality predicts stable percepts" subheadings
9. The acronym for Granger Causality should be provided (GC)
10. Mentions of results found in supplementary section should be expanded into statements about what those results show, in addition to where in the supplementary section they can be found
11. ("All statistical models have been presented..") – the first letter of the first word in these brackets should be lowercase
12. Discussion, paragraph 1: "namely, the interaction of them" should be "namely, this interaction"
13. There is considerable focus in the discussion on the role of SPL, despite the results focusing on PCU/V1
14. "on the group level" should be "at the group level"
15. "-400-200 ms" suggests an overall 600-ms duration (if '-400 - -200 ms' is what was meant, this should be indicated where appropriate in the text)
16. Supplementary: ">3mm displacement" might be amended to "translations and/or rotations of > 3mm/°, respectively", given that both factors (indicated in mm or °) are accounted for in motion correction
17. Supplementary: "on Python" could be "(implemented) in Python"
18. Supplementary: here it is "PCu", whereas it is "PCU" in the main text
19. Supplementary, sFig. 3: no regions are labeled on the brains, nor are they mentioned in the legend (and scale label font size is quite small – same for sFigs. 4, 10, 11, 12, 13, 14)
20. Supplementary, Table 2: in the main text, this table is actually referred to as Table 1 when the coordinates are discussed; there is no mention of why there are different numbers of coordinates for each of the regions

Reviewer #2 (Remarks to the Author):

Review of the manuscript: Perturbation or Function? Intrinsic brain dynamics in the Default Mode Network predict involuntary fluctuations of visual awareness

The authors of this manuscript presented binocular rivalry stimuli and non-rivalrous replay variants and used simultaneously recorded fMRI- and EEG data to investigate the causal role of the endogenous dynamics of brain areas from the Default Mode Network (DMN) for spontaneous endogenous perceptual reversals.

They applied several sophisticated data analysis methods (e.g. the time-delay embedded Hidden Markov Model or Granger Causality) in order to isolate four transient brain states with condition specific occurrence frequencies and durations. As main findings the authors report the isolation of one specific transient brain state, mainly based on activity of brain areas from the DMN and a quasi-causal relation between the precuneus as one of those brain areas and the primary visual cortex.

Overall the manuscript touches the ongoing discussion about neural correlates of consciousness, the tightly related phenomenon of perceptual instability during binocular rivalry and the question about the underlying brain processes and the contributing brain areas. In the last decades the debate about the driving forces underlying spontaneous perceptual reversals during observation of binocular rivalry stimuli and ambiguous figures has been focused on the opposing bottom-up and top-down explanatory approaches. From this point of view the present manuscript contributes an interesting novel perspective together with a similarly interesting data base.

Having said this in the first place, I listed a number of shortcomings of the current version of this manuscript that need to be addressed before acceptance:

General Points

(1) Complexity

Frist of all the extensive set of data analysis tools makes the manuscript highly demanding and comes with a considerable danger to lose a large number of readers, simply due to complexity reasons. I regard it thus as urgently necessary to fulfill some minimal requirements to make the manuscript easier to understand.

1.1. Resolving Shortcuts

One urgent suggestion is, of course, that EVERY shortcut used should at least once be spelled out. In the present manuscript there are many examples, where this is not the case, e.g. “ICN”, “BM” or ECG (in the supplementary material).

1.2. Provide intuitive explanations

The authors should also provide for any statistical tool 1 – 3 sentences that explain the basic idea. I was personally unfamiliar with many tools and had to stop reading several times, in order to look for myself for the underlying literature and receive a basic idea, what certain methods, where only their names were mentioned, do. A reader should be able to go through the manuscript with at least a basic understanding of the logic underlying a certain analysis step, without extensively studying secondary literature.

To be honest, the help of the extensive supplementary material, provided by the authors, was also very limited in this respect.

(2) “causal effects”

All of us know that our current knowledge about brain processing does not really allow making strong statements about causation. This was even emphasized by Dr. Granger, when

he presented his Granger Causality. The authors should thus be careful with making strong statements about causality (e.g. page 23) ...

(3) Roadmap

Given the complexity of the data analysis it would be nice if the authors could provide a more general roadmap displaying the basic steps and their purposes. The authors tried this already in their sFig. 1. However, an additional and much more general type would be helpful.

Example:

Method: ICN => Purpose: identifying contributing component brain networks being involved in certain perceptual processing steps (or experimental conditions)

Method: HMM => Purpose: identify and characterize local transient brain states

Method: Granger Causality => Purpose: identify hierarchical relations between brain regions

(4) Multiple Testing

The authors performed a huge number of statistical tests. They report about FWE corrections. However, I doubt that this is applied on a global level, for really all tests realized in the present analysis.

If I am correct with this assumption, I'm fine with it, if the authors make an explicit statement about the statistical limitation that comes with multiple testing.

(5) The endogenously/ exogenously dominant percepts are used interchangeably with BR/RPL. For ease of reading it might be good to stick to either using endogenous/exogenous or BR/RPL after defining which condition is which

(6) Time course of perceptual reversals

Understanding the neural processes underlying spontaneous perceptual reversals, the precise time point of the reversal event is necessary, e.g. in order to know which activity is pre-reversal and which activity is post-reversal and probably even secondary to the reversal-related neural processes. Narrowing down the reversal time point, however, is difficult – in the first place, because the reversal is an endogenous brain process and the experimenter only gets to know about it, when the participant communicates her/his first person experience of the endogenous event. Using iris diameters or eye-movement patterns in non-response paradigms (Brascamp et al. 2018) is an interesting perspective but apparently provide not precise enough time stamps.

Several years ago we have designed the so-called onset-paradigm (a simpler version primarily introduced by O'Donnell et al. 1988) to study the chain of processes underlying a spontaneous perceptual reversal of the Necker cube and other ambiguous figures (for a review see Kornmeier and Bach 2012). In a nutshell, the basic idea was to present the stimuli discontinuously (800 ms on, 400 ms off), and therewith to synchronize the onsets of perceptual reversals with the onset of the stimulus and thus use stimulus onset as time reference for the reversal event. This allowed us a remarkable temporal resolution of the processes underlying Necker cube reversals, which we then compared with a related chain of processes when disambiguated cube variants reversed in a computer-driven manner. We found remarkable similarities but also interesting differences. Both our group and Michael Pitts and colleagues demonstrated that the basic idea of the onset paradigm also works with binocular rivalry stimuli and moreover, that similar results can be measured (for a review see Pitts and Britz 2011), although we were not as successful to show this for BR stimuli (O'Shea et al. 2013).

Relevant for the present manuscript are the following findings:

Our group (Ehm et al. 2011) and concurrently and independently Britz et al. (Britz et al. 2009) found right-hemispheric EEG activity in the gap immediately before (around 100-300 ms) an endogenous perceptual reversal. The onset-paradigm thus clearly allowed confining this right-hemispheric activity to a pre-reversal time window. This may be a relevant finding for the authors' discussion in the second paragraph of their page 22.

Further, we found that reaction times jitter intra-individually with ± 100 ms (interquartile range) and discussed this in relation to the reversal event (Kornmeier and Bach 2012). The authors of the present manuscript use participants' manual response as reference time point. They thus need to take into account that the reversal event had occurred about 100 ms earlier. The zero point in their graph is thus not the time point of the reversal but rather the time point ± 100 ms after the reversal. Given our data, the authors may even consider shortening their temporal ROI (in a follow-up analysis or follow-up experiments), which may increase the signal-to-noise ratio of the reversal-related signals. ...

Of course it is always a little problematic if a reviewer requires the authors to cite his own work. However, in the current case this may be defensible: The authors made a big effort to narrow down the processes underlying spontaneous perceptual reversals both in time (focusing on causal aspects) and in space.

In our view our onset paradigm provides the most precise temporal description of the processing chain underlying spontaneous perceptual reversals. And although there are significant differences between classical ambiguous figures, like the Necker cube, and binocular rivalry stimuli, there seem to be also several similarities, particularly concerning processing chains, as Pitts and Britz (Pitts and Britz 2011) worked out in their review, but as also much earlier Blake and Logothetis pointed out in their seminal review (Blake and Logothetis 2002). Furthermore our findings have since been replicated by several labs around the globe (e.g. Pitts et al. 2007; Intaite et al. 2010; Sandberg et al. 2014; Kornmeier et al. 2019; Abdallah and Brooks 2020).

We thus think that this work needs to be at least discussed in relation to the authors' findings.

Major Points

Page 2: "...We know that external perturbations can interrupt or bias the global cascades of information propagation (e.g. visual masking and priming) ..."

In the sentences before the cited one, the authors elaborate about resting state brain activity. I think in the cited sentence they write instead about processing of sensory information. However, this is not entirely clear. By this reason it is also not really clear how visual masking and priming fit into this sentence. The authors should explain the ideas behind this sentence in more detail and add some references from the priming and masking literature.

Page 2: In 2001 Randolph Blake provided an encompassing review about BR. The authors should consider citing this profound work (Blake 2001), when they introduce the BR stimuli.

Page 3, end of paragraph 1: Sentence is difficult to read "...However, the intrinsic dynamics of brain networks may have a lot of interactions with, and by which modulate, the activity of local sensory regions."

Better: "However, the intrinsic dynamics of brain networks may have a lot of interactions with the activity of the local sensory regions, and as a result modulate said region."

Page 5:

- define “RSS”
- “Akaike Information Criterion“ => provide at least one informational sentence about what this measure does ...

Page 6: The present manuscript only indicates in a sub clause, that parts of the data have already been analyzed and the results published in another paper. The authors should make this more explicit. Moreover, they need to indicate what the major difference and “unique selling point” of the present data analysis in comparison to the previous publication is.

Page 6, Fig. 1: It is reasonable to represent data from red and green stimuli in the same colors (Fig. 1). However, it may be hard to distinguish for some of your readers (9% of men, 0.8 % of women). Consider using shadings in addition to color coding. Further, it is unclear to me why in Fig. 1d there are symbols (triangle and diamond) behind the boxplots. Does this address the color-blindness problem? To me it looks confusing and a simple shading would already do the trick with only little visual clutter.

Page 8, Fig. 2: Define FPCN and ICN

Page 8, line 2 of section “EEG evoked responses and source reconstruction”: “The EEG recording with its excellent temporal resolution gave us the opportunity to better locate the neural cause of a perceptual change...” How do we know if it is a cause? Could it not simply be a correlation?

Page 8, Fig. 2: BM14 is listed on the circular plot but not in the text next to it.

Page 10: What is the exact logic behind the selection of the time window for voltage map subtractions in Fig. 3d right.

Page 11, line 22: “... We targeted our dynamic analyses at the temporal window of the second (with 0.1-s post-response padding) before every perceptual transition as a trade-off between having enough sample points and having enough specificity for the targeted events.” This should read: “... before every response of a perceptual transition...”

Page 12: The arguments for the a priori choice of 4 states is not entirely clear to me. The authors wrote: “... We chose 4 states because firstly we were interested in the differences of the neural activity among the four experimental conditions...” Does this argument make sense if one can assume that each of the four (or more) microstates can occur within each of the four conditions? Exactly this is what I observe in Fig. 4 bottom right. This point needs more clarification!

Page 13, Legend to Figure 4: “... Across all the experimental conditions, signals from significant peak voxels within the same ROIs were averaged ... “ For me this sentence is not clear. Essentially the authors have spatial and temporal ROIs. I suppose they average across the data within a temporal ROI but I am not sure. This should be explained in more detail.

Page 14: The authors should clarify that a mixed perceptual transition means a transition from a stable percept to a mixed percept (if I am right) ... and the respective for dominant perceptual transitions ...

Page 18: “... In the BR literature, it is commonly believed that a switch of perception is

caused by a slip of equilibrium due to neuronal noise, and/or the constant decay of suppression that one neural population has over the other, due to habituation effects (10, 11, 44). Apart from these, attention effects have also been shown to play a role (10, 45). Therefore, the PCU→V1 top-down modulation could reflect either an intrinsically evoked attention effect, or an effect of the global adaptation/perceptual habituation. These two possibilities would lead to opposite trial-by-trial predictions. ... “

I cannot completely follow these considerations. There is indeed a debate about whether perceptual reversals are driven in a bottom-up manner or rather in a top-down manner. I am fine up to this point. However, attention-induced higher reversal rates are only reported if participants are explicitly instructed to volitionally increase reversal rates. The latter is more effective for classical ambiguous figures than for BR stimuli. However, without instruction, I do not see the case that higher reversal rates should be expected.

Page 20, Figure 6: Please define “dBIC”

SM, Equations 1 and 2: If you provide a mathematical formula you should also define the different variables used in this formula.

SM, Page 3: Please specify how gradient artifacts were removed from the EEG.

SM, Page 4:

- What is the meaning of ResMS
- What is the trial length for the fMRI analysis?

SM, Page 5: Was there an exclusion criterion for participants that show a lot of movement artifacts? (e.g. participants with more than 10 % of bad volumes are excluded from the study). In case you did not have this criterion, why so?

Page 5: „One sample T-tests against zero for finding baseline activation for each condition were also been conducted in order to provide priors for EEG source reconstruction.“

Better: “have been”

SM: Figure 2:

- You should explain what is meant by FE
- The text describes negative free energy. The ordinate shows positive values. Please explain this.
- You should add the following sentence to the legend of Figure 2: “The higher the Negative Free Energy, the more evidence the model presented.”

SM: Figures 3 and 4: The color bars have no units

SM, Page 6:

- Please provide a version of MNE-Python
- The link to the MNE-Python homepage is given twice

SM, Page 7: The forward model should be described in more detail. Did you use boundary element method or finite element method? How many shells/ tissues did you differentiate? What conductivities did you assume for your forward model?

SM, Page 7: There are dozens of different approaches to calculate inverse solutions. Please

specify your motivation to choose sLORETA and minimum norm estimates/IID. Why did you not select a more recently developed and more sophisticated inverse solver, e.g. multiple sparse priors (MSP) or cMEM? Beamforming could have been another suitable option since it performs well on noisy environments (e.g. in a scanner) and in the frequency domain.

SM, Page 7: Typo: “Signal hanging” probably means “signal hanning”.

SM, Page 8:

“...The Hidden Markov Model (HMM) as a general framework assumes a hidden sequence of a finite number of states which drives the observed time series (7). The data at each time point can be explained by a mixture of states, where the mixture weights are the state probabilities. ... “

The present manuscript is very demanding from a data analysis point of view. I am involved in EEG but not an expert for the analysis tools used here. Having had a look into Vidaurre et al. 2018 I – at least believe – that the concept of brain states is borrowed from Lehman et al.’s basic microstate idea. And as I understood the authors correctly, they assume such (micro)states re-occurring multiple times across a time series. If we understood the basic concept correctly, we should expect no superposition of brain states at one single time point, but instead distinct brain states for each data point. If this is correct, then the above-cited sentences are misleading.

Minor points

Page 4: “... Therefore, this paradigm had a 2×2 factorial design of dominant and mixed perception types for the BR and RPL conditions ... “

You may change this sentence to:

“... Therefore, this paradigm had a 2×2 factorial design of dominant (red and green percepts taken together) and mixed perception types for the BR and RPL conditions ... “

Page 5 first paragraph: “... formeda...” => “... formed a ... ”

Page 5 line 11: “greenpercept” -> “green percept”

Page 6, Figure 1 line 7: “...can generate alternating percept...” -> “... can generate alternating percepts”

Page 7: “... over- lapped ... “ => “... overlapped ... “

Page 11, line 11: Should the reference not be of the paper instead of the URL? -> Diego Vidaurre, Andrew J. Quinn, Adam P. Baker, David Dupret, Alvaro Tejero-Cantero and Mark W. Woolrich (2016) Spectrally resolved fast transient brain states in electrophysiological data. NeuroImage. Volume 126, Pages 81–95.

=> At least the reference should also be added!

Page 18, line 7: Same section title as above

Page 18, line 12: “onthe” -> “on the”

Page 22: fMRIactivation => fMRI activation

There are some formatting problems with reference number 36.

SM, Page 6, line 3: reference the mne toolbox should be with the paper not the URL:
Alexandre Gramfort, Martin Luessi, Eric Larson, Denis A. Engemann, Daniel Strohmeier, Christian Brodbeck, Roman Goj, Mainak Jas, Teon Brooks, Lauri Parkkonen, and Matti S. Hämäläinen. MEG and EEG data analysis with MNE-Python. *Frontiers in Neuroscience*, 7(267):1–13, 2013. doi:10.3389/fnins.2013.00267.

SM, Page 10, line 4: reference should be paper not URL: Pedregosa, F., Varoquaux, G., Gramfort, A., Michel, V., Thirion, B., Grisel, O., Blondel, M., Prettenhofer, P., Weiss, R., Dubourg, V., Vanderplas, J., Passos, A., Cournapeau, D., Brucher, M., Perrot, M., & Duchesnay, E. (2011). Scikit-learn: Machine Learning in Python. *Journal of Machine Learning Research*, 12, 2825–2830.

SM, Page 11, line 6: the URL of the Matlab code sends me to the matlab homepage

SM, Page 12, line 4: URL doesn't work because there is a space in it

Reference list:

- Abdallah D, Brooks JL. 2020. Response dependence of reversal-related ERP components in perception of ambiguous figures. *Psychophysiology*. 57.
- Blake R. 2001. A Primer on Binocular Rivalry, Including Current Controversies. *Brain Mind*. 2:5–38.
- Blake R, Logothetis NK. 2002. Visual competition. *Nat Rev Neurosci*. 3:13–21.
- Brascamp J, Sterzer P, Blake R, Knapen T. 2018. Multistable Perception and the Role of the Frontoparietal Cortex in Perceptual Inference. *Annu Rev Psychol*. 69:77–103.
- Britz J, Landis T, Michel CM. 2009. Right parietal brain activity precedes perceptual alternation of bistable stimuli. *Cereb Cortex*. 19:55–65.
- Ehm W, Bach M, Kornmeier J. 2011. Ambiguous figures and binding: EEG frequency modulations during multistable perception. *Psychophysiology*. 48:547–558.
- Intaite M, Koivisto M, Ruksenas O, Revonsuo A. 2010. Reversal negativity and bistable stimuli: Attention, awareness, or something else? *Brain Cogn*. 74:24–34.
- Kornmeier J, Bach M. 2012. Ambiguous figures – what happens in the brain when perception changes but not the stimulus. *Front Hum Neurosci*. 6:1–23.
- Kornmeier J, Friedel Evelyn, Hecker L, Schmidt S, Wittmann M. 2019. What happens in the brain of meditators when perception changes but not the stimulus? *PLOS ONE*. 14:e0223843.
- O'Donnell BF, Hendler T, Squires NK. 1988. Visual evoked potentials to illusory reversals of the Necker cube. *Psychophysiology*. 25:137–143.
- O'Shea RP, Kornmeier J, Roeber U. 2013. Predicting visual consciousness electrophysiologically from intermittent binocular rivalry. *PLoS ONE*. 8:e76134.
- Pitts MA, Britz J. 2011. Insights from intermittent binocular rivalry and EEG. *Front Hum Neurosci*. 5:107.
- Pitts MA, Nerger JL, Davis TJR. 2007. Electrophysiological correlates of perceptual reversals for three different types of multistable images. *J Vis*. 7:1–14.
- Sandberg K, Barnes GR, Bahrami B, Kanai R, Overgaard M, Rees G. 2014. Distinct MEG correlates of conscious experience, perceptual reversals and stabilization during binocular rivalry. *NeuroImage*. 100:161–175.

Reviewer #3 (Remarks to the Author):

This study re-analyzed a published EEG-fMRI dataset on binocular rivalry. Based on EEG-informed analysis, the previous study revealed suppressed BOLD activities in posterior region of the default model network (DMN) associated with perceptual transitions. The current study used more sophisticated analysis approaches on the EEG data and suggest that the precuneus (PCU), a key node of DMN, plays a causal role for perceptual switch. There are quite some analysis and results, and rather complicated. I brief them as follows:

They first found reduced global field power (GFP) in BR vs. RPL condition at about 300ms before the button report of dominant percept. Source localization based on the prior of fMRI activation revealed deactivation of a range of brain areas including some of the DMN nodes. However, the fMRI results didn't reveal any deactivations. Then they used a four states Hidden Markov Modeling (HMM) algorithm to fit the time course of EEG sources from DMN regions and V1. The occupancy of state 4 was found much longer for the rivalry than the replay condition. Although there were also obvious differences in other states, they used state 4 for the following analysis. Then a spectrum analysis was performed on power and coherence of different frequency bands between different regions from different states. Only alpha band result was used the following analysis. For the alpha coherence, state 4 shows significant PCU-V1 phase coherence. The PCU-rIPL coherence was also significant, but it was not mentioned in the text. Then they selected the PCU-V1 coherence for granger causality analysis and found no correlation of GC->V1 with perceptual duration at the trial-by-trial level. But there was a correlation at the group level only when the trial was dominated by state 4. Then they came to the conclusion that the causal effect from precuneus (PCU) to V1 can predict the following duration of perceptual state.

These are some interesting results from these analyses, but the main conclusion comes from a long way with many steps of arbitrary selections. Thus I feel the evidence for the final conclusion are relatively weak.

Another major concern is for the definition of time of perceptual switch. The current study used the button press of perceptual report for the time of switch, while the human reaction time on average is about 400ms. The deactivation they found on the EEG signal (about 300ms before button press) may occur after the perceptual switch. Thus the DMN activity they found associated with this time period should be a consequence rather than the cause of conscious perception. I suggest the author to use the frequency tagging from the EEG signals to decode the "real" perceptual switch from the visual cortex.

Reviewer #4 (Remarks to the Author):

In their study the authors aim at analyzing the impact of intrinsic brain activity on behavior by using a paradigm of binocular rivalry. The topic itself is of high interest. The authors assemble a set of literature which underlines the relevance of their question and use advanced methods in the analysis pipeline.

My fundamental concern is that the time window in focus does not capture intrinsic brain activity before the percept, but during perception and therefore cannot be labelled predictive. While the results may still be interesting they cannot be interpreted in the way presented in the manuscript. In detail, the authors used the time point of button press response as the time of perception and neglected the reaction times, which typically is a few 100 ms depending on the instruction to the subjects. Most results rely on a time window of 1 sec before the button press response and therefore most likely contain a mixture of pre-percept signal and the sensation of perception. During the 1 sec period before the button press subjects experienced very different conditions, the (actual) change to a dominant color (RPL dominant), the change to a color, which they knew would not last (RPL mixed), the emergence of an ocular dominance (BR dominant) or a state of perceptive uncertainty (BR mixed). Therefore the robust splitting into 4 states seems plausible. The interpretation here needs to be refocused. The button press being a few hundred milliseconds after the precept change is also

inline with the strong difference in global field power 400 ms to 300 ms before button press (Fig 3a).
Suggested additions:

- One could give an estimate of the subjects' reaction times from conditions 'RPL dominant' and 'RPL mixed'.
- It would be interesting to analyze the transitions through states 1 to 4 across the time interval. Maybe there is a higher prevalence of e.g. state 4 in BR dominant (or mixed) at a certain point in time. If this time point would precede the putative time of percept switch (as estimated from the (individual) reaction times) this would be an interesting information.
- There should be a way of quantifying direction of phase difference or causality from the information on the different states themselves. Maybe this is depicted in sFig. 13, and would be of interest, e.g. to clarify if GC (PCU -> V1) is a hallmark of state 4.

Minor:

- The overall description of the succession of analyses is quite complicated. It would probably benefit from focusing on the EEG analysis strategies right away. The importance of the concurrent fMRI is not obvious. The overlap of EEG and fMRI localizations stated on p.10 is not obvious to me, especially in the precentral gyrus. The fMRI analysis has its relevance in the EEG source localization, but maybe it would be enough to shift it to the supplementary information. With regards to the source localization in the EEG it never became clear to me which strategy was favored in the end and if fMRI was essential here. Please clarify.
- Fig. 3c: The error bars are asymmetric. Which measures are shown ?
- Fig. 3d: Why is the contrast shown for exactly this time window ?
- p. 9: I did not find finer grained results on frequency (and time windows).
- P. 11: Which figure or table shows the source level EEG activation contrast including the stated regions, e.g. PCC? (caption in Fig. 4 says 'sensor level')
- P. 11: I don't understand the phrase '(with 0.1-s post-response padding)' in lower page. Please elaborate.
- The caption of Fig. 4 contains methods details which should probably be shifted to the supplements.
- In the state life times possible ceiling effects should be addressed.
- Might GC (PCU -> V1) correlate with the presence of state 4 ?
- SI: In the caption of sFig. 2 please hint at table s1. Which model was finally chosen and why ?
- SI p. 11: I wonder if the Viterbi path can be properly estimated if subject data as well as event segments are concatenated.
- What does sFig. 13 mean? Does it represent the alpha-band? What is the meaning of the 95%-CI? Why are some regions depicted twice like PCU in sFig. 13b left side?

Typos:

- SI: p.5 before 'EEG data processing': remove 'been'
- P. 11: 'We set the HMM algorithm to extract 4 states from the data, which are the concatenated regional signals of all trials from all conditions and subjects (Fig. 4).' 'Which' is not correct.
- First paragraph on p. 14 should be to Figure 5b instead of 4b
- P.17 after Fig. 5 wrong title insert ?
- SI p. 10, first line after title: 'used _in_ this study'
- SI p. 12: 'resulted state' is not a proper formulation

22nd of March 2022

Dear Reviewers,

We would like to thank the Editor for their consideration of our manuscript for publication in Nature Communications, and all of the Reviewers for their positive, insightful and constructive feedback for our submission. In this letter and the revised version of the manuscript, we have aimed to address all comments raised by the Reviewers.

There are two main concerns (methodological complexity and temporal precision) expressed commonly by the reviewers. Regarding the methodological complexity, we have (1) provided more methodological details as requested, and shifted some details from the manuscript to the supplementary materials, (2) provided detailed explanations to the mathematical tools introduced in the manuscript and the analysis pipeline in the supplementary materials, (3) added an overview of the key methods involved in this study before going through the results section, (4) reorganised the structure of the results sections, with section titles more clearly reflecting the purpose of the corresponding analysis. Regarding the temporal imprecision of using the motor response to mark a trial, we have (1) justified and discussed possible limitations of the response-locked analysis that we conducted, (2) provided evidence for the notion that the temporal precision of the response-locked analysis does not influence the result of the subsequent Hidden Markov Modelling (HMM), (3) refocused the narratives in the manuscript to the HMM results, including strengthening the arguments that the HMM states reflect the endogenous neural activities.

We have also addressed other issues raised by the reviewers point-by-point, including resolving shortcuts, adding more references, providing more results beyond what has already been reported etc.. To best accommodate the reviewers' comments and to adhere to the journal requirements, we also reduced the length of the paper, to a state where no further reduction can be done without losing key points.

In our detailed responses below, we address each of the reviewers' comments in turn, with reviewer comments in black text (led by bullet points), our replies in blue text (following each comment), and with the referred revised text from the article inside black text boxes. All changes in the article are highlighted in yellow.

We are grateful to have been given the opportunity to revise our manuscript with the help and the input by the four reviewers and look forward to hearing from the Editor.

Reviewer #1 (Remarks to the Author):

Summary:

This study aims to explore the relationship between (conscious) perception/visual awareness and the intrinsic dynamics of large-scale networks (esp. the DMN). More specifically, the objective is to determine whether such intrinsic dynamics serve a role in perception/awareness. To do this, the authors utilize data from an EEG-fMRI experiment that had two main conditions: 1) binocular rivalry (BR) and 2) perceptually matched replay (RPL). First, fMRI contrasts resulted in the observed activation within cortical regions that are considered to be part of the DMN. These results constrained, or were used to guide, subsequent EEG analysis. EEG results indicated a particular time period where endogenous vs. exogenous information could be distinguished (-400 - -200 ms), as well as the cortical sources associated with this distinction (including several regions of the DMN). Within these regions, HMM (with 4 states) was used to probe intrinsic dynamics (e.g., switching rate and fractional occupancy), whereby state 4 was found to be associated with the BR (dominant) condition and suggested to be important for the prediction/production of perceptual fluctuations. Further spectral considerations of the states (with emphasis here on the alpha band) indicated a precuneus-V1 phase coherence for state 4 which was also associated with faster perceptual transitions. The authors conclude that these results point to the idea that intrinsic dynamics may not be epiphenomenal, and rather predictors of perception (and its fluctuations), as shown within the DMN.

General Comments:

Overall, given the missing information about the endogenous features/sources of perception, this study is important in helping to address that very point. This paper is manifold in its objectives, which include: 1) investigating intrinsic dynamics and their relationship to perception, and 2) establishing a role for the DMN (specific cortical regions) in endogenously-driven perception/awareness. The findings suggest that the DMN might be involved in task-related activity, as is shown by fMRI activation for endogenously- vs. exogenously-driven perception. Moreover, they help to identify a time period that is important for such a distinction, as well as the implicated neural correlates. These considerations will help to address gaps in the literature related to (conscious) perception/awareness.

'Major' Comments:

These may be some points to address:

There are methodological details that, although were pertinent to the previously published articles, are also important here:

1. The results of the power analysis that led to 20 participants were not included or described (even if previously published) (Is there sufficient power for this study?)

We appreciate and share the reviewer's concern for an adequately powered study and we considered this issue before we started working with this dataset. We considered the study to be adequately powered before we decided to use this dataset. This study adopted a within-subject experimental design. For each participant, each experimental session/condition of rivalry or replay consisted of 5 consecutive 42-second blocks of continuous stimulus presentation followed by 12 seconds of rest, and 5 total blocks for each stimulus type. Each participant repeated the binocular rivalry and replay sessions 5-7 times to increase replicability, which generated about 240 scanning sessions and 25767 trials in total for analysis (about 6 scanning sessions and 327 trials on average for each participant in each experimental condition). We consider this study is adequately powered for a within-subject analysis, by which our standard neuroimaging processing (i.e., fMRI activation and EEG evoked-response analyses) was conducted.

We think that the statistical bottleneck in this study mainly exists in the analyses that sought to establish a significant relationship at the group level. In these situations, within-subject variability was disregarded, and only an averaged value across trials was used to represent a subject, causing a drastic shrink in the sample size. Therefore, we selected the group-level tests (model 1-4 as below) to conduct post-hoc power calculations, which will serve as a lower-bound of power for all the tests in this study. Please see the following statistical model descriptions (and their placement in the main text), effect size measures, degrees of freedom and the post-hoc power estimations, calculated with the software G*power (<https://stats.idre.ucla.edu/other/gpower/>) which we used to formally conduct the post-hoc power analysis. In the revised manuscript, we reported Cohen's F^2 , in addition to other effect-size indicators that we have already reported (i.e., the F or T score).

- 1) Model description (Page 12, Line 268): The interaction effect between states and conditions to the state FO; FOs were averaged cross trials within each state, condition and subject.
($F_{9,304} = 22.86$, Cohen's F squared = 0.68, power = 1.00)
- 2) Model description (Page 15, Line 351): The linear regression of the K4 lifetime to perceptual duration; K4 lifetime and perceptual duration were averaged across trials within each subject respectively for the BR (dominant) and BR (mixed) conditions.
($t = -2.28$, $df = 37$, Cohen's F squared = 0.12, Power = 0.70)
- 3) Model description (Page 16, Line 370): the interaction effect between K4 lifetime and the PCU->V1 granger causality against the perceptual duration; K4 lifetime and perceptual duration were averaged across trials within each subject and K4 lifetime category for the BR (dominant) condition.
($t = -3.01$, $df = 76$, Cohen's F squared = 0.13, power = 0.89)

- 4) Model description (Page 16, Line 373): linear regression between the PCU->V1 granger causality and perceptual duration (when K4 lifetime > 0.95s); K4 lifetime and perceptual durations were averages across trials within each subject for the K4 lifetime category of “State 4 > 0.95s” in the BR (dominant) condition.
($t = -3.00$, $df = 11$, Cohen’s F squared = 0.41, power = 0.70)

Because post-hoc power analysis is usually discouraged, we did not include the power value in the manuscript. However, these statistical tests strongly indicate a reasonably well-powered study, with a power ranging between [0.7 1]. In fact, our statistical inference and final conclusion was drawn from the interaction model (model 3) which has a power of 0.89.

2. Presumably, gaze was tracked throughout the entirety of the experiment, but this was not mentioned or described

We agree with the reviewer that the eye tracking data would have provided an extra way of validating our results, but unfortunately, gaze was not tracked in the original study. Although lacking eye tracking data, this multimodal dataset is valuable enough, since simultaneous recording of the EEG-fMRI is still relatively scarce.

3. The number of predictors in the creation of GLMs for fMRI analysis was not clear, nor what the predictors were used to model

We thank the reviewer for their request of a clarification of the modelling details; The revised text is provided below:

SI, Page 5, Line 122-131:

To fit the fMRI timeseries of each scanning session, the modelled event-related BOLD responses associated with the main effects (dominant green percept, dominant red percept and mixed percept) were taken as the main regressors, along with the non-neuronal confounds (WM, CSF and 6 movement parameters) and the block effects (the 5 task blocks). To increase reliability, each participant was scanned in multiple sessions, with about half (~6) of the sessions under the BR condition and another half under the RPL. Therefore, the first-level GLM for each participant was modelled with a mixed-level factorial design, with session/condition and perceptual types as different factors. The global session effects were also specified as covariates.

4. Several contrasts were conducted, though thresholds and multiple comparison details should be further emphasized in the corresponding sections

We thank the reviewer for this advice. We have added the methodological details for addressing the multiple comparison problem in our manuscript as below:

SI, Page 6, Line 141-150:

For the group-level GLM, a random effect (RFX) design was used, with random effect being the intercept of within-subject GLM fitting, and fixed effect being the main effects/contrasts of interest. Specifically, weighted beta coefficient maps from individuals were fed into a one-sample t-test to make group-level inferences. To address the multiple comparison problem, cluster-level inferences were used and corrected for the family-wise error rate within the framework of Gaussian random field implemented in SPM (version 12) (Siegmund & Worsley, 1995). Clusters were defined by a default voxel-level threshold of 0.001 (uncorrected). The family-wise error rate was controlled at the cluster level, and a threshold of P-corrected < .05 was used to determine significance among clusters.

Since the reviewer's concern of the multiple comparison problem arose from the multiple contrasts we adopted, we wondered if the reviewer is suggesting that we should adjust for the beta contrasts in the fMRI activation study. If that is the case, we would like to state that in the neuroimaging studies, it is not a standard practice to correct for the number of contrasts. Since we already corrected the multiple comparisons for the independent tests done for all the voxels with a standard procedure, it is not appropriate to arbitrarily impose a more stringent P-value (Althouse, 2016; Benjamini, 2010).

A contrast in this context is different than post-hoc contrasts adopted in the case of ANOVA, which is mainly where the multiple-comparison problem arises (Lee & Lee, 2018). The contrasts in our context were essentially (weighted) beta coefficients associated with the regressors, which were estimated in one go from the same GLM. When considered in the group-level, different beta-coefficient maps were used for constructing different group-level GLMs, with different statistical hypotheses. Since these group-level GLMs do not share a same null distribution or a same alpha rate, (hence having no alpha rate inflation), there is no need to correct the alpha rate in this situation.

5. The analyses that were conducted at the individual vs group level should be more clearly indicated / separated in the text.

We apologise for not being clear about these methodological details. In this newer version of our manuscript we have clarified the levels of analyses and have separated the sections describing the models for the two levels. These changes were made to address the previous two questions, so please also see the revised text above.

6. One methodological consideration revolves around ROI analysis and why the authors did not first identify key DMN regions and extract data from these regions / use only these regions to guide their analysis (given the importance of the DMN here)

We thank the reviewer for their reflection on the motivation of our data analyses, and we agree with the reviewer that it would have been clearer and simpler to just choose the DMN nodes in the beginning for guiding our analysis pipeline. However, there are both theoretical and practical considerations behind the present analysis scheme. Firstly, as we stated in the introduction the DMN is typically known as a “resting-state” network, so we felt obligated to show that the DMN is actually involved in this task, before we went on to investigate the DMN ROIs’ dynamic properties during the task. Hence, a verification of this point constitutes the first part of the study. Secondly, the DMN consists of large brain areas with extensive heterogeneity, and we still lack enough prior knowledge to pinpoint which subregions within the DMN we should be looking into if we were to follow an a-priori anatomical region definition. We think by constraining our ROIs within the significant regions obtained from the activation analyses we increased the sensitivity of subsequent analyses. Therefore, by using an fMRI-prior guided EEG evoked-response analyses, we identified the significant regions within the DMN, and we extracted the timeseries from these regions for the subsequent dynamic-state modelling.

7. A brief Methods section included in the main text would be useful in guiding the reader, in addition to have a full-length version of the Methods in the supplementary section

We thank the reviewer for this great suggestion. We have added a paragraph about the method roadmap before introducing the neuroimaging results (please see below):

Manuscript, Page 6, Line 131-142:

Multimodal neuroimaging analyses

Different than the goal of the original study using this dataset, which was to reveal the fMRI activation associated with BR-induced perceptual transitions³⁷, our goal was to test the hypothesis that the endogenous neural dynamics in high-level cortical areas (i.e. DMN) can influence low-level (i.e. primary visual cortex) information processing. Given the controversies regarding the DMN’s role during tasks^{32,34,38}, we first confirmed the involvement of DMN regions in this task using ANOVA F tests with the fMRI dataset. Then using the EEG dataset, we conducted evoked response analyses for two reasons: (1) to narrow down the critical time window preceding a perceptual transition, in order to facilitate further modelling of the dynamic neural process; (2) to validate that the DMN’s involvement is also reflected on the EEG data. Finally, within the specified time window and using DMN source signals, we adopted a time- delay embedded Hidden Markov Model to look for the transient spatiotemporal patterns that may represent endogenous neural triggers of the upcoming, involuntary perceptual

changes during BR. The analysis pipeline is presented in the supplementary information (SI; sFig. 1).

There are also some theoretical points to address:

1. Although the fMRI analysis suggests that DMN regions were active in distinguishing between endogenously- vs. exogenously-driven perception, additional analysis might be considered to determine the direction of the effect in order to determine the effect that is driving the activation (for example, is it the increased endogenous-related or suppressed exogenous-related activity that is driving the activation in those regions?)

We thank the reviewers for pointing out this. In the revised manuscript, we present t-test (one-tailed) results in addition to the original F-test (two-tailed) result, which we hope addresses the reviewer's question. Please see the updated Fig. 2 in the main text (page 8):

Manuscript, Page 7, Line 158-177:

Fig. 2. FMRI activation revealing the DMN's involvement in the current task. Subplots (a), (b) and (c) show the significant clusters [$P_{\text{voxel}} < 0.001$ (uncorrected) & $P_{\text{cluster}} < 0.05$ (family-wise error corrected)] respectively for the contrast “BR (dominant) - RPL (dominant)”, “BR (mixed) - RPL (mixed)” and the interaction effect between perceptual generating (BR vs. RPL) and perceptual (dominant vs. mixed) conditions. The interaction analysis (c) shows the regions mostly sensitive to the condition differences. For statistical testing, we adopted linear mixed-effects models, where the paired T-tests and F-test were carried out at the individual level, while statistical inferences were made at the population level with group-level one-sample T tests (input being the individual-level estimators). Hence, the colour bars of the subplots (a)-(c) indicate the T scores from the group-level testing. The circular plot in (c) shows the ICN affiliation of the significant clusters for the interaction effect. “ICN involvement” is a measure of correspondence between an activation map and large-scale networks with well-established cognitive function, as provided by the BrainMap (BM) meta-analysis database³⁹. The BM number on the circular plot indicates the number of the ICN-BM network atlas. Cognitive domains of the networks that have minute involvement in this task are not listed in the figure. These are BM1: Limbic and medial-temporal areas; BM3: Bilateral BG and thalamus; BM4: Bilateral anterior insula/frontal opercula and anterior cingulate gyrus; BM5: Midbrain; Cerebellum; BM6: Superior and middle frontal gyri; Sensorimotor; BM9: Superior

parietal lobule; Frontoparietal (perception-somesthesis-pain); BM10: Middle and inferior temporal gyri; Frontoparietal (cognition-language); BM11: Lateral posterior occipital cortex; BM14: Cerebellum; BM16: Transverse temporal gyri; BM17: Dorsal precentral gyri, central sulci, postcentral gyri, superior and inferior cerebellum; BM19: Artefactual component; BM20: Artefactual component.

We would like to comment that this question may not be satisfactorily answered until we completely understand what is the origin of deactivation per se. We will elaborate on this point below:

As the reviewer pointed out, the activation of the DMN for the contrast BR > RPL during the dominant phase might be caused by suppressed exogenous-related activity, and this is exactly what we saw from the BOLD-response main effect associated with the dominant perceptual report (see sFig. 3. “fMRI event-related”). This is expected because the DMN “deactivation” associated with a visual onset in a normal viewing condition is a robust finding in the literature (Raichle et al., 2001; Raichle & Snyder, 2007). However, the cognitive role and/or the physiological cause of the DMN deactivation is still largely unknown (Callard & Margulies, 2014; Gu et al., 2019; Pashaie, 2021; Spreng, 2012).

In the EEG evoked response analysis, we could see that during the 200-400 ms interval before a dominant perceptual transition, the DMN nodes (e.g. ACC and PCC) stood out as the most activated regions (with the top 10% strongest activation) in the BR condition, but it was not the case for the RPL condition (see revised sFigure 3b). In fact, the field potential measure used in the EEG largely reflects the variance or energy of the ERP waves which is irrelevant to the actual signs of the EEG waves from recording. Although the aforementioned ACC and PCC seemed to be more active during BR than RPL in the EEG source reconstruction, this difference did not survive the significance test for the BR > RPL contrast; and this may be due to the fact that there is a decrease of the global field potential for the BR vs. RPL condition (Fig. 3a). In addition, the source reconstruction of the [-400 -200] ms is just one snapshot from a dynamic process, while the slow response of the BOLD might reflect an averaged effect across and beyond this time window. It is also noteworthy that although multi-modal evidence is promising, the correspondence between the EEG measures and the fMRI BOLD activation is still controversial (Portnova et al., 2018). Hence, we only drew a safe conclusion in the text that the DMN is somehow involved in the process, regardless of the direction of the activation (by using the interaction analysis).

We did not report all the activation results in the main text because they reveal an overly complex picture which did not help with our main story (however, we presented the baseline activation results in the revised Supplementary materials, as sFig. 3). As we have stated before, it is still not completely understood what deactivation means for fMRI, and the signs for the EEG recording are arbitrary, so we feel that the reviewer’s question concerns some broader neuroscientific questions than what our study can address. We should emphasise however, that it is not the focus of the current study to discover activation or deactivation of regions; instead, we considered neural events as dynamic processes with continuous

fluctuations (i.e., transient activations and deactivations in the neural activity). We believe this approach is probably more important than static analyses for revealing some new insights into fast neural encoding which is in our case.

SI, Page 18, Line 416-422:

sFig. 3. (a) The fMRI event-related activation (without contrasts) for the dominant percept in both BR and RPL condition. (b) The EEG source-level evoked activation (without contrasts) during [-200 -400] ms before response to a dominant percept in both BR and RPL conditions. The brain map shows the brain regions with the top 10% of the T-scores.

2. Given the involvement of the DMN and the link found between PCU and V1, it would be good to explore how the other DMN regions affect (if at all) the PCU > V1 relationship for perceptual transitions (e.g., through dynamic causal modeling?)

We appreciate the reviewer's extended interest in our study and we agree it would be good to see how other DMN regions behave in this context. The reason that we

focused on the PCU is because the PCU was the only DMN region with a functional coupling with V1, as a significant neural feature of State 4. This functional coupling demonstrated opposite patterns between BR (dominant) and RPL (dominant), the key contrast that we are interested in. We have added this argument in the revised text.

Manuscript Page 15 Line 352-356:

We further investigated the cognitive implication of the top-down connectivity from the PCU to V1, since we are interested in how the DMN exerts an influence on the V1, and PCU was the only DMN region shown to have a functional coupling with V1. The functional coupling between the PCU and V1 stood out as a significant neural feature of State 4; and it also demonstrated opposite patterns between BR (dominant) and RPL (dominant), the key contrast that we are interested in.

We should also note that phase coherence measures the consistency of phase difference, while granger causality measures lagged (auto-)correlation, which is also related to phase difference. Therefore, both measures are concerned with the temporal structure of oscillatory waves. However, DCM is based on a totally different methodology. It is a generative model which relies on local neural oscillator modelling and Bayesian model comparisons, even though its causal inference is still based on Granger causality (GC). The main reason we chose GC over DCM is for its methodological comparability with the previous phase coherence results from the HMM section.

3. Examining cortical modulations stemming (esp. temporally) from V1 in the endogenous vs. exogenous conditions would help to provide further clarification or insight into how the results lead to a 'diversion of the visual processing pathway', as well as how this leads to visual awareness.

We thank the reviewer for this point of view. However, information propagation from V1 to higher cortex is a perceptual process (Mashour et al., 2020), rather than a pre-perceptual process that we focused on here. Our follow-up investigation of the Granger-causality (GC) from V1 to PCU suggests that (1) GC (V1→PCU) is significantly smaller than GC (PCU→V1) during BR (dominant) trials ($t = 2.93$, $p = 0.00$); and (2) it does not have the behavioural correlation ($t = -1.55$, $p = 0.15$), as opposed to what we found for the direction of PCU→V1 (see Fig. b [$0.95 < \text{State 4}$]). These suggest that the causal effect from V1 to PCU is not key to the involuntary percept switches during BR. All of the data and codes required for generating these results have been uploaded to the online repository.

We would like to comment that thanks to this experiment paradigm, DMN regions' activity can be interpreted as driving the perceptual changes even without a causal modelling. This is because that the external stimuli did not change for the BR condition, then the endogenous brain activity is the only source of variance that the perceptual fluctuations are contingent on. The signal (of an altered image) stemming from V1 will propagate to higher aspects of cortical hierarchies, which is a perception

process supposedly indifferent between the BR and RPL conditions and is out of the scope of our study.

4. Although the DMN is the large-scale network of focus here, a short mention of whether it is the only source of perturbations to result in perceptual fluctuations and/or how the other large-scale networks mentioned might be involved in producing visual awareness.

We thank the reviewer for pointing out this important aspect. In the revised text, we have added relevant comments to that, please see the highlighted changes in the manuscript (as below). We would like to point out this is probably an issue of terminology rather than absence. Earlier studies discussed regions rather than networks, and since with the fMRI activation study, we replicated the earlier findings, for comparison purpose, we used region naming rather than network terminology.

Manuscript, Page17, Line 406-413:

These high-level regions have been localised to frontoparietal regions by previous functional and structural MRI studies, including the frontal eye field, superior/middle/inferior frontal gyrus and superior/inferior parietal lobule (SPL/IPL)⁵⁰⁻⁵². Especially the right-lateralised SPL was suggested to play a causal role in the BR transition by a Transcranial magnetic stimulation (TMS) study⁵⁰. SPL is not investigated in our study, but parts of our results hinted at its relevance. Our critical state, State 4, was characterised not only by an increased coupling between PCU and V1, but also that between PCU and rIPL. Although the IPL is usually considered to be part of the DMN, it is anatomically adjacent and functionally coupled to the SPL which is a core of the posterior attentional control system^{53,54}.

Minor Comments:

1. Abstract: "causal effect" could be further clarified in terms of Granger causality

We thank the reviewer's suggestion. We have changed it accordingly (see below):

Manuscript Page 1, Line 26:

The interaction between the lifetime of this state and the PCU->V1 Granger-causal effect is correlated with the rate of perceptual fluctuation.

2. FMRI activation study (this subheading might suggest that an entirely separate study was conducted): "over- lapped" should be changed to 'overlapped'

We thank the reviewer for spotting this typo. We are sorry for this occasional failure of text rendering upon a PDF conversion. We have changed it (along with several others) in the revision.

3. Fig. 2/legend: Acronyms should be defined in full when first mentioned (e.g., ICN, BM, FCPN)

We are sorry for our omission. In the revision acronyms were checked throughout the paper and supplementary-materials document, and full names were provided for them.

4. Hidden neural states within DMN ... subsection: the HMM acronym should be defined (this acronym is only defined in the supplementary section)

Please see comment above.

5. Fig. 5 legend: “pair-wise” should be ‘pairwise’, “(clock-wise)” should be ‘(clockwise)’

We thank the reviewer for this correction. We have changed accordingly.

6. Fig. 5 legend: font size of cortical region names on the circular bundle plots is quite small

We thank the reviewer for this feedback. We have changed accordingly, please see the updated plot as below (Manuscript page 14):

7. PCU-V1 phase ... subsection: “Therefore, we next focused into..” – the ‘into’ should be changed to ‘on’ or similar

As above.

8. There are two “PCU-V1 phase coherence and Granger Causality predicts stable percepts” subheadings

We thank the reviewer for spotting this mistake. We have reorganised our result subheadings and fixed this problem. The reorganised subheadings are presented below:

- Results
 - Behavioural analyses
 - Multimodal neuroimaging analyses
 - fMRI activation analysis
 - EEG evoked response analysis
 - Dynamic neural pattern analyses
 - Endogenous neural states underlie upcoming perceptual transitions
 - State spectral information reveals PCU-V1 phase coherence
 - Behavioural relevance and cognitive implications of the critical state

9. The acronym for Granger Causality should be provided (GC)

We thank the reviewer for spotting this and we have fixed it.

10. Mentions of results found in supplementary section should be expanded into statements about what those results show, in addition to where in the supplementary section they can be found

We thank the reviewer for this feedback. We have fixed it in the revision.

11. (“All statistical models have been presented..”) – the first letter of the first word in these brackets should be lowercase

We have fixed this.

12. Discussion, paragraph 1: “namely, the interaction of them” should be “namely, this interaction”

As above.

13. There is considerable focus in the discussion on the role of SPL, despite the results focusing on PCU/V1

We thank the reviewer for this valuable feedback. SPL needs its share of discussion because a considerable amount of BR literature (especially fMRI studies) has emphasised the relevance of this region. But we agree with the reviewer that its proportion should not exceed the discussion of the PCU. In the revised text, we reduced this part (Page 16) and enlarged the part discussing the PCU.

Manuscript, Page 18-19, Line 450-463:

Although the PCU has been associated with visuospatial integration and imagery⁶³, and its role in BR transition could be seen as a sign of voluntary cognitive processes, either attentional control or memory retrieval^{64,65}, we rather propose that the PCU-V1 coupling might be an underlying pre-conscious process, an indication of how intrinsic dynamics exert influence onto the information encoding in the primary visual cortex. Previous studies have suggested the PCU's role in consciousness. For example, recent literature using graph theoretic metrics and information theory has provided a new perspective for how we understand consciousness; it suggests that the posteromedial cortex (PMC; PCC/PCU), standing out as a hub for the whole-brain information exchange, affords the complexity that consciousness emerges from⁶⁶⁻⁶⁸. The recent ground-breaking single-case report, based on the subjective feelings of self-dissociation accompanied with the seizure onset originating in the PMC (around BA 31) and the fact that these feelings could be reliably replicated upon direct electrical stimulation in the PMC, convincingly showcased the PCU's relevance to consciousness⁶⁹. Meanwhile, there have been theoretical arguments which hinted on the PCU's role in information integration between the internal and external worlds^{70,71}, a fundamental element to conscious processing^{72,73}.

14. "on the group level" should be "at the group level"

We have fixed it.

15. "-400-200 ms" suggests an overall 600-ms duration (if '-400 - -200 ms' is what was meant, this should be indicated where appropriate in the text)

This has now been corrected.

16. Supplementary: ">3mm displacement" might be amended to "translations and/or rotations of > 3mm/°, respectively", given that both factors (indicated in mm or °) are accounted for in motion correction

We thank the reviewer for pointing it out. The rotation angular differences can be converted to distance differences (by projecting to a sphere). The displacement used in ArtRepair is the maximum voxel displacement resulting from the combined effect of individual translation and rotation (Mazaika et al., 2009). We have added the information in the SI (see changed text as below):

SI, Page 4 Line 97-100:

Fast motions were inspected for each scan of every subject. Volumes with > 3mm displacement (i.e., the voxel displacement resulting from the combined effect of individual translations and rotations) compared to the previous volume were considered to be contaminated by fast motion and were deweighted in the GLM design using the ArtRepair toolbox (Mazaika et al., 2009).

17. Supplementary: "on Python" could be "(implemented) in Python"

We have fixed it.

18. Supplementary: here it is "PCu", whereas it is "PCU" in the main text

As above.

19. Supplementary, sFig. 3: no regions are labeled on the brains, nor are they

mentioned in the legend (and scale label font size is quite small – same for sFigs. 4, 10, 11, 12, 13, 14)

We showed the number of timepoints on purpose. The regional information is given in the figure caption. The labelling of data-points in the raw auto-covariance pattern gives information of the size of the micro sliding-windows that we used for capturing the states.

We have enlarged the font size of x/y-axis tick labels and colour bar scales in main Fig. 5, sFigs 3, 4, 10, 11, 12, 13, 14 (the original sFig 3 was shifted to the main text; sFigs. 10-14 from the original version correspond to the sFigs. 12-15 in the revision).

20. Supplementary, Table 2: in the main text, this table is actually referred to as Table 1 when the coordinates are discussed; there is no mention of why there are different numbers of coordinates for each of the regions

We are grateful that the reviewer spotted this mistake, and we have changed it accordingly. There are different numbers of coordinates because they are based on the significant peak voxels identified from the source-level evoked activation analysis.

We added a paragraph in the method to explain how these coordinates were identified, please see the change as below:

SI page 10-11, line 245-264

Given our research interest in the default mode network (DMN) we constrained the subsequent modelling within the 8 regions: bilateral parahippocampal gyri (HP), bilateral inferior parietal lobules (IPL), ACC, PCC, PCU, and the primary visual cortex (V1). To ensure experimental sensitivity, we extracted signals from the significant peak voxels (confined within the DMN) in the previous source-level evoked-response analysis. The contrasts used for determining the regional involvement in the task were BR (dominant) vs. RPL (dominant), BR (mixed) vs. RPL (mixed) and the interaction effects between the two factors; both contrast directions were considered. Hence, 1-second (+0.1s post-response padding) source signals from these coordinates were extracted for all trials (trials being the 1-s epoch before a subjective report of a percept change). When multiple coordinates were identified within a same ROI (identified by automated anatomical labelling), the average of their signals was used as the representative signal of that ROI. The anatomical labels for the peak coordinates were identified using the Talairach Atlas (<http://www.talairach.org/>), upon a conversion to the Talairach space. The list of the

peak voxel coordinates used for this purpose are presented in the Supplementary material Table 2. The full results of all significant voxels/clusters upon all contrasts are available from an online repository:

[https://htmlpreview.github.io/?https://github.com/Aubrey-Lyu/BR-](https://htmlpreview.github.io/?https://github.com/Aubrey-Lyu/BR-project/blob/master/Analysis-)
[project/blob/master/Analysis-](https://htmlpreview.github.io/?https://github.com/Aubrey-Lyu/BR-project/blob/master/Analysis-)

[3 EEG/results/evokeResponse result table permutationtest.html](https://htmlpreview.github.io/?https://github.com/Aubrey-Lyu/BR-project/blob/master/Analysis-3_EEG/results/evokeResponse_result_table_permutationtest.html)

References for Reviewer 1:

- Allen, M., & Friston, K. J. (2016). From cognitivism to autopoiesis: Towards a computational framework for the embodied mind. *Synthese*, 1–24.
<https://doi.org/10.1007/s11229-016-1288-5>
- Althouse, A. D. (2016). Adjust for Multiple Comparisons? It's Not That Simple. *The Annals of Thoracic Surgery*, 101(5), 1644–1645.
<https://doi.org/10.1016/j.athoracsur.2015.11.024>
- Avena-Koenigsberger, A., Misic, B., & Sporns, O. (2018). Communication dynamics in complex brain networks. *Nature Reviews Neuroscience*, 19(1), 17.
<https://doi.org/10.1038/nrn.2017.149>
- Benjamini, Y. (2010). Simultaneous and selective inference: Current successes and future challenges. *Biometrical Journal*, 52(6), 708–721.
<https://doi.org/10.1002/bimj.200900299>
- Callard, F., & Margulies, D. S. (2014). What we talk about when we talk about the default mode network. *Frontiers in Human Neuroscience*, 8.
<https://doi.org/10.3389/fnhum.2014.00619>
- Carmel, D. P., Freeman, E., Lavie, N., & Rees, G. (2004). Working memory maintains perceptual biases during binocular rivalry. *Journal of Vision*, 4(8), 246–246. <https://doi.org/10.1167/4.8.246>
- Carmel, D., Walsh, V., Lavie, N., & Rees, G. (2010). Right parietal TMS shortens dominance durations in binocular rivalry. *Current Biology*, 20(18), R799–R800. <https://doi.org/10.1016/j.cub.2010.07.036>

- Cavanna, A. E., & Trimble, M. R. (2006). The precuneus: A review of its functional anatomy and behavioural correlates. *Brain*, *129*(3), 564–583.
<https://doi.org/10.1093/brain/awl004>
- Friston, K. (2010). The free-energy principle: A unified brain theory? *Nature Reviews Neuroscience*, *11*(2), 127–138. <https://doi.org/10.1038/nrn2787>
- Gu, H., Hu, Y., Chen, X., He, Y., & Yang, Y. (2019). Regional excitation-inhibition balance predicts default-mode network deactivation via functional connectivity. *NeuroImage*, *185*, 388–397.
<https://doi.org/10.1016/j.neuroimage.2018.10.055>
- Herbet, G., Lafargue, G., de Champfleury, N. M., Moritz-Gasser, S., le Bars, E., Bonnetblanc, F., & Duffau, H. (2014). Disrupting posterior cingulate connectivity disconnects consciousness from the external environment. *Neuropsychologia*, *56*, 239–244.
<https://doi.org/10.1016/j.neuropsychologia.2014.01.020>
- Igelström, K. M., & Graziano, M. S. A. (2017). The inferior parietal lobule and temporoparietal junction: A network perspective. *Neuropsychologia*, *105*, 70–83. <https://doi.org/10.1016/j.neuropsychologia.2017.01.001>
- Kanai, R., Bahrami, B., & Rees, G. (2010). Human Parietal Cortex Structure Predicts Individual Differences in Perceptual Rivalry. *Current Biology*, *20*(18), 1626–1630. <https://doi.org/10.1016/j.cub.2010.07.027>
- Kozák, L. R., van Graan, L. A., Chaudhary, U. J., Szabó, Á. G., & Lemieux, L. (2017). ICN_Atlas: Automated description and quantification of functional MRI activation patterns in the framework of intrinsic connectivity networks. *Neuroimage*, *163*, 319–341. <https://doi.org/10.1016/j.neuroimage.2017.09.014>
- Lee, S., & Lee, D. K. (2018). What is the proper way to apply the multiple comparison test? *Korean Journal of Anesthesiology*, *71*(5), 353–360.
<https://doi.org/10.4097/kja.d.18.00242>
- Luppi, A. I., Craig, M. M., Pappas, I., Finioia, P., Williams, G. B., Allanson, J., Pickard, J. D., Owen, A. M., Naci, L., Menon, D. K., & Stamatakis, E. A. (2019). Consciousness-specific dynamic interactions of brain integration and functional diversity. *Nature Communications*, *10*.
<https://doi.org/10.1038/s41467-019-12658-9>

- Lyu, D., Pappas, I., Menon, D. K., & Stamatakis, E. A. (2021). A Precuneal Causal Loop Mediates External and Internal Information Integration in the Human Brain. *Journal of Neuroscience*. <https://doi.org/10.1523/JNEUROSCI.0647-21.2021>
- Mashour, G. A., Roelfsema, P., Changeux, J.-P., & Dehaene, S. (2020). Conscious Processing and the Global Neuronal Workspace Hypothesis. *Neuron*, *105*(5), 776–798. <https://doi.org/10.1016/j.neuron.2020.01.026>
- Mazaika, P., Hoefft, F., Glover, G., & Reiss, A. (2009). Methods and Software for fMRI Analysis of Clinical Subjects. *Neuroimage*, *47*. [https://doi.org/10.1016/S1053-8119\(09\)70238-1](https://doi.org/10.1016/S1053-8119(09)70238-1)
- Parvizi, J., Braga, R. M., Kucyi, A., Veit, M. J., Pinheiro-Chagas, P., Perry, C., Sava-Segal, C., Zeineh, M., Staalduinen, E. K. van, Henderson, J. M., & Markert, M. (2021). Altered sense of self during seizures in the posteromedial cortex. *Proceedings of the National Academy of Sciences*, *118*(29). <https://doi.org/10.1073/pnas.2100522118>
- Pashaie, R. (2021). Potential Effect of Endothelial Signaling on Cerebral Blood Flow Response to Neural Activation. *2021 10th International IEEE/EMBS Conference on Neural Engineering (NER)*, 655–659. <https://doi.org/10.1109/NER49283.2021.9441110>
- Portnova, G. V., Teterova, A., Balaev, V., Atanov, M., Skiteva, L., Ushakov, V., Ivanitsky, A., & Martynova, O. (2018). Correlation of BOLD Signal with Linear and Nonlinear Patterns of EEG in Resting State EEG-Informed fMRI. *Frontiers in Human Neuroscience*, *11*. <https://www.frontiersin.org/article/10.3389/fnhum.2017.00654>
- Raichle, M. E., MacLeod, A. M., Snyder, A. Z., Powers, W. J., Gusnard, D. A., & Shulman, G. L. (2001). A default mode of brain function. *Proceedings of the National Academy of Sciences*, *98*(2), 676–682. <https://doi.org/10.1073/pnas.98.2.676>
- Raichle, M. E., & Snyder, A. Z. (2007). A default mode of brain function: A brief history of an evolving idea. *NeuroImage*, *37*(4), 1083–1090; discussion 1097–1099. <https://doi.org/10.1016/j.neuroimage.2007.02.041>

- Roy, A. V., Jamison, K. W., He, S., Engel, S. A., & He, B. (2017). Deactivation in the posterior mid-cingulate cortex reflects perceptual transitions during binocular rivalry: Evidence from simultaneous EEG-fMRI. *NeuroImage*, *152*, 1–11. <https://doi.org/10.1016/j.neuroimage.2017.02.041>
- Scocchia, L., Valsecchi, M., Gegenfurtner, K. R., & Triesch, J. (2014). Differential effects of visual attention and working memory on binocular rivalry. *Journal of Vision*, *14*(5), 13–13. <https://doi.org/10.1167/14.5.13>
- Seghier, M. L. (2013). The angular gyrus: Multiple functions and multiple subdivisions. *The Neuroscientist: A Review Journal Bringing Neurobiology, Neurology and Psychiatry*, *19*(1), 43–61. <https://doi.org/10.1177/1073858412440596>
- Siegmund, D. O., & Worsley, K. J. (1995). Testing for a signal with unknown location and scale in a stationary Gaussian random field. *The Annals of Statistics*, *23*(2), 608–639.
- Smallwood, J., Bernhardt, B. C., Leech, R., Bzdok, D., Jefferies, E., & Margulies, D. S. (2021). The default mode network in cognition: A topographical perspective. *Nature Reviews Neuroscience*, *22*(8), 503–513. <https://doi.org/10.1038/s41583-021-00474-4>
- Spreng, R. N. (2012). The Fallacy of a “Task-Negative” Network. *Frontiers in Psychology*, *3*. <https://doi.org/10.3389/fpsyg.2012.00145>
- van den Heuvel, M. P., & Sporns, O. (2013). Network hubs in the human brain. *Trends in Cognitive Sciences*, *17*(12), 683–696. <https://doi.org/10.1016/j.tics.2013.09.012>
- Wilcke, J. C., O’Shea, R. P., & Watts, R. (2009). Frontoparietal activity and its structural connectivity in binocular rivalry. *Brain Research*, *1305*, 96–107. <https://doi.org/10.1016/j.brainres.2009.09.080>

Reviewer #2 (Remarks to the Author):

Review of the manuscript: Perturbation or Function? Intrinsic brain dynamics in the Default Mode Network predict involuntary fluctuations of visual awareness

The authors of this manuscript presented binocular rivalry stimuli and non-rivalrous replay variants and used simultaneously recorded fMRI- and EEG data to investigate the causal role of the endogenous dynamics of brain areas from the Default Mode Network (DMN) for spontaneous endogenous perceptual reversals.

They applied several sophisticated data analysis methods (e.g. the time-delay embedded Hidden Markov Model or Granger Causality) in order to isolate four transient brain states with condition specific occurrence frequencies and durations. As main findings the authors report the isolation of one specific transient brain state, mainly based on activity of brain areas from the DMN and a quasi-causal relation between the precuneus as one of those brain areas and the primary visual cortex.

Overall the manuscript touches the ongoing discussion about neural correlates of consciousness, the tightly related phenomenon of perceptual instability during binocular rivalry and the question about the underlying brain processes and the contributing brain areas. In the last decades the debate about the driving forces underlying spontaneous perceptual reversals during observation of binocular rivalry stimuli and ambiguous figures has been focused on the opposing bottom-up and top-down explanatory approaches. From this point of view the present manuscript contributes an interesting novel perspective together with a similarly interesting data base.

Having said this in the first place, I listed a number of shortcomings of the current version of this manuscript that need to be addressed before acceptance:

General Points

(1) Complexity

Frist of all the extensive set of data analysis tools makes the manuscript highly demanding and comes with a considerable danger to lose a large number of readers, simply due to complexity reasons. I regard it thus as urgently necessary to fulfil some minimal requirements to make the manuscript easier to understand.

1.1. Resolving Shortcuts

One urgent suggestion is, of course, that EVERY shortcut used should at least once be spelled out. In the present manuscript there are many examples, where this is not the case, e.g. “ICN”, “BM” or ECG (in the supplementary material).

We apologise for missing spelling out some terminologies in their first time of usage, and we thank the reviewer for pointing it out. For the revision, we have double checked both manuscript and the supplementary material and made sure that all abbreviations were spelt out in their first occurrence (a special omission case is the measurement unit “kOhm”).

1.2. Provide intuitive explanations

The authors should also provide for any statistical tool 1 – 3 sentences that explain the basic idea. I was personally unfamiliar with many tools and had to stop reading several times, in order to look for myself for the underlying literature and receive a basic idea, what certain methods, where only their names were mentioned, do. A reader should be able to go through the manuscript with at least a basic understanding of the logic underlying a certain analysis step, without extensively studying secondary literature.

To be honest, the help of the extensive supplementary material, provided by the authors, was also very limited in this respect.

We thank the reviewer for this great suggestion. We provided additional explanations to address this point and one that the reviewer made further down. The focus of this particular response is the method complexity that the reviewer pointed out.

The first part of the analyses used standard neuroimaging techniques, namely, fMRI activation and evoked potential analyses for the ERPs. We then used the time-delay embedded HMM (TDE-HMM) method for discovering dynamic patterns within EEG timeseries. Details on TDE-HMM were firstly published in Nature Communications only a few years ago (Vidaurre et al., 2018). Notably, it is not a single method but involves a pipeline of *encoding* (e.g., autoregressive pattern clustering) and *decoding* (e.g., non-negative matrix factorisation) techniques. We have tried to compress the methodological details as much as possible so as not to divert the reader’s attention away from our main point; however, we absolutely agree with the reviewer that we need more intuitive explanations for the novel methods. In this revision, we have provided several lines of text to explain what the algorithms are trying to achieve before presenting the results:

Manuscript page 10, line 229-238:

Dynamic neural pattern analyses

To uncover the endogenous neural activities that trigger the upcoming perceptual change, we applied a dynamic neural pattern analysis to the 1-second (s) time window before every

subjective indication of a perceptual change. We identified the dynamic neural patterns/states by using the time-delay embedded Hidden Markov Model (TDE-HMM) toolbox (<https://github.com/OHBA-analysis/HMM-MAR>) which provides a series of encoding and decoding methods to (1) discover the transient spatiotemporal patterns inherent in the source signals of all trials, (2) extract spectral information (i.e. the power and phase coherence among ROIs in canonical frequency bands) from the states for interpretation. According to our hypothesis: if the intrinsic fluctuation of the DMN regional activity influences the perceptual transitions during BR, we should expect to find a correspondence between the neural dynamics of DMN regions and the upcoming perceptual transitions.

(2) “causal effects”

All of us know that our current knowledge about brain processing does not really allow making strong statements about causation. This was even emphasized by Dr. Granger, when he presented his Granger Causality. The authors should thus be careful with making strong statements about causality (e.g. page 23) ...

We thank the reviewer for making this point, and we understand the reviewer’s concerns. In our revision, we have checked our phrasing and avoided making strong claims about causal relationships. When we were talking about Granger causality as a mathematical measure, we made sure to refer to this as “Granger-causality”. Please see our changes at the following highlighted text:

Manuscript page 1, line 26 (abstract):

The interaction between the lifetime of this state and the PCU->V1 Granger-causal effect is correlated with the rate of perceptual fluctuation.

Manuscript page 18, line 435:

However, this relationship only existed at the group level, but not for trial-by-trial prediction; therefore, it only establishes the behavioural relevance of the Granger-causal effect of PCU to V1.

However, irrespective of Granger causality, we made statements about the endogenous neural activities causing the perceptual change (Line 363, page 15), which is based on the fact that during the experiment, the external stimuli remained unchanged, hence it is fair to assume that the perceptual change can only be driven by the endogenous neural activities. The same logic for inferring “causality” has been used in a previous article, also concerned with pre-stimulus neural activities/states (Britz, Pitts, et al., 2010). For example, they used the phrases like the following: “*It is thus possible that these internally generated alternations arise from a common process that may be reflected by common pre-stimulus brain states that **cause***”

perceptual change.” We are grateful to the reviewer for suggesting we should read this article, which we have referenced in our manuscript.

(3) Roadmap

Given the complexity of the data analysis it would be nice if the authors could provide a more general roadmap displaying the basic steps and their purposes. The authors tried this already in their sFig. 1. However, an additional and much more general type would be helpful. Example:

Method: ICN => Purpose: identifying contributing component brain networks being involved in certain perceptual processing steps (or experimental conditions)

Method: HMM => Purpose: identify and characterize local transient brain states

Method: Granger Causality => Purpose: identify hierarchical relations between brain regions

We thank the reviewer for this feedback. To address this issue, we have made three major changes to the manuscript.

(a) We provided a general roadmap of the various analyses and their purpose in the beginning of the results section. Please see the section as below (change highlighted):

Manuscript, page 6, line 130-142:

Multimodal neuroimaging analyses

Different than the goal of the original study using this dataset, which was to reveal the fMRI activation associated with BR-induced perceptual transitions³⁷, our goal was to test the hypothesis that the endogenous neural dynamics in high-level cortical areas (i.e. DMN) can influence low-level (i.e. primary visual cortex) information processing. Given the controversies regarding the DMN’s role during tasks^{32,34,38}, we first confirmed the involvement of DMN regions in this task using ANOVA F tests with the fMRI dataset. Then using the EEG dataset, we conducted evoked response analyses for two reasons: (1) to narrow down the critical time window preceding a perceptual transition, in order to facilitate further modelling of the dynamic neural process; (2) to validate that the DMN’s involvement is also reflected on the EEG data. Finally, within the specified time window and using DMN source signals, we adopted a time- delay embedded Hidden Markov Model to look for the transient spatiotemporal patterns that may represent endogenous neural triggers of the upcoming, involuntary perceptual changes during BR. The analysis pipeline is presented in the supplementary information (SI; sFig. 1).

(b) We have reorganised the structure of the results section and provided intuitive explanations for each result section in the section titles:

- Results
 - Behavioural analyses
 - Multimodal neuroimaging analyses
 - fMRI activation analysis
 - EEG evoked response analysis
 - Dynamic neural pattern analyses
 - Characterising endogenous neural states that underlie upcoming perceptual transitions
 - State spectral information reveals PCU-V1 phase coherence
 - Behavioural relevance and cognitive implications of the states

(c) We have updated the flowchart of the analysis pipelines. As suggested by the reviewer, the statistics are explanted in the context of their research questions.

SI page 16, line 379-408:

sFig. 1. Analysis pipeline for the simultaneous fMRI-EEG data. With the preprocessed fMRI and EEG, we conducted an initial data exploration (i.e. activation studies for fMRI and evoked-response analysis for EEG) by following standard neuroimaging analysis procedures. To ascertain the task-related regions at the network level, we adopted the Intrinsic-Connectivity-Network (ICN) atlas to evaluate fMRI activation patterns. Since this experiment adopted a continuous BR design where there is no explicit onset of a trial (while the offset is the manual indication of a perceptual transition), we constrained the trial to 1 second (s) by iteratively taking epochs from long to short durations (e.g., from 5 to 3 to 1 s) before a manual response, until we found significant evoked-potential differences between the BR and RPL conditions that were comparable to the previous literature. To initiate the dynamic neural state analysis, we took the source signals (1 s) before every manual report from DMN peak coordinates, shown to be significantly involved in this task by source-level evoked response analysis. With the constrained temporal and spatial range, we set up the time-delay embedded Hidden Markov Model, which is a state-of-art analysis pipeline designed to discover the intrinsic dynamic patterns among a given set of brain regions. We searched for four common spatiotemporal patterns (i.e., states, with a micro-window of ~100 ms) that persist in all trials (of all conditions and participants). The states were estimated at each timepoints of a trial; hence, despite being transient, they were the most robust patterns recurring across time. To establish the states' perceptual relevance, we conducted ANOVA and cross validation analyses, using the states' presence to predict the perceptual type of the upcoming transition for each trial. Finally, to interpret the states, we extracted the spectral information from the states and conducted permutation tests across the states, which revealed the most distinctive neural features of each state. We further focused on certain neural features of the critical state to investigate how intrinsic DMN dynamics may have an influence on primary visual cortex.

(4) Multiple Testing

The authors performed a huge number of statistical tests. They report about FWE corrections. However, I doubt that this is applied on a global level, for really all tests realized in the present analysis. If I am correct with this assumption, I'm fine with it, if the authors make an explicit statement about the statistical limitation that comes with multiple testing.

We appreciate the reviewer's concern regarding the multiple comparison problem. If we understand correctly, the reviewer suggests that we do family-wise error corrections for all the statistics we used in this study. However, we believe that we have followed the standard procedures to correct for the family-wise error rate. Specifically, the fMRI activation study relies on the random field theory to deal with the multiple comparison problem (Nichols, 2012), the multiple comparison for the ERP analyses were dealt with the cluster-based permutation test (Pernet et al., 2015), and we used permutation tests for the behavioural correspondence of the states with corrected family-wise error rate. In fact, in most cases, we did not only rely on the standard frequentist statistics where the alpha inflation problem arises. For example, we have reported the Akaike information criterion for statistical

inferences, in addition to the P value from the frequentist testing framework (page 4, 11, 14, 15). In addition, the algorithm of the TDE-HMM relies on Bayesian statistics, of which the inference was not based on P value but on free energy. Therefore, we feel it would not be appropriate to lump all these analyses together and correct together.

Even for the tests within the frequentist testing framework, according to our understanding of the multiple comparison problem, it is not appropriate to correct for the tests that do not share the same null distribution. The inflation of Type-1 error occurs when the same collection of observed data was used to support a set of statistical inferences, which is especially a case for the post-hoc comparisons in an ANOVA (Lee & Lee, 2018). However, this logic does not apply when the multiple tests are conducted on different datasets, addressing different hypotheses. Because, when significance tests (as well as statistical inferences) are conducted for different datasets, their null distributions are different, and the chance of a finding being a false discovery for one null distribution does not contribute to the chance of false discovery for another distribution. Conversely, over correcting the p value will correspondingly inflate type 2 errors (false negatives), which is not recommended (Althouse, 2016; Jafari & Ansari-Pour, 2019).

(5) The endogenously/ exogenously dominant percepts are used interchangeably with BR/RPL. For ease of reading it might be good to stick to either using endogenous/exogenous or BR/RPL after defining which condition is which

We thank the reviewer for this consideration. There were indeed two cases where we addressed the BR and RPL conditions interchangeably with the terms endogenous and exogenous; we have corrected these and now use BR/RPL throughout the manuscript. When necessary (e.g. line 350, and Figure 1 caption), we kept wording such as “endogenously driven perceptual changes” as we wanted to remind the readers of the essential difference between the BR and RPL conditions.

(6) Time course of perceptual reversals

Understanding the neural processes underlying spontaneous perceptual reversals, the precise time point of the reversal event is necessary, e.g. in order to know which activity is pre- reversal and which activity is post-reversal and probably even secondary to the reversal- related neural processes. Narrowing down the reversal time point, however, is difficult – in the first place, because the reversal is an endogenous brain process and the experimenter only gets to know about it, when the participant communicates her/his first person experience of the endogenous event. Using iris diameters or eye-movement patterns in non-response paradigms (Brascamp et al. 2018) is an interesting perspective but apparently provide not precise enough time stamps.

Several years ago we have designed the so-called onset-paradigm (a simpler version primarily introduced by O'Donnell et al. 1988) to study the chain of processes underlying a spontaneous perceptual reversal of the Necker cube and other ambiguous figures (for a review see Kornmeier and Bach 2012). In a nutshell, the basic idea was to present the stimuli discontinuously (800 ms on, 400 ms off), and therewith to synchronize the onsets of perceptual reversals with the onset of the stimulus and thus use stimulus onset as time reference for the reversal event. This allowed us a remarkable temporal resolution of the processes underlying Necker cube reversals, which we then compared with a related chain of processes when disambiguated cube variants reversed in a computer-driven manner. We found remarkable similarities but also interesting differences. Both our group and Michael Pitts and colleagues demonstrated that the basic idea of the onset paradigm also works with binocular rivalry stimuli and moreover, that similar results can be measured (for a review see Pitts and Britz 2011), although we were not as successful to show this for BR stimuli (O'Shea et al. 2013).

Relevant for the present manuscript are the following findings:

Our group (Ehm et al. 2011) and concurrently and independently Britz et al. (Britz et al. 2009) found right-hemispheric EEG activity in the gap immediately before (around 100-300 ms) an endogenous perceptual reversal. The onset-paradigm thus clearly allowed confining this right-hemispheric activity to a pre-reversal time window. This may be a relevant finding for the authors' discussion in the second paragraph of their page 22.

Further, we found that reaction times jitter intra-individually with ± 100 ms (interquartile range) and discussed this in relation to the reversal event (Kornmeier and Bach 2012). The authors of the present manuscript use participants' manual response as reference time point. They thus need to take into account that the reversal event had occurred about 100 ms earlier. The zero point in their graph is thus not the time point of the reversal but rather the time point ± 100 ms after the reversal. Given our data, the authors may even consider shortening their temporal ROI (in a follow-up analysis or follow-up experiments), which may increase the signal-to-noise ratio of the reversal-related signals. ...

Of course it is always a little problematic if a reviewer requires the authors to cite his own work. However, in the current case this may be defensible: The authors made a big effort to narrow down the processes underlying spontaneous perceptual reversals both in time (focusing on causal aspects) and in space.

In our view our onset paradigm provides the most precise temporal description of the processing chain underlying spontaneous perceptual reversals. And although there are significant differences between classical ambiguous figures, like the Necker cube, and binocular rivalry stimuli, there seem to be also several similarities,

particularly concerning processing chains, as Pitts and Britz (Pitts and Britz 2011) worked out in their review, but as also much earlier Blake and Logothetis pointed out in their seminal review (Blake and Logothetis 2002). Furthermore our findings have since been replicated by several labs around the globe (e.g. Pitts et al. 2007; Intaite et al. 2010; Sandberg et al. 2014; Kornmeier et al. 2019; Abdallah and Brooks 2020). We thus think that this work needs to be at least discussed in relation to the authors' findings.

We thank the reviewer for their suggestions. We have read through the literature recommended by the reviewer, and extended our discussion of the present results where appropriate given the journal format constraints. We have already referenced three of Blake (and Brascamp)'s more recent review papers (Blake et al., 2014; Knapen et al., 2011; Tong et al., 2006) in our original manuscript. We thank the reviewer for suggesting the reviews by Pitts and Britz (2011), Kornmeier and Bach (2012) and Brascamp et al. (2018), which we think are highly relevant to our work. Please see the following additional references we have added as suggested:

23. Brascamp, J., Sterzer, P., Blake, R. & Knapen, T. Multistable Perception and the Role of the Frontoparietal Cortex in Perceptual Inference. *Annu. Rev. Psychol.* 69, 77–103 (2018).
49. Kornmeier, J. & Bach, M. Ambiguous Figures – What Happens in the Brain When Perception Changes But Not the Stimulus. *Front. Hum. Neurosci.* 6, 51 (2012).
53. Britz, J., Pitts, M. A. & Michel, C. M. Right parietal brain activity precedes perceptual alternation during binocular rivalry. *Hum. Brain Mapp.* 32, 1432–1442 (2010).
54. Britz, J., Landis, T. & Michel, C. M. Right parietal brain activity precedes perceptual alternation of bistable stimuli. *Cereb. Cortex N. Y. N 1991* 19, 55–65 (2009).

After reviewing the reference list kindly provide by the reviewer, we appreciate the merit of the intermittent BR paradigm that the reviewer proposed. Not only can it effectively alleviate the problem of getting stuck in one dominant image for too long, which often happens in participants, it also provides the opportunity to do stimulus-locked ERP analysis, thus affording the opportunity of identifying the reversal event with greater temporal precision. However, our study is based on the publicly available dataset, which adopted a continuous BR paradigm; therefore, the response-locked ERP analysis is really the only option we were left with.

We are aware of the disadvantage of the response-locked ERP analysis adopted in the study. Please see our statement of this limitation (Because of the word limit imposed by the journal, we had to move this part into the supplementary materials):

SI page 7, Line 161-167:

We acknowledge the limitation of the response-locked analyses (as constrained by the experimental design), as the epochs incorporate a mixture of perceptual and motor processes. If we accept RPL as a perceptually-matched condition, any contrasted differences between the BR and RPL should elucidate the endogenously-driven neural features that are distinct in the BR condition, rather than the noise or motor-related features that are assumed to be common to both conditions.

We appreciate the reviewer's concerns regarding the temporal precision of the reversal timing. To identify the exact moment of visual awareness is indeed a problem, both theoretically and practically. However, we wanted to point out that the analysis we used to investigate the pre-perceptual endogenous neural activity is not the response-locked evoked analysis, but the time-delay embedded Hidden Markov Model (TDE-HMM); and identifying the precise moment of the reversal moment is not necessary for this analysis. Firstly, the TDE-HMM analysis was conducted at the single trial level, while the +/-100 ms that the reviewer suggested to disregard from the time window of interest was based on the group-average results, which means that most reversal events do not happen exactly at the 100 ms before a manual response, and hence not being precise either. Although we understand the motivation of leaving out the last 100 ms time window from our analysis, we do not think its benefits outweigh its drawbacks, such as the risk of missing out relevant information as well as the inter-trial variability contained within the 100-ms neural signals. Secondly, the TDE-HMM is a data-driven method. Based on the data structure, the state characterisation will be computed at each timepoint of a trial. If a state was discovered solely from the last 100 ms of a trial, which might be related to post-reversal perceptual or motor confounds as the reviewer pointed out, we should be able to see its higher presence during the last 100 ms. But in fact, the critical state, State 4, which dominates the pre-response trials of the BR (dominant) condition, did not show stronger presence (quite the opposite) in the last 100 ms time window (sFig. 7). Therefore, we think our statement about the pre-perceptual endogenous neural activity holds.

SI page 7, line 459-462:

sFig. 7. The frequency (or proportion) cross trials of the State 4 being present at each time point in the BR (dominant) condition.

We will next discuss our justification for utilising a 1-s time window for the TDE-HMM dynamic model:

SI Page 9-10, Line 220-244

Selection of time window and regions of interest for the TDE-HMM dynamic process modelling

Instead of modelling the dynamic process for the whole brain during the whole experiment, we narrowed down the time window and regions of interest (ROIs), in order to increase the method's sensitivity and avoid overfitting¹⁰. Our previous evoked-response analysis helped us to narrow down the time of interest to a window between [-400 -200] ms before a subjective report of perceptual change, which is when the pre-response ERPs showed the biggest GFP difference between BR and RPL. However, the endogenous neural activity triggering an upcoming transition might take effect even earlier than that¹¹. Therefore, we modelled the dynamic process in the 1-second (with 100-ms post-response padding) window before every report of a perceptual change. The 1-s window was chosen as a trade-off between having enough sample points and having enough specificity to the targeted events. It has been suggested in the literature that endogenous perceptual disambiguation is associated with posterior parietal activity 50 ms before the onset of a bistable stimulus¹²; and the upper limit of reaction time to it is about 600 ms^{11,13}, while the upper limit of pure motor execution is about 150 ms^{11,14}. Therefore, the 1-s window should be able to cover the whole dynamic neural process towards a perceptual transition and a little further before. 25767 trials were generated in total, considering all 4 conditions: BR/RPL (dominant) (i.e., transitions from a mixed to a dominant percept in the BR or RPL setting; n=7795/7148), and BR/RPL (mixed) (i.e., transitions from a dominant to a mixed percept in the BR or RPL setting; n=5786/5038). As the algorithm computes the latent states recurring across all timepoints of all trials, the state delineation is assumed to be driven by the endogenous neural states which have a robust spatiotemporal structure recurring through time^{10,15-18}.

Finally, we wanted to justify our use of the TDE-HMM method. Traditional methods using evoked potentials based on grand averaging have enriched our knowledge about visual awareness, but we are still largely agnostic what those early components upon a stimulus onset actually mean. In reality, neural activities do not exist as discrete events, but rather, all the neural activities happening before an explicit motor response (including the pre-perceptual, perceptual and pre-motor) constitute a congruent process which leads to the response. Using the TDE-HMM technique, we are able to model the whole 1-s window as the continuous process that leads to the response; and this, we believe, is the advantage rather than a drawback of the present study.

Major Points

- Page 2: "...We know that external perturbations can interrupt or bias the global cascades of information propagation (e.g. visual masking and priming) ..."

In the sentences before the cited one, the authors elaborate about resting state brain activity. I think in the cited sentence they write instead about processing of sensory information. However, this is not entirely clear. By this reason it is also not really clear how visual masking and priming fit into this sentence. The authors should explain the ideas behind this sentence in more detail and add some references from the priming and masking literature.

We thank the reviewer for this feedback. By introducing visual masking or priming we were meant to give an example of how external perturbation influences the "information propagation" for visual processing, as an analogy to the internal perturbation caused by the intrinsic neural activity. However, we realised that this analogy might cause more confusion than it settles, so we just focused our arguments on the intrinsic neural activities in this newer version (see changes below).

Manuscript page 2, Line 34-47:

Being aware is believed to be a globally "illuminated" inner state when locally encoded information gets propagated through downstream pathways and becomes accessible to other processing streams in the brain¹. However, the mechanism of the propagation process is largely unknown. Recent theoretical developments in brain dynamics suggest that the spontaneous information propagation may be empowered by the intrinsic ignition of neural activity². Empirically, intrinsic brain activity has been extensively studied recently during resting state when no external stimuli are presented to participants. At rest, neural activity from certain distant regions is correlated forming what are known as large-scale networks³. We hypothesise

that the information propagation during the state of “being aware” can be influenced by intrinsic perturbations from the endogenous dynamics of large-scale brain networks, by which the brain’s intrinsic activity might cause a “butterfly effect” to the downstream perceptual or cognitive events. In this study, we used the well-known Binocular Rivalry (BR) paradigm, with a simultaneous functional Magnetic resonance imaging (fMRI)-electroencephalogram (EEG) implementation, to investigate whether and how intrinsic brain activity of a large-scale brain network influences the involuntary perceptual fluctuations during bi-stable visual awareness.

- Page 2: In 2001 Randolph Blake provided an encompassing review about BR. The authors should consider citing this profound work (Blake 2001), when they introduce the BR stimuli.

We thank the reviewer for suggesting this review paper. We cited one of Blake’s more recent review papers on BR (Blake et al., 2014) when we introduced the BR stimuli. We also cited another two review papers by Blake and used some of his arguments in our paper. We agree that the review paper published by Blake (2001) on Nature Neuroscience is profound and influential, hence we have cited that in our update version as suggested.

- Page 3, end of paragraph 1: Sentence is difficult to read “...However, the intrinsic dynamics of brain networks may have a lot of interactions with, and by which modulate, the activity of local sensory regions.”
Better: “However, the intrinsic dynamics of brain networks may have a lot of interactions with the activity of the local sensory regions, and as a result modulate said region.”

We thank the reviewer for this suggestion, please see the following change:

Manuscript, Page 3, Line 64-66:

However, the intrinsic dynamics of brain networks may have a lot of interactions with primary sensory regions, and as a result modulate the local sensory processing (Brascamp et al., 2018; Hohwy et al., 2008; Mashour et al., 2020; Pezzulo et al., 2021).

- Page 5:
 - define “RSS”
 - “Akaike Information Criterion“ => provide at least one informational sentence about what this measure does ...

We apologise for the omission of the statistical details, please see the updated sentence:

Manuscript, page 4, line 109-111:

Individual differences of the percept duration during BR were significant (Fig. 1b) (Residual Sum of Squares [RSS] was reduced by 1490, $p = 0.00$ according to a χ^2 test, and the Akaike Information Criterion [AIC] was decreased by 1057.5. Both measures suggest that there was stronger model evidence for a model that considers individual differences comparing to the null model).

▪ Page 6: The present manuscript only indicates in a sub clause, that parts of the data have already been analyzed and the results published in another paper. The authors should make this more explicit. Moreover, they need to indicate what the major difference and “unique selling point” of the present data analysis in comparison to the previous publication is.

We appreciate the reviewer’s concern about the reuse of a previously published dataset in the current study. We made this information explicit in the methods section (SI page 2, 3, 4). We have also indicated this information in the figure legend in page 5. In this newer version of the manuscript, we made it more explicit in the result section (page 4, line 99).

Results

Behavioural analyses

This experiment has been previously validated, analysed and published (Jamison et al., 2015; Roy et al., 2017).

Although we used previously published dataset, our study is novel both in its research questions and methodologies. There were two previous publications using this dataset. The earlier one by Jamison et al. (2015) used this dataset to introduce a data analysis technique, and is not concerned with BR or visual awareness per se. The more recent publication by Roy et al. (2017) investigated fMRI activation during BR. We used their preprocessed datasets as we didn’t have access to their raw data, but our study is different from theirs in terms of the experimental questions we asked and the methodological approaches we adopted. According to the reviewer’s suggestion, we have added a few sentences in the result section to make this point clearer (manuscript Page 6, Line 131-133):

Multimodal neuroimaging analyses

Different than the goal of the original study using this dataset, which was to reveal fMRI activation associated with BR-induced perceptual transitions³⁷, our goal was to test the hypothesis that the endogenous neural dynamics in high-level cortical areas (i.e. DMN) can influence low-level (i.e. primary visual cortex) information processing.

Page 6, Fig. 1: It is reasonable to represent data from red and green stimuli in the same colors (Fig. 1). However, it may be hard to distinguish for some of your readers (9% of men, 0.8 % of women). Consider using shadings in addition to color coding. Further, it is unclear to me why in Fig. 1d there are symbols (triangle and diamond) behind the boxplots. Does this address the color-blindness problem? To me it looks confusing and a simple shading would already do the trick with only little visual clutter.

We thank the reviewer for pointing out potential issues with our colour choices. In fact, we have used colour-blind friendly palette for plotting ([http://www.cookbook-r.com/Graphs/Colors_\(ggplot2\)/#a-colorblind-friendly-palette](http://www.cookbook-r.com/Graphs/Colors_(ggplot2)/#a-colorblind-friendly-palette), 6th section: “A colorblind-friendly palette”). As the reviewer suggested, we have tried textile/shape filling for the box plot, as shown below.

To be extra safe, we also tested out our visualisation from colour-blind perspectives, tested with the online colour-blind simulator: <https://www.color-blindness.com/coblis->

color-blindness-simulator/. Please see the tested results as below. We think our colour choices are acceptable.

	Blue weak	Red/green weak	Blue blind	Red/green blind
Fig. 1 Colour choices	Percept  Green  Mixed  Red	Percept  Green  Mixed  Red	Percept  Green  Mixed  Red	Percept  Green  Mixed  Red
Fig. 5 Colour choices	States  K=1  K=2  K=3  K=4	States  K=1  K=2  K=3  K=4	States  K=1  K=2  K=3  K=4	States  K=1  K=2  K=3  K=4

For the Fig 1d, we have changed the visualisation as below, hoping this will make the message clearer.

- Page 8, Fig. 2: Define FPCN and ICN

We thank the reviewer for pointing out the problem with the acronym. We have fixed it in the revision. Please see the revised figure below.

Manuscript Page 6:

- Page 8, line 2 of section “EEG evoked responses and source reconstruction”: “The EEG recording with its excellent temporal resolution gave us the opportunity to better locate the neural cause of a perceptual change...” How do we know if it is a cause? Could it not simply be a correlation?

Here we used the phrase “neural cause of a perceptual change” because the perceptual content changes when the external stimuli don’t; and hence we assume the change originates from endogenous neural activities.

We absolutely agree with the reviewer that the evoked potential (-400 ~ -200 ms) that we reported in this section did not necessarily represent endogenous neural activities that “cause” the upcoming perceptual change. Indeed, previous literature suggests that the perceptual processes about 350 ms before a manual response are already post-awareness (Kornmeier & Bach, 2012), possibly indicating realisation of

a perceptual transition. Please see the revised text where we explicitly pointed this out:

Manuscript page 8, line 186-189:

Though based on response-locked analyses, this result is comparable with previous literature dominated by stimulus-locked analyses, which suggests that the P300-like ERP component from parietal activity (roughly happening 300 ms before a key press) probably indicates a conscious recognition of a perceptual disambiguation⁴⁰.

To avoid misunderstanding, we have made sure that no inference about endogenous neural cause was made in the evoked-response analysis section, and we have rephrased the misleading sentence that the reviewer pointed out.

Manuscript, page 8, Line 180-182:

EEG evoked response analysis

The EEG recording with its excellent temporal resolution allowed us to investigate a short time interval right before manual indication of a perceptual change.

However, we would like to add here that our TDE-HMM analysis allows us to make inferences about possible endogenous neural causes. We have discussed this in more detail in response to the reviewer's earlier comments.

-
- Page 8, Fig. 2: BM14 is listed on the circular plot but not in the text next to it.

To avoid too much text in the figure, we chose not to provide descriptions for ICN-BM14, along with other domains that have miniscule involvement in the task. In the revised version, we listed these cognitive domains in the figure legend.

Manuscript Page 7:

Fig. 2. FMRI activation revealing the DMN's involvement in the current task. Subplots (a), (b) and (c) show the significant clusters [$P_{\text{voxel}} < 0.001$ (uncorrected) & $P_{\text{cluster}} < 0.05$ (family-wise error corrected)] respectively for the contrast "BR (dominant) - RPL (dominant)", "BR (mixed) - RPL (mixed)" and the interaction effect between perceptual generating (BR vs. RPL) and perceptual (dominant vs. mixed) conditions. The interaction analysis (c) shows the regions mostly sensitive to the condition differences. For statistical testing, we adopted linear mixed-effects models, where the paired T-tests and F-test were carried out at the individual level, while statistical inferences were made at the population level with group-level one-sample T tests (input being the individual-level estimators). Hence, the colour bars of the subplots (a)-(c) indicate the T scores from the group-level testing. The circular plot in (c) shows the ICN affiliation of the significant clusters for the interaction effect. "ICN involvement" is a measure of correspondence between an activation map and large-scale networks with well-established cognitive function, as provided by the BrainMap (BM) meta-analysis database³⁹. The BM number on the circular plot indicates the number of the ICN-BM network atlas. Cognitive domains of the networks that have negligible involvement in this task are not listed in the figure. These are BM1: Limbic and medial-temporal areas; BM3: Bilateral BG and thalamus; BM4: Bilateral anterior insula/frontal opercula and anterior cingulate gyrus; BM5: Midbrain; Cerebellum; BM6: Superior and middle frontal gyri; Sensorimotor; BM9: Superior

parietal lobule; Frontoparietal (perception-somesthesia-pain); BM10: Middle and inferior temporal gyri; Frontoparietal (cognition-language); BM11: Lateral posterior occipital cortex; BM14: Cerebellum; BM16: Transverse temporal gyri; BM17: Dorsal precentral gyri, central sulci, postcentral gyri, superior and inferior cerebellum; BM19: Artefactual component; BM20: Artefactual component.

- Page 10: What is the exact logic behind the selection of the time window for voltage map subtractions in Fig. 3d right.

In Fig 3d, we wanted to demonstrate the exact time evolution of topographies that gives rise to the GFP difference across the two perceptual sources during the dominant percept. Hence, we chose a wider, continuous time-window of [-450, -210] ms, and decided to plot voltage map frames at every 80 ms. We note that the 80ms snapshots are chosen arbitrarily for visualization purposes only, in order to fit 4 equidistant voltage topographic frames in time.

To avoid confusion, we have rearranged this figure as shown below. The contrasted ERP topography was shifted to the subplot (a), above the time frame during which there were significant differences between the BR and RPL conditions.

Manuscript Page 9, line 210-226:

Fig. 3. Event-Related Potentials for different perceptual conditions (a) Event-Related Potential contrast between BR and RPL for dominant percept at the sensor level. The global field power (GFP) in the RPL condition was significantly higher than that in BR condition from -400 to -200 ms before the reported change in percept at 0 s. This is indicated in the plot with the black line ($p\text{-corr} < 0.05$

significant clusters, by 1-sample permutation cluster t-test). The ERP topography for the “BR (dominant) vs. RPL (dominant)” contrast is displayed at the top of the significant time window. (b) No significant difference was found in the GFP between the BR (mixed) and RPL (mixed) conditions. (c) GFP difference across sources of perceptual transitions and types of percept. The interaction effect was not significant ($p\text{-corr} = 0.12$). The plot shows the averaged GFP across the time window between -400 and -200 ms and its 95% confidence intervals for each group. (d) ERP topographies in both of the BR and RPL conditions, dynamically changing during the critical time window ([-450 -210] ms) before a response to dominant percept. (e) Source localisation of the evoked responses contrasted between BR and RPL conditions, and for the interaction effect of the two variables, during the [-400 -200] ms before a subjective report. Plotted brain regions/voxels survived the significance test with FWE-corrected non-parametric P-values < 0.05 .

- Page 11, line 22: “...We targeted our dynamic analyses at the temporal window of the second (with 0.1-s post-response padding) before every perceptual transition as a trade-off between having enough sample points and having enough specificity for the targeted events.”

This should read: “... before every response of a perceptual transition...”

We thank the reviewer for spotting this imprecise description, we have changed accordingly:

before every response of a perceptual transition as a trade-off between having enough sample points and having enough specificity for the targeted events.

- Page 12: The arguments for the a priori choice of 4 states is not entirely clear to me.

The authors wrote: “...We chose 4 states because firstly we were interested in the differences of the neural activity among the four experimental conditions...”

Does this argument make sense if one can assume that each of the four (or more) microstates can occur within each of the four conditions? Exactly this is what I observe in Fig. 4 bottom right. This point needs more clarification!

The initial selection of the number of components in this kind of method is usually arbitrary. In general, specifying 4 components makes the results easier to understand, since we have 4 perceptual conditions. In this instance, the four states we found turned out to be meaningful as explained in the manuscript. For the record, we tried different number of components, but we didn't report these results to keep the narrative of the paper as concise as possible.

To clarify this, we have revised the text as below:

Manuscript Page 10, Line 239-242:

We set the HMM algorithm to extract 4 states among the regional signals of all trials from all conditions and subjects (Fig. 4). Four components were specified as we expect the states to be interpretable in terms of the experimental conditions, supposing that the visiting time of the four states in the four conditions would be significantly different. In addition, 4 components have often been specified for discovering the EEG microstates in existing literature⁴¹, and were shown to have correspondence with the ICN dynamics measured from the BOLD signals⁴². The resulting auto-covariance patterns of the 4 states are presented in the SI (sFig. 4). All of the comparisons were carried out within subjects and the difference was then grouped together for the population-level inference.

The reviewer was right that the hidden states were the common features of all trials, regardless of the conditions. This is because the states were estimated directly from the data without being contrasted. This is a feature of data-driven analyses, whether it is EEG microstates (da Cruz et al., 2020), or the TDE-HMM states (Quinn et al., 2018) that we adopted here. We assume that there are several fundamental spatiotemporal patterns underlying all trials, regardless of the conditions. Although the hidden states are common, their presence (i.e., *fractional occupancy* in a probabilistic form, and *lifetime* in a deterministic form; both indices were used in this study) can vary among conditions.

-
- Page 13, Legend to Figure 4: "... Across all the experimental conditions, signals from significant peak voxels within the same ROIs were averaged ... " For me this sentence is not clear. Essentially the authors have spatial and temporal ROIs. I suppose they average across the data within a temporal ROI but I am not sure. This should be explained in more detail.

We thank the reviewer for this suggestion, and we agree that this sentence was not very clear. The ROIs that the reviewer refers to are spatial ROIs. They are spatially constrained within the anatomical boundaries which are commonly considered to make up the DMN, and further constrained to the coordinates that were shown to be significant in all of the experimental contrasts. [i.e. BR (dominant) vs. RPL (dominant); BR (mixed) vs. RPL (mixed); and the F test among the four conditions]. Based on their spatial localisation, we extracted the source signals uniformly 1-s before every manual indication of a perceptual change. When more than one peak voxels were identified within the same anatomically-defined region, the average of their signals was taken as representative timeseries for the ROI. In the revised version, we described this procedure in more details and shifted it to the supplementary materials due to the word limit:

SI Page 10-11, Line 245-265:

Given our research interest in the default mode network (DMN) we constrained the subsequent modelling within the 8 regions: bilateral parahippocampal gyri (HP), bilateral inferior parietal lobules (IPL), ACC, PCC, PCU, and the primary visual cortex (V1). To ensure experimental sensitivity, we extracted signals from the significant peak voxels (confined within the DMN) from the previous source-level evoked-response analysis. The contrasts used for determining the regional involvement in the task were BR (dominant) vs. RPL (dominant), BR (mixed) vs. RPL (mixed) and the interaction effects between the two factors; both contrast directions were considered. Hence, 1-second (+0.1s post-response padding) source signals from these coordinates were extracted for all trials (trials being the 1-s epoch before a subjective report of a percept change). When multiple coordinates were identified within a same ROI (identified by automated anatomical labelling), the average of their signals was used as the representative signal of that ROI. The anatomical labels for the peak coordinates were identified using the Talairach Atlas (<http://www.talairach.org/>), upon a conversion to the Talairach space. The list of the peak voxel coordinates used for this purpose are presented in Supplementary material Table 2. The full results of all significant voxels/clusters upon all contrasts are available from an online repository:

https://htmlpreview.github.io/?https://github.com/Aubrey-Lyu/BR-project/blob/master/Analysis-3_EEG/results/evokedResponse_result_table_permutationtest.html

-
- Page 14: The authors should clarify that a mixed perceptual transition means a transition from a stable percept to a mixed percept (if I am right) ... and the respective for dominant perceptual transitions ...

We appreciate this suggestion, please see change as below.

Due to the limitation of word count, we have shifted this part to the SI:

SI page 10, line 236-244:

Therefore, the 1-s window should be able to cover the whole dynamic neural process towards a perceptual transition and a little further before. 29785 trials were generated in total, considering all 4 conditions: BR/RPL (dominant) (i.e., transitions from a dominant to a mixed percept in the BR or RPL setting), and BR/RPL (mixed)

(i.e., transitions from a mixed to a dominant percept in the BR or RPL setting). As the algorithm computes the latent states recurring across all timepoints of all trials, the state delineation is assumed to be driven by the endogenous neural states which have a robust spatiotemporal structure recurring through time (Cole et al., 2014, 2016; Margulies et al., 2016; Theodoni et al., 2011; Vidaurre et al., 2018).

- Page 18: “... In the BR literature, it is commonly believed that a switch of perception is caused by a slip of equilibrium due to neuronal noise, and/or the constant decay of suppression that one neural population has over the other, due to habituation effects (10, 11, 44). Apart from these, attention effects have also been shown to play a role (10, 45). Therefore, the PCU→V1 top-down modulation could reflect either an intrinsically evoked attention effect, or an effect of the global adaptation/perceptual habituation. These two possibilities would lead to opposite trial-by-trial predictions. ...” I cannot completely follow these considerations. There is indeed a debate about whether perceptual reversals are driven in a bottom-up manner or rather in a top-down manner. I am fine up to this point. However, attention-induced higher reversal rates are only reported if participants are explicitly instructed to volitionally increase reversal rates. The latter is more effective for classical ambiguous figures than for BR stimuli. However, without instruction, I do not see the case that higher reversal rates should be expected.

We appreciate the reviewer’s feedback. Due to the word count limitation, we deleted the paragraph that the reviewer pointed out above, which detailed the motivation behind the hypothesis testing of the trial-level linear regression between the GC (PCU->V1) and the time interval before a perceptual change. This test was not significant at the trial level, and a lengthy motivation of this test might divert the audience’ attention away from the main point.

Further, we feel the reviewer’s statement about the attention’s effect on BR does not entirely reflect the literature. Although volitional control over BR is limited, it is well known that both voluntary and involuntary attention can modulate the BR rates (Paffen et al., 2006; Paffen & Alais, 2011; Watanabe et al., 2011). Contrary to the effect of voluntary attention which can prolong the duration of dominant percepts (Blake & Logothetis, 2002; Chong et al., 2005), involuntary attention was suggested to have an opposite effect. Notably, in Blake’s influential Nature review (Blake & Logothetis, 2002), it was suggested that *involuntary attention* can rescue a stimulus from suppression, thrusting it into conscious awareness, and thus shortening the reversal interval. This hypothesis is supported by the empirical study, where the involuntary attention to the BR stimuli was found to speed up the reversal rate (Paffen et al., 2006). In a more recent Neuron paper, it was shown that perceptual alternation ceased when no attention was paid to the BR task itself (Zhang et al.,

2011). This alludes to the well-known theoretical proposition that visual awareness (or access consciousness) intrinsically requires attention (Tsuchiya & Koch, 2008). Based on the empirical evidence and theoretical argument, we could hypothesise that attention (most likely involuntary attention in our case) could induce higher alternation rate in BR perception; and the PCU's involvement might indicate this attention effect given its functional affiliation and anatomical adjacency to the fronto-parietal control network.

- Page 20, Figure 6: Please define “dBIC”
In the revised version we gave the full name of the term.

Their relationship is also dependent on the lifetime of State 4. The GC effect size was approximated by the improvement of model evidence (parameterised by decreased **Bayesian Information Criterion** (dBIC)).

- SM, Equations 1 and 2: If you provide a mathematical formula you should also define the different variables used in this formula.

We apologise for this omission, please see the update as below:

SI page 14

Mathematical equations:

$$X_1(t) = \sum_{j=1}^p b_{11,j} X_1(t-j) + E_1(t) \quad (1)$$

$$X_1(t) = \sum_{j=1}^p b_{11,j} X_1(t-j) + \sum_{j=0}^p b_{12,j} X_2(t-j) + E_1(t) \quad (2);$$

where X_1 , X_2 are two timeseries for $t = 1 \dots T$, $E_1(t)$ is a white Gaussian random vector, $b_{11,j}$ and $b_{12,j}$ are the correlation coefficients respectively for the autoregressive model of X_1 , and the multivariate-autoregressive model between X_1 and X_2 , for every backshift of j within the maximal time-lag p . For model 1, we used the “best” p adaptive to the V1 timeseries, for maximising the variance that can be explained by the null, i.e. autoregressive, model. Since we are agnostic to the time lag that the PCU (i.e. X_2) is supposed to lead V1 (i.e. X_1), for model 2, we estimated the models with all possible values for p from 1 to 20 (leaving at least 20 time points for the coefficient to be robustly estimated). The improvement of model 2 relative to model 1 was parameterised by the decrease in BIC (dBIC). Averaged dBIC across all p from 1 to 20 was calculated as an unbiased evaluation of the GC effect.

- SM, Page 3: Please specify how gradient artifacts were removed from the EEG.

The publicly available dataset has been preprocessed (Jamison et al., 2015). According to their procedure, the gradient artifact was removed using a PCA-based optimal basis set (OBS) algorithm. This information has been added in our revision: SI Page 4, Line 84-86

Gradient artifacts were removed using a PCA-based optimal basis set (OBS) algorithm, and cardioballistic artifacts were removed based on a combination of ICA, OBS, and an information-theoretic rejection criterion (Jamison et al., 2015).

- SM, Page 4:
 - What is the meaning of ResMS
 - What is the trial length for the fMRI analysis?

We thank the reviewer for pointing out. In the revised SI, we have specified the ResMS images as the images of “residual sum of squares/variance”, a name adopted from the SPM convention. And we also added the number of the fMRI volumes that we have got in this dataset. Please see updated text below:

SI page 3-4, line 72-78:

129 volumes were acquired for most of the participants, which covered the whole experiment and aligned temporally with the EEG recording. There were 7 participants who had longer scanning sessions (with 130-141 volumes), but no participant’s data were discarded. During the scanning session, the timing of each volume acquisition was recorded and used for event-related activation study. As a convention, the first 5 volumes were considered unreliable due to the initiation of the scanner and have been excluded for the following processing.

-
- SM, Page 5: Was there an exclusion criterion for participants that show a lot of movement artifacts? (e.g. participants with more than 10 % of bad volumes are excluded from the study). In case you did not have this criterion, why so?

We thank the reviewer for pointing this out. No participant's data were excluded for our study. The criterion for discarding a whole-session has been suggested to be 30% bad volumes of the resting-state sessions for a clinical cohort, according to the officially released publication of the motion-scrub toolbox *ArtRepair* (Mazaika et al., 2011). However, task-based data are generally more reliable than the resting-state, and so is the healthy young adult group versus the clinical cohort. This dataset has been quality controlled by the research group that made it publicly available. There was only one scanning session (out of 10) for one participant that had 34% bad volumes.

-
- Page 5: „One sample T-tests against zero for finding baseline activation for each condition were also been conducted in order to provide priors for EEG source reconstruction.“

Better: “have been”

We are grateful for the reviewer's careful read. We have changed the wording accordingly.

-
- SM: Figure 2:
 - You should explain what is meant by FE
 - The text describes negative free energy. The ordinate shows positive values. Please explain this.
 - You should add the following sentence to the legend of Figure 2: “The higher the Negative Free Energy, the more evidence the model presented.”

We apologise that this confusion arose. We conducted the analysis by following this methodological paper (Henson et al., 2010; the pioneer paper using variational Bayes to judge the fMRI priors' utility). In this paper, in Figures 5, 9, 10 the Y-axis of “Negative Free Energy” indicated positive values. However, by revisiting the mathematic computation of this value, we believe that the reviewer was correct that the negative free energy is a negative value.

Model Evidence

Given a probabilistic model of some data, the log of the evidence can be written as

$$\begin{aligned}\log p(Y) &= \int q(\theta) \log p(Y) d\theta \\ &= \int q(\theta) \log \frac{p(Y, \theta)}{p(\theta|Y)} d\theta \\ &= \int q(\theta) \log \left[\frac{p(Y, \theta)q(\theta)}{q(\theta)p(\theta|Y)} \right] d\theta \\ &= \int q(\theta) \log \left[\frac{p(Y, \theta)}{q(\theta)} \right] d\theta \\ &+ \int q(\theta) \log \left[\frac{q(\theta)}{p(\theta|Y)} \right] d\theta\end{aligned}$$

where $q(\theta)$ is the approximate posterior. Hence

$$\log p(Y) = -F + KL(q(\theta)||p(\theta|Y))$$

(from UCL online slides:

<https://www.fil.ion.ucl.ac.uk/~wpenny/bayes-inf/variational-ucl.pdf>)

The confusion arises probably because the negative sign was arbitrarily put into the first term (“-F”), just so that the Free Energy can be positive so as to facilitate interpretation and model comparison. That was also probably why in the literature negative free energy and FE can be used interchangeably (e.g., see López, et al. 2014). To avoid confusion, we will stick with the term of Free Energy in the article.

We have reproduced the plot with the Y-axis label as “Free energy” and have also added the requested information in the figure caption (See the change below):

SI page 17:

sFig. 2. Model comparison for different inverse modelling configurations. The free energy, which approximates the log model evidence, is used for model comparison. The inverse model (No. 2) with the highest free energy was chosen for further analyses (Henson et al., 2010; López et al., 2014). The model specifications are described in the Supplementary table 1.

-
- SM: Figures 3 and 4: The color bars have no units

The bars in the original SM Fig. 3 & 4 indicate T-score, which is a unit-less measure. In the updated version, we integrated the improved manuscript figures. Please see Fig. 2 and 3e in the manuscript, where we have labelled the colour bar as T-score.

- SM, Page 6:
 - Please provide a version of MNE-Python
 - The link to the MNE-Python homepage is given twice

We have added a version and deleted the second URL for the MNE-python homepage.

- SM, Page 7: The forward model should be described in more detail. Did you use boundary element method or finite element method? How many shells/ tissues did you differentiate? What conductivities did you assume for your forward model?

We apologise for omitting this information and we are grateful that the reviewer has pointed this out. We used the 3-shell spherical method for the forward model. Although we appreciate that the boundary element method is a more sophisticated and realistic than the 3-shell spherical head model, it is more computationally intensive and prone to errors when there is any mishap in the extracted brain tissues, e.g., scalp, skull and brain tissues (Michel & Brunet, 2019). We used a default assumption of their conductivity ratios, which are commonly specified to be 1, 1/80, 1. Hence, their conductivities are assumed to be 0.3300, 0.0042 and 0.3300 S/m respectively. Please see additional information in the SI as below:

SI page 8, line 184-187:

We used the 3-shell spherical method for the forward model. Scalp, skull and brain tissues were segmented. Their conductivity ratios were specified to be 1, 1/80, 1 (conductivities being 0.3300, 0.0042 and 0.3300 Siemens/meter, respectively).

- SM, Page 7: There are dozens of different approaches to calculate inverse solutions. Please specify your motivation to choose sLORETA and minimum norm estimates/IIID. Why did you not select a more recently developed and more sophisticated inverse solver, e.g. multiple sparse priors (MSP) or cMEM? Beamforming could have been another suitable option since it performs well on noisy environments (e.g. in a scanner) and in the frequency domain.

We are grateful that the reviewer gave us a chance to explain our algorithm selection. We acknowledge that the inverse problem is tricky, as it can have infinitely multiple solutions but we lack a ground truth to validate them. We can compare

different models from a statistical point of view, but that does not necessarily ensure a most biologically plausible solution. Although there have been more sophisticated solutions, we believe that a more complicated method does not always mean better. We used the sLORETA and IID because these two are native to the SPM toolbox that we are familiar with. Especially, sLORETA is a physiologically justified model with the assumption that activity in neighbouring voxels is correlated. Furthermore, according to a recent review, the LORETA-family algorithms were favoured for their physiological plausibility (Dattola et al., 2020).

MSP was originally proposed by the SPM group and is also available in the toolbox (Friston et al., 2008), but its biggest advantage (i.e., the usage of sparse priors) overlaps with our application of the empirical fMRI-activation priors. In fact, it has been shown that MSP does not improve the result more than what the use of fMRI priors already does (Henson et al., 2010). This is predictable because MSP searches for the optimal priors among the assumed several hundred sparse priors; if fMRI activation is assumed to reflect neural activities (hence the most plausible priors), then the calculation for other priors in the MSP algorithm should be redundant. Combining fMRI priors with the EEG source reconstruction is however the biggest advantage of using this multimodal dataset. For those reasons we did not use MSP. The reason we did not use Beamforming, apart from the fact that it is not available in SPM, is that we could not find a solution of integrating fMRI priors with the Beamforming technique. Since our choice of the methodology is standard (especially for SPM users) (Henson et al., 2019), and the above reasoning is lengthy and beyond the main point of the article, with the reviewer's agreement, we would like to leave the reasoning out of the main text.

-
- SM, Page 7: Typo: "Signal hanging" probably means "signal hanning". We thank the reviewer for spotting this typo. We have changed it accordingly.

-
- SM, Page 8:
"...The Hidden Markov Model (HMM) as a general framework assumes a hidden sequence of a finite number of states which drives the observed time series (7). The data at each time point can be explained by a mixture of states, where the mixture weights are the state probabilities. ..."
The present manuscript is very demanding from a data analysis point of view. I am involved in EEG but not an expert for the analysis tools used here. Having had a look into Vidaurre et al. 2018 I – at least believe – that the concept of brain states is borrowed from Lehman et al.'s basic microstate idea. And as I understood the authors correctly, they assume such (micro)states re-occurring multiple times across a time series. If we understood the basic concept correctly, we should expect no superposition of brain states at one single time point, but instead distinct brain states for each data point. If this is correct, then the above-cited sentences are misleading.

The reviewer is right in that as an idealised model, the Markov chain formalises the hidden states to be discrete, however in practice, the states were estimated to be probabilistic. Please see the following descriptions of the model from the original paper (Vidaurre et al., 2018):

“As a general framework, the HMM assumes that a time series can be described using a hidden sequence of a finite number of states, such that, at each time point, only one state is active. **In practice, because the HMM is a probabilistic model, the inference process acknowledges uncertainty and assigns a probability of being active to each state at each time point. Effectively, this amounts to having a mixture of models (or states) explaining the data at each time point, where the mixture weights are the state probabilities.**”

Thanks to the reviewer’s feedback, we clarified our explanation of the Hidden in the updated version as follows:

SI page 11, line 267-270

The Hidden Markov Model (HMM) as a general framework assumes a hidden sequence of a finite number of states which drives the observed time series (Bishop, 2006). **In practice, the algorithm adopts a probabilistic model which infers the probability of each state being active at each time point (order = 0).**

Regarding the other point related to EEG microstates, we also thank the reviewer for giving us the opportunity to explain our perspective on the relationship between the EEG microstates and the hidden states that we used for this project. We are aware of the idea of the EEG microstates (quasi-stable topographical structure lasting transiently for ~100ms) that has been around in the EEG literature for decades. In fact, we indeed cited several major references discussing EEG microstates in our manuscript:

- D. Lehmann, R. D. Pascual-Marqui, C. Michel, EEG microstates. Scholarpedia 4, 7632 (2009).
- J. Britz, D. Van De Ville, C. M. Michel, BOLD correlates of EEG topography reveal rapid resting-state network dynamics. NeuroImage 52, 1162–1170 (2010).
- C. M. Michel, T. Koenig, EEG microstates as a tool for studying the temporal dynamics of whole-brain neuronal networks: A review. NeuroImage 180, 577–593 (2018).

Although we believe there is probably an intrinsic link between the EEG microstates and the hidden neural states that we found, we did not make an explicit connection between them, because (1) EEG microstates were introduced in the literature as

transient patterns of the scalp topography for the measure of Global Field Potential (GFP) (Lehmann et al., 2009; Michel & Koenig, 2018) and this is not the measure that the TDE-HMM is based on. The observation model used in our study is an autoregressive model, which inherently captures the oscillatory information (power and phase coherence among ROIs), thus being a much more sophisticated model than GFP (which essentially measures standard deviation over trials). (2) Although the term EEG microstates came up earlier in the literature, it might not be the origin of the idea of dynamic modelling. From our perspective, what lies at the core of this kind of analyses is to decompose the high-dimensional timeseries into certain spatiotemporal patterns. This idea might be attributed all the way back to unsupervised signal separation, hidden Markov modelling or dimension reduction from the field of signal processing. These different methods, as it seems to us, are just alternative ways to discover the transient spatiotemporal patterns that reoccur in the timeseries. Indeed, there has been a recent critique on EEG microstates, of which the main point is that the EEG microstates are not discrete, and the authors suggested that “EEG microstates are better conceptualized as spatially and temporally continuous, rather than discrete activations of neural populations.” (Mishra et al., 2020) This is another reason why we believe the method used in our study deserves credibility on its own.

Minor points

- Page 4: “... Therefore, this paradigm had a 2×2 factorial design of dominant and mixed

perception types for the BR and RPL conditions ... “

You may change this sentence to:

“... Therefore, this paradigm had a 2×2 factorial design of dominant (red and green percepts taken together) and mixed perception types for the BR and RPL conditions ... “

The suggested change has been implemented as below:

Therefore, this paradigm had a 2×2 factorial design of dominant (red and green percepts taken together) and mixed perception types for the BR and RPL conditions.

- Page 5 first paragraph: “... formeda...” => “... formed a ...” Page 5 line 11: “greenpercept” -> “green percept”

We thank the reviewer for their careful reading, we have changed accordingly.

- Page 6, Figure 1 line 7: “...can generate alternating percept...” -> “... can generate alternating percepts”

As above.

- Page 7: “... over- lapped ... “ => “... overlapped ... “

As above.

- Page 11, line 11: Should the reference not be of the paper instead of the URL?
-> Diego Vidaurre, Andrew J. Quinn, Adam P. Baker, David Dupret, Alvaro Tejero-Cantero and Mark W. Woolrich (2016) Spectrally resolved fast transient brain states in electrophysiological data. NeuroImage. Volume 126, Pages 81–95.
=> At least the reference should also be added! Page 18, line 7: Same section title as above Page 18, line 12: “onthe” -> “on the”
Page 22: fMRIactivation => fMRI ac

We are very sorry for the formatting problem, as this document was originally written in latex. The formatting problem was caused during the format conversion from the PDF to Word. We have corrected these problems in the newer version.

- There are some formatting problems with reference number 36.

As above.

- SM, Page 6, line 3: reference the mne toolbox should be with the paper not the URL: Alexandre Gramfort, Martin Luessi, Eric Larson, Denis A. Engemann, Daniel Strohmeier, Christian Brodbeck, Roman Goj, Mainak Jas, Teon Brooks, Lauri Parkkonen, and Matti S. Hämäläinen. MEG and EEG data analysis with MNE-Python. *Frontiers in Neuroscience*, 7(267):1–13, 2013. doi:10.3389/fnins.2013.00267.

We thank the reviewer for pointing it out. We have fixed it. See the change below:

SI page 7, line 155

It was conducted on the platform of MNE-python (version 0.23.4)⁶

(<https://mne.tools/stable/index.html>).

Reference:

6. Gramfort, A. *et al.* MEG and EEG data analysis with MNE-Python. *Front. Neurosci.* **7**, 267 (2013).

- SM, Page 10, line 4: reference should be paper not URL: Pedregosa, F., Varoquaux, G., Gramfort, A., Michel, V., Thirion, B., Grisel, O., Blondel, M., Prettenhofer, P., Weiss, R., Dubourg, V., Vanderplas, J., Passos, A., Cournapeau, D., Brucher, M., Perrot, M., & Duchesnay, E. (2011). Scikit-learn: Machine Learning in Python. *Journal of Machine Learning Research*, 12, 2825–2830.

We thank the reviewer for pointing this out. We have fixed it. See the change below:

SI page 13, line 310

The cross validation (CV) was conducted with the *Scikit-learn* toolbox¹² (<https://scikit-learn.org/stable/>) on Python.

Reference:

12. Pedregosa, F. *et al.* Scikit-learn: Machine Learning in Python. *J. Mach. Learn. Res.* **12**, (2012).

- SM, Page 11, line 6: the URL of the Matlab code sends me to the matlab homepage
SM, Page 12, line 4: URL doesn't work because there is a space in it

We thank the reviewer for their observation. We have fixed the aforementioned URL problems. See the change below:

SI page 14, line 336-338

The GC was calculated from the original signals of PCU and V1 for every trial/epoch, by using the *granger_cause_1* toolbox²¹⁻²² released from the MATLAB Central File Exchange (https://www.mathworks.com/matlabcentral/fileexchange/59390-granger_cause_1).

References:

21. Granger, C. W. J. Investigating Causal Relations by Econometric Models and Cross-spectral Methods. *Econometrica* **37**, 424–438 (1969).
22. Atukeren, E. The relationship between the F-test and the Schwarz criterion: Implications for Granger-causality tests. *Econ. Bull.* **30**, 494–499 (2010).

Reference list (from the reviewer):

- Abdallah D, Brooks JL. 2020. Response dependence of reversal-related ERP components in perception of ambiguous figures. *Psychophysiology*. 57.
- Blake R. 2001. A Primer on Binocular Rivalry, Including Current Controversies. *Brain Mind*. 2:5–38.
- Blake R, Logothetis NK. 2002. Visual competition. *Nat Rev Neurosci*. 3:13–21.
- Brascamp J, Sterzer P, Blake R, Knapen T. 2018. Multistable Perception and the Role of the Frontoparietal Cortex in Perceptual Inference. *Annu Rev Psychol*. 69:77–103.
- Britz J, Landis T, Michel CM. 2009. Right parietal brain activity precedes perceptual alternation of bistable stimuli. *Cereb Cortex*. 19:55–65.
- Ehm W, Bach M, Kornmeier J. 2011. Ambiguous figures and binding: EEG frequency modulations during multistable perception. *Psychophysiology*. 48:547–558.
- Intaite M, Koivisto M, Ruksenas O, Revonsuo A. 2010. Reversal negativity and bistable stimuli: Attention, awareness, or something else? *Brain Cogn*. 74:24–34.
- Kornmeier J, Bach M. 2012. Ambiguous figures – what happens in the brain when perception changes but not the stimulus. *Front Hum Neurosci*. 6:1–23.
- Kornmeier J, Friedel Evelyn, Hecker L, Schmidt S, Wittmann M. 2019. What happens in the brain of meditators when perception changes but not the stimulus? *PLOS ONE*. 14:e0223843.
- O'Donnell BF, Hendler T, Squires NK. 1988. Visual evoked potentials to illusory reversals of the Necker cube. *Psychophysiology*. 25:137–143.
- O'Shea RP, Kornmeier J, Roeber U. 2013. Predicting visual consciousness electrophysiologically from intermittent binocular rivalry. *PLoS ONE*. 8:e76134.

Pitts MA, Britz J. 2011. Insights from intermittent binocular rivalry and EEG. *Front Hum Neurosci.* 5:107.

Pitts MA, Neger JL, Davis TJR. 2007. Electrophysiological correlates of perceptual reversals for three different types of multistable images. *J Vis.* 7:1–14.

Sandberg K, Barnes GR, Bahrami B, Kanai R, Overgaard M, Rees G. 2014. Distinct MEG correlates of conscious experience, perceptual reversals and stabilization during binocular rivalry. *NeuroImage.* 100:161–175.

References for Reviewer 2:

Althouse, A. D. (2016). Adjust for Multiple Comparisons? It's Not That Simple. *The Annals of Thoracic Surgery*, 101(5), 1644–1645.
<https://doi.org/10.1016/j.athoracsur.2015.11.024>

Atukeren, E. (2010). The relationship between the F-test and the Schwarz criterion: Implications for Granger-causality tests. *Economics Bulletin*, 30(1), 494–499.

Bishop, C. (2006). Sequential Data. In *Pattern Recognition and Machine Learning* (pp. 605–646). Springer-Verlag. <https://www.springer.com/gp/book/9780387310732>

Blake, R., Brascamp, J., & Heeger, D. J. (2014). Can binocular rivalry reveal neural correlates of consciousness? *Philosophical Transactions of the Royal Society B: Biological Sciences*, 369(1641). <https://doi.org/10.1098/rstb.2013.0211>

Blake, R., & Logothetis, N. K. (2002). Visual competition. *Nature Reviews Neuroscience*, 3(1), 13–21. <https://doi.org/10.1038/nrn701>

Boldi, R. (2016). *Granger_Cause_1* [MATLAB Central File Exchange].
https://www.mathworks.com/matlabcentral/fileexchange/59390-granger_cause_1

Brascamp, J., Sterzer, P., Blake, R., & Knapen, T. (2018). Multistable Perception and the Role of the Frontoparietal Cortex in Perceptual Inference. *Annual Review of Psychology*, 69, 77–103. <https://doi.org/10.1146/annurev-psych-010417-085944>

Britz, J., Pitts, M. A., & Michel, C. M. (2010). Right parietal brain activity precedes perceptual alternation during binocular rivalry. *Human Brain Mapping*, 32(9), 1432–1442. <https://doi.org/10.1002/hbm.21117>

- Britz, J., Van De Ville, D., & Michel, C. M. (2010). BOLD correlates of EEG topography reveal rapid resting-state network dynamics. *NeuroImage*, *52*(4), 1162–1170. <https://doi.org/10.1016/j.neuroimage.2010.02.052>
- Callard, F., & Margulies, D. S. (2014). What we talk about when we talk about the default mode network. *Frontiers in Human Neuroscience*, *8*. <https://doi.org/10.3389/fnhum.2014.00619>
- Chong, S. C., Tadin, D., & Blake, R. (2005). Endogenous attention prolongs dominance durations in binocular rivalry. *Journal of Vision*, *5*(11), 6–6. <https://doi.org/10.1167/5.11.6>
- Cole, M. W., Bassett, D. S., Power, J. D., Braver, T. S., & Petersen, S. E. (2014). Intrinsic and task-evoked network architectures of the human brain. *Neuron*, *83*(1), 238–251. <https://doi.org/10.1016/j.neuron.2014.05.014>
- Cole, M. W., Ito, T., Bassett, D. S., & Schultz, D. H. (2016). Activity flow over resting-state networks shapes cognitive task activations. *Nature Neuroscience*, *19*(12), 1718–1726. <https://doi.org/10.1038/nn.4406>
- da Cruz, J. R., Favrod, O., Roinishvili, M., Chkonia, E., Brand, A., Mohr, C., Figueiredo, P., & Herzog, M. H. (2020). EEG microstates are a candidate endophenotype for schizophrenia. *Nature Communications*, *11*(1), 3089. <https://doi.org/10.1038/s41467-020-16914-1>
- Dattola, S., Morabito, F. C., Mammone, N., & La Foresta, F. (2020). Findings about LORETA Applied to High-Density EEG—A Review. *Electronics*, *9*(4), 660. <https://doi.org/10.3390/electronics9040660>
- Debecker, J., & Desmedt, J. E. (1970). Maximum capacity for sequential one-bit auditory decisions. *Journal of Experimental Psychology*, *83*(3, Pt.1), 366–372. <https://doi.org/10.1037/h0028848>
- Deco, G., & Kringelbach, M. L. (2017). Hierarchy of Information Processing in the Brain: A Novel ‘Intrinsic Ignition’ Framework. *Neuron*, *94*(5), 961–968. <https://doi.org/10.1016/j.neuron.2017.03.028>
- Friston, K., Harrison, L., Daunizeau, J., Kiebel, S., Phillips, C., Trujillo-Barreto, N., Henson, R., Flandin, G., & Mattout, J. (2008). Multiple sparse priors for the

- M/EEG inverse problem. *NeuroImage*, 39(3), 1104–1120.
<https://doi.org/10.1016/j.neuroimage.2007.09.048>
- Gramfort, A., Luessi, M., Larson, E., Engemann, D. A., Strohmeier, D., Brodbeck, C., Goj, R., Jas, M., Brooks, T., Parkkonen, L., & Hämäläinen, M. (2013). MEG and EEG data analysis with MNE-Python. *Frontiers in Neuroscience*, 7, 267.
<https://doi.org/10.3389/fnins.2013.00267>
- Granger, C. W. J. (1969). Investigating Causal Relations by Econometric Models and Cross-spectral Methods. *Econometrica*, 37(3), 424–438.
<https://doi.org/10.2307/1912791>
- Henson, R. N., Abdulrahman, H., Flandin, G., & Litvak, V. (2019). Multimodal Integration of M/EEG and fMRI Data in SPM12. *Frontiers in Neuroscience*, 13, 300.
<https://doi.org/10.3389/fnins.2019.00300>
- Henson, R. N., Flandin, G., Friston, K. J., & Mattout, J. (2010). A Parametric Empirical Bayesian framework for fMRI-constrained MEG/EEG source reconstruction. *Human Brain Mapping*, 31(10), 1512–1531. <https://doi.org/10.1002/hbm.20956>
- Hohwy, J., Roepstorff, A., & Friston, K. (2008). Predictive coding explains binocular rivalry: An epistemological review. *Cognition*, 108(3), 687–701.
<https://doi.org/10.1016/j.cognition.2008.05.010>
- Jafari, M., & Ansari-Pour, N. (2019). Why, When and How to Adjust Your P Values? *Cell Journal (Yakhteh)*, 20(4), 604–607. <https://doi.org/10.22074/cellj.2019.5992>
- Jamison, K. W., Roy, A. V., He, S., Engel, S. A., & He, B. (2015). SSVEP Signatures of Binocular Rivalry During Simultaneous EEG and fMRI. *Journal of Neuroscience Methods*, 243, 53–62. <https://doi.org/10.1016/j.jneumeth.2015.01.024>
- Knapen, T., Brascamp, J., Pearson, J., van Ee, R., & Blake, R. (2011). The role of frontal and parietal brain areas in bistable perception. *The Journal of Neuroscience: The Official Journal of the Society for Neuroscience*, 31(28), 10293–10301. <https://doi.org/10.1523/JNEUROSCI.1727-11.2011>
- Kornmeier, J., & Bach, M. (2006). Bistable perception—Along the processing chain from ambiguous visual input to a stable percept. *International Journal of Psychophysiology*, 62(2), 345–349.
<https://doi.org/10.1016/j.ijpsycho.2006.04.007>

- Kornmeier, J., & Bach, M. (2012). Ambiguous Figures – What Happens in the Brain When Perception Changes But Not the Stimulus. *Frontiers in Human Neuroscience*, 6, 51. <https://doi.org/10.3389/fnhum.2012.00051>
- Kozák, L. R., van Graan, L. A., Chaudhary, U. J., Szabó, Á. G., & Lemieux, L. (2017). ICN_Atlas: Automated description and quantification of functional MRI activation patterns in the framework of intrinsic connectivity networks. *Neuroimage*, 163, 319–341. <https://doi.org/10.1016/j.neuroimage.2017.09.014>
- Lee, S., & Lee, D. K. (2018). What is the proper way to apply the multiple comparison test? *Korean Journal of Anesthesiology*, 71(5), 353–360. <https://doi.org/10.4097/kja.d.18.00242>
- Lehmann, D., Pascual-Marqui, R. D., & Michel, C. (2009). EEG microstates. *Scholarpedia*, 4(3), 7632. <https://doi.org/10.4249/scholarpedia.7632>
- López, J. D., Litvak, V., Espinosa, J. J., Friston, K., & Barnes, G. R. (2014). Algorithmic procedures for Bayesian MEG/EEG source reconstruction in SPM. *NeuroImage*, 84, 476–487. <https://doi.org/10.1016/j.neuroimage.2013.09.002>
- Margulies, D. S., Ghosh, S. S., Goulas, A., Falkiewicz, M., Huntenburg, J. M., Langs, G., Bezgin, G., Eickhoff, S. B., Castellanos, F. X., Petrides, M., Jefferies, E., & Smallwood, J. (2016). Situating the default-mode network along a principal gradient of macroscale cortical organization. *Proceedings of the National Academy of Sciences*, 113(44), 12574–12579. <https://doi.org/10.1073/pnas.1608282113>
- Mashour, G. A., Roelfsema, P., Changeux, J.-P., & Dehaene, S. (2020). Conscious Processing and the Global Neuronal Workspace Hypothesis. *Neuron*, 105(5), 776–798. <https://doi.org/10.1016/j.neuron.2020.01.026>
- Mazaika, P. K., Glover, G. H., & Reiss, A. L. (2011). Rapid motions in pediatric and clinical populations. *Psychiatry*, 65(11), 1315–1323.
- Michel, C. M., & Brunet, D. (2019). EEG Source Imaging: A Practical Review of the Analysis Steps. *Frontiers in Neurology*, 10, 325. <https://doi.org/10.3389/fneur.2019.00325>

- Michel, C. M., & Koenig, T. (2018). EEG microstates as a tool for studying the temporal dynamics of whole-brain neuronal networks: A review. *NeuroImage*, *180*, 577–593. <https://doi.org/10.1016/j.neuroimage.2017.11.062>
- Mishra, A., Englitz, B., & Cohen, M. X. (2020). EEG microstates as a continuous phenomenon. *NeuroImage*, *208*, 116454. <https://doi.org/10.1016/j.neuroimage.2019.116454>
- Nichols, T. E. (2012). Multiple testing corrections, nonparametric methods, and random field theory. *NeuroImage*, *62*(2), 811–815. <https://doi.org/10.1016/j.neuroimage.2012.04.014>
- Paffen, C. L. E., & Alais, D. (2011). Attentional Modulation of Binocular Rivalry. *Frontiers in Human Neuroscience*, *5*. <https://doi.org/10.3389/fnhum.2011.00105>
- Paffen, C. L. E., Alais, D., & Verstraten, F. A. J. (2006). Attention speeds binocular rivalry. *Psychological Science*, *17*(9), 752–756. <https://doi.org/10.1111/j.1467-9280.2006.01777.x>
- Pedregosa, F., Varoquaux, G., Gramfort, A., Michel, V., Thirion, B., Grisel, O., Blondel, M., Prettenhofer, P., Weiss, R., Dubourg, V., Vanderplas, J., Passos, A., Cournapeau, D., Brucher, M., Perrot, M., Duchesnay, E., & Louppe, G. (2012). Scikit-learn: Machine Learning in Python. *Journal of Machine Learning Research*, *12*.
- Pernet, C. R., Latinus, M., Nichols, T. E., & Rousselet, G. A. (2015). Cluster-based computational methods for mass univariate analyses of event-related brain potentials/fields: A simulation study. *Journal of Neuroscience Methods*, *250*, 85–93. <https://doi.org/10.1016/j.jneumeth.2014.08.003>
- Pezzulo, G., Zorzi, M., & Corbetta, M. (2021). The secret life of predictive brains: What's spontaneous activity for? *Trends in Cognitive Sciences*, *25*(9), 730–743. <https://doi.org/10.1016/j.tics.2021.05.007>
- Quinn, A. J., Vidaurre, D., Abeyesuriya, R., Becker, R., Nobre, A. C., & Woolrich, M. W. (2018). Task-Evoked Dynamic Network Analysis Through Hidden Markov Modeling. *Frontiers in Neuroscience*, *12*. <https://doi.org/10.3389/fnins.2018.00603>

- Roy, A. V., Jamison, K. W., He, S., Engel, S. A., & He, B. (2017). Deactivation in the posterior mid-cingulate cortex reflects perceptual transitions during binocular rivalry: Evidence from simultaneous EEG-fMRI. *NeuroImage*, *152*, 1–11. <https://doi.org/10.1016/j.neuroimage.2017.02.041>
- Smallwood, J., Bernhardt, B. C., Leech, R., Bzdok, D., Jefferies, E., & Margulies, D. S. (2021). The default mode network in cognition: A topographical perspective. *Nature Reviews Neuroscience*, *22*(8), 503–513. <https://doi.org/10.1038/s41583-021-00474-4>
- Smith, S. M., Fox, P. T., Miller, K. L., Glahn, D. C., Fox, P. M., Mackay, C. E., Filippini, N., Watkins, K. E., Toro, R., Laird, A. R., & Beckmann, C. F. (2009). Correspondence of the brain's functional architecture during activation and rest. *Proceedings of the National Academy of Sciences of the United States of America*, *106*(31), 13040–13045. <https://doi.org/10.1073/pnas.0905267106>
- Sprengh, R. N. (2012). The Fallacy of a “Task-Negative” Network. *Frontiers in Psychology*, *3*. <https://doi.org/10.3389/fpsyg.2012.00145>
- Theodoni, P., Panagiotaropoulos, T. I., Kapoor, V., Logothetis, N. K., & Deco, G. (2011). Cortical Microcircuit Dynamics Mediating Binocular Rivalry: The Role of Adaptation in Inhibition. *Frontiers in Human Neuroscience*, *5*. <https://doi.org/10.3389/fnhum.2011.00145>
- Tong, F., Meng, M., & Blake, R. (2006). Neural bases of binocular rivalry. *Trends in Cognitive Sciences*, *10*(11), 502–511. <https://doi.org/10.1016/j.tics.2006.09.003>
- Tsuchiya, N., & Koch, C. (2008). Attention and consciousness. *Scholarpedia*, *3*(5), 4173. <https://doi.org/10.4249/scholarpedia.4173>
- Vidaurre, D., Hunt, L. T., Quinn, A. J., Hunt, B. A. E., Brookes, M. J., Nobre, A. C., & Woolrich, M. W. (2018). Spontaneous cortical activity transiently organises into frequency specific phase-coupling networks. *Nature Communications*, *9*(1), 2987. <https://doi.org/10.1038/s41467-018-05316-z>
- Watanabe, M., Cheng, K., Murayama, Y., Ueno, K., Asamizuya, T., Tanaka, K., & Logothetis, N. (2011). Attention But Not Awareness Modulates the BOLD Signal in the Human V1 During Binocular Suppression. *Science*, *334*(6057), 829–831. <https://doi.org/10.1126/science.1203161>

Zhang, P., Jamison, K., Engel, S., He, B., & He, S. (2011). Binocular Rivalry Requires Visual Attention. *Neuron*, 71(2), 362–369.
<https://doi.org/10.1016/j.neuron.2011.05.035>

Reviewer #3 (Remarks to the Author):

(Author's note: the reviewer has combined points in one paragraph. We manually separated these points and highlighted them with bold texts.)

This study re-analyzed a published EEG-fMRI dataset on binocular rivalry. Based on EEG-informed analysis, the previous study revealed suppressed BOLD activities in posterior region of the default model network (DMN) associated with perceptual transitions. The current study used more sophisticated analysis approaches on the EEG data and suggest that the precuneus (PCU), a key node of DMN, plays a causal role for perceptual switch. There are quite some analysis and results, and rather complicated. I brief them as follows:

(1) They first found reduced global field power (GFP) in BR vs. RPL condition at about 300ms before the button report of dominant percept. Source localization based on the prior of fMRI activation revealed deactivation of a range of brain areas including some of the DMN nodes. **However, the fMRI results didn't reveal any deactivations.**

We apologise for not communicating our results well. We reported deactivations in the original manuscript (Page 6, Line 166); however, what we showed in the original Fig. 2. was the result of the F-test (two-tailed). To avoid confusion, we created new figures to present both T-test contrasts and the F-test global effect in the revised manuscript (see below new Fig. 2).

The reason that we reported F-test results is because we did not only rely on activation to infer regional engagement during the task. Focusing on activation or deactivation alone would have been ambiguous for interpretation in our case because (1) the deactivation of DMN is well-known for task vs rest but the cognitive implication of this is still unknown. It has been suggested that the deactivation of DMN during tasks serves a function, as the lack of this has been associated with brain disorders and cognitive failures (Callard & Margulies, 2014; Spreng, 2012). (2) During the bi-stable task, the perceptual reversal is continuous. Both up and down deflections of neural activity are necessary components of this dynamical process. However, to avoid confusion, we reported the paired T tests in addition to the F-test in the updated version of the Figure 2. Please see the change as below:

Manuscript Page 7, Line 158-177:

Fig. 2. FMRI activation revealing the DMN's involvement in the current task. Subplots (a), (b) and (c) show the significant clusters [$P_{\text{voxel}} < 0.001$ (uncorrected) & $P_{\text{cluster}} < 0.05$ (family-wise error corrected)] respectively for the contrasts "BR (dominant) - RPL (dominant)", "BR (mixed) - RPL (mixed)" and the interaction effect between perceptual generating settings (BR vs. RPL) and perceptual contents (dominant vs. mixed). The interaction analysis (c) shows the regions mostly sensitive to the condition differences. For statistical testing, we adopted linear mixed-effects models, where the paired T tests and F test were carried out at the individual level, while statistical inferences were made at the population level with group-level one-sample T tests (input being the individual-level estimators). Hence, the colour bars of the subplots (a)-(c) indicate the T scores from the group-level testing. The circular plot in (c) shows the ICN affiliation of the significant clusters for the interaction effect. "ICN involvement" is a measure of correspondence between an activation map and large-scale networks with well-established

cognitive function, as provided by the BrainMap (BM) meta-analysis database³⁹. The BM number on the circular plot indicates the number of the ICN-BM network atlas. Cognitive domains of the networks that have negligible involvement in this task are not listed in the figure. These are BM1: Limbic and medial-temporal areas; BM3: Bilateral BG and thalamus; BM4: Bilateral anterior insula/frontal opercula and anterior cingulate gyrus; BM5: Midbrain; Cerebellum; BM6: Superior and middle frontal gyri; Sensorimotor; BM9: Superior parietal lobule; Frontoparietal (perception-somesthesia-pain); BM10: Middle and inferior temporal gyri; Frontoparietal (cognition-language); BM11: Lateral posterior occipital cortex; BM14: Cerebellum; BM16: Transverse temporal gyri; BM17: Dorsal precentral gyri, central sulci, postcentral gyri, superior and inferior cerebellum; BM19: Artefactual component; BM20: Artefactual component.

(2) Then they used a four states Hidden Markov Modeling (HMM) algorithm to fit the time course of EEG sources from DMN regions and V1. The occupancy of state 4 was found much longer for the rivalry than the replay condition. **Although there were also obvious differences in other states, they used state 4 for the following analysis.** Then a spectrum analysis was performed on power and coherence of different frequency bands between different regions from different states. **Only alpha band result was used the following analysis.** For the alpha coherence, state 4 shows significant PCU-V1 phase coherence. **The PCU-rIPL coherence was also significant, but it was not mentioned in the text.** Then they selected the PCU-V1 coherence for granger causality analysis and found no correlation of GC->V1 with perceptual duration at the trial-by-trial level. But there was a correlation at the group level only when the trial was dominated by state 4. Then they came to the conclusion that the causal effect from precuneus (PCU) to V1 can predict the following duration of perceptual state. **These are some interesting results from these analyses, but the main conclusion comes from a long way with many steps of arbitrary selections. Thus I feel the evidence for the final conclusion are relatively weak.**

We appreciate the reviewer's concise summary, and their criticism about our selective follow-up analyses of the results. However, we would like to clarify that the final step of the follow-up analysis was based on distinct features of the data, rather than arbitrary selections. In addition, we would like to point out that the feature selection does not change our main conclusion. We will expand our response as follows:

The reviewer mentioned that "*Only alpha band result was used the following analysis.*" However, this is a misunderstanding: each state encompasses the full spectrum of power and phase coherence. In fact, we used all **States' FO/lifetime** for subsequent analyses (as described in "Endogenous neural states underlie upcoming perceptual transitions" and "State spectral information reveals PCU-V1 coherence"). The misconception might arise from what is shown in Fig. 5, where we only presented the alpha band result. This reduced presentation is necessary because the resulting states are multi-dimensional. It is a common practice to selectively present results when they

are multidimensional; for example, only the first and second dimensions are usually presented for the PCA analyses. In our case, we have at least 6 dimensions to present (states, conditions, frequency bands, regions, power and phase coherence). The limitation of the journal format makes it difficult to present every aspect of the result in the main text. We chose the alpha band results to present in the main text, because it demonstrates an interesting pattern which is more interpretable, and more relatable to the previous literature (see our reasoning in Page 16, Line 381-393).

In the last result section of the main text, we focused on State 4 for further investigation. This is because only State 4 is correlated with the behavioural measure, as we reported in Fig. 6a, where we can see relationships with behaviour for all states. In addition, from the previous analysis on state FO vs. condition correspondence (Fig. 5a,b), we could also see that State 4 was visited singularly the most in the BR (dominant) condition, i.e. the condition of interest. Within State 4, PCU is the only region whose phase coherence with V1 (alpha band) was shown significantly modulated by the task, that's why we further investigated the PCU → V1 relationship. Specifically, the coupling between PCU and V1 was significantly increased in State 4 (which characterises the BR [dominant]) and significantly decreased in State 2 (characterising the RPL [dominant]); while such difference was not seen for the PCU-rIPL connectivity. Apart from the discovered features based on this current dataset, PCU is also well known for its association with visuospatial integration (Cavanna, 2007; Cavanna & Trimble, 2006).

Given the aforementioned reasons, we hope the reviewer can be convinced that selective presentation of multi-dimensional dataset as such are necessary and our approach is justified. Please see supplementary materials for a full set of the result.

As to the point that the reviewer made that “*the main conclusion comes from a long way with many steps of arbitrary selections.*”, we would like to emphasise that, even without the last part of the analyses, our main conclusion still holds. In the paper, we concluded that the pre-onset neural activities within the DMN can predict the upcoming perceptual transition. Before we zoomed into the featured state (State 4), we showed that the extracted DMN and V1 signals were characterised by four hidden states (each with a full-spectral unfolding of the regional power and inter-regional coherence), which can be used to predict the upcoming perceptual transitions (according to the multiple linear regression and the cross validation). By providing the additional analysis step (i.e. the cognitive relevance of the top-down connectivity between PCU and V1), we attempted to make the conclusion more convincing.

Finally, we thank the reviewer for pointing out the rIPL and PCU coherence which was also significant in State 4. We have added relevant contents in the updated version:

Manuscript Page 13, Line 300-304:

At the alpha band, State 4 was characterised by an increased phase coherence between PCU and V1, and between PCU and rIPL, and State 1 was characterised by an overall increased power in the DMN and V1 regions, accompanied by a general decreased phase coherence among the posterior DMN nodes (PCC, bilateral HP and IPL). These two states were often present during

the BR (dominant) trials [FO = 0.62 +/- 0.27, 0.29 +/- 0.22 (mean +/- standard deviation), respectively for State 4 and 1].

Manuscript Page 17, Line 410-413:

SPL is not investigated in our study, but parts of our results hinted at its relevance. Our critical state, State 4, was characterised not only by an increased coupling between PCU and V1, but also that between PCU and rIPL. Although the IPL is usually considered to be part of the DMN, it is anatomically adjacent and functionally coupled to the SPL which is a core of the posterior attentional control system^{53,54}.

(3) **Another major concern is for the definition of time of perceptual switch.** The current study used the button press of perceptual report for the time of switch, while the human reaction time on average is about 400ms. The deactivation they found on the EEG signal (about 300ms before button press) may occur after the perceptual switch. Thus the DMN activity they found associated with this time period should be a consequence rather than the cause of conscious perception. I suggest the author to use the frequency tagging from the EEG signals to decode the “real” perceptual switch from the visual cortex.

We appreciate the reviewer’s concerns regarding the temporal precision of the reversal timing, and we apologise for this misunderstanding. In the revised manuscript, we have made sure that we did not claim that the evoked-response result indicates pre-perceptual neural cause (Manuscript page 8, line 227-230), and we further clarified the point that the HMM-TDE is independent of the evoked-response analysis (Supplementary Materials Page 9-10, Line 288-319). To clarify the reviewer’s concern, the state we discovered was not limited to the [-400 -200] ms time window. We used the traditional ERP analysis firstly as a sanity check of the data, which also established correspondence with the previous literature; and secondly to provide a lower-bound to the window selection for the subsequent TDE-HMM analysis. Please see the revised text with more detailed explanations of these two points:

Manuscript page 8, line 186-189

Though based on response-locked analyses, this result is comparable with previous literature dominated by stimulus-locked analyses, which suggests that the P300-like ERP component from parietal activity (roughly happening 300 ms before a key press) probably indicates a conscious recognition of a perceptual disambiguation⁴⁰.

SI Page 9-10, Line 220-244

Selection of time window and regions of interest for the TDE-HMM dynamic process modelling

Instead of modelling the dynamic process for the whole brain during the whole experiment, we narrowed down the time window and regions of interest (ROIs), in order

to increase the method's sensitivity and avoid overfitting¹⁰. Our previous evoked-response analysis helped us to narrow down the time of interest to a window between [-400 -200] ms before a subjective report of perceptual change, which is when the pre-response ERPs showed the biggest GFP difference between BR and RPL. However, the endogenous neural activity triggering an upcoming transition might take effect even earlier than that¹¹. Therefore, we modelled the dynamic process in the 1-second (with 100-ms post-response padding) window before every report of a perceptual change. The 1-s window was chosen as a trade-off between having enough sample points and having enough specificity to the targeted events. It has been suggested in the literature that endogenous perceptual disambiguation is associated with posterior parietal activity 50 ms before the onset of a bistable stimulus¹²; and the upper limit of reaction time to it is about 600 ms^{11,13}, while the upper limit of pure motor execution is about 150 ms^{11,14}. Therefore, the 1-s window should be able to cover the whole dynamic neural process towards a perceptual transition and a little further before. 25767 trials were generated in total, considering all 4 conditions: BR/RPL (dominant) (i.e., transitions from a mixed to a dominant percept in the BR or RPL setting; n=7795/7148), and BR/RPL (mixed) (i.e., transitions from a dominant to a mixed percept in the BR or RPL setting; n=5786/5038). As the algorithm computes the latent states recurring across all timepoints of all trials, the state delineation is assumed to be driven by the endogenous neural states which have a robust spatiotemporal structure recurring through time^{10,15-18}.

Regarding the reviewer's suggestion of using an alternative method to identify the precise timing of perceptual transition: There has been effort in the literature using M/EEG frequency tagging, multivariate pattern decoding, or pupillometry to improve the temporal precision of the perceptual events, however, the decoding of the neural or the physiological features is still open to interpretation (Brascamp et al., 2018; Jamison et al., 2015; Sandberg et al., 2014). We consider perception to be a continuous process rather than discrete moments, that is why we applied the dynamic process modelling. We agree with the reviewer that the 1-s time window before a motor response contains post-perceptual events. However, the post-perceptual events cannot discriminate between the RPL and BR conditions as the states can do (see SI figure 6, 7 and manuscript Fig. 5). In addition, the TDE-HMM algorithm is supposed to discover the latent states that persist over all given timepoints and trials. What drives the clustering

of states is unlikely to be the fast reversal event (~50-70ms), which is short (10% of all timepoints) and supposedly only happens in BR (dominant) trials. According to previous literature, state detection is driven by the endogenous neural states which supposedly have a relatively consistent and robust spatiotemporal structure (Cole et al., 2014, 2016; Deco & Jirsa, 2012; Theodoni et al., 2011; Vidaurre et al., 2018; Wang et al., 2019).

References for Reviewer 3:

- Brascamp, J., Sterzer, P., Blake, R., & Knapen, T. (2018). Multistable Perception and the Role of the Frontoparietal Cortex in Perceptual Inference. *Annual Review of Psychology*, 69, 77–103. <https://doi.org/10.1146/annurev-psych-010417-085944>
- Britz, J., Landis, T., & Michel, C. M. (2009). Right parietal brain activity precedes perceptual alternation of bistable stimuli. *Cerebral Cortex (New York, N. Y.: 1991)*, 19(1), 55–65. <https://doi.org/10.1093/cercor/bhn056>
- Britz, J., Pitts, M. A., & Michel, C. M. (2010). Right parietal brain activity precedes perceptual alternation during binocular rivalry. *Human Brain Mapping*, 32(9), 1432–1442. <https://doi.org/10.1002/hbm.21117>
- Callard, F., & Margulies, D. S. (2014). What we talk about when we talk about the default mode network. *Frontiers in Human Neuroscience*, 8. <https://doi.org/10.3389/fnhum.2014.00619>
- Carmel, D., Walsh, V., Lavie, N., & Rees, G. (2010). Right parietal TMS shortens dominance durations in binocular rivalry. *Current Biology*, 20(18), R799–R800. <https://doi.org/10.1016/j.cub.2010.07.036>
- Cavanna, A. E. (2007). The Precuneus and Consciousness. *CNS Spectrums*, 12(7), 545–552. <https://doi.org/10.1017/S1092852900021295>
- Cavanna, A. E., & Trimble, M. R. (2006). The precuneus: A review of its functional anatomy and behavioural correlates. *Brain*, 129(3), 564–583. <https://doi.org/10.1093/brain/awl004>
- Cole, M. W., Bassett, D. S., Power, J. D., Braver, T. S., & Petersen, S. E. (2014). Intrinsic and task-evoked network architectures of the human brain. *Neuron*, 83(1), 238–251. <https://doi.org/10.1016/j.neuron.2014.05.014>

- Cole, M. W., Ito, T., Bassett, D. S., & Schultz, D. H. (2016). Activity flow over resting-state networks shapes cognitive task activations. *Nature Neuroscience*, *19*(12), 1718–1726. <https://doi.org/10.1038/nn.4406>
- Debecker, J., & Desmedt, J. E. (1970). Maximum capacity for sequential one-bit auditory decisions. *Journal of Experimental Psychology*, *83*(3, Pt.1), 366–372. <https://doi.org/10.1037/h0028848>
- Deco, G., & Jirsa, V. K. (2012). Ongoing Cortical Activity at Rest: Criticality, Multistability, and Ghost Attractors. *Journal of Neuroscience*, *32*(10), 3366–3375. <https://doi.org/10.1523/JNEUROSCI.2523-11.2012>
- He, B. J. (2018). Robust, Transient Neural Dynamics during Conscious Perception. *Trends in Cognitive Sciences*, *22*(7), 563–565. <https://doi.org/10.1016/j.tics.2018.04.005>
- Igelström, K. M., & Graziano, M. S. A. (2017). The inferior parietal lobule and temporoparietal junction: A network perspective. *Neuropsychologia*, *105*, 70–83. <https://doi.org/10.1016/j.neuropsychologia.2017.01.001>
- Jamison, K. W., Roy, A. V., He, S., Engel, S. A., & He, B. (2015). SSVEP Signatures of Binocular Rivalry During Simultaneous EEG and fMRI. *Journal of Neuroscience Methods*, *243*, 53–62. <https://doi.org/10.1016/j.jneumeth.2015.01.024>
- Jensen, O., & Mazaheri, A. (2010). Shaping Functional Architecture by Oscillatory Alpha Activity: Gating by Inhibition. *Frontiers in Human Neuroscience*, *4*. <https://doi.org/10.3389/fnhum.2010.00186>
- Kanai, R., Bahrami, B., & Rees, G. (2010). Human Parietal Cortex Structure Predicts Individual Differences in Perceptual Rivalry. *Current Biology*, *20*(18), 1626–1630. <https://doi.org/10.1016/j.cub.2010.07.027>
- Koivisto, M., & Revonsuo, A. (2010). Event-related brain potential correlates of visual awareness. *Neuroscience & Biobehavioral Reviews*, *34*(6), 922–934. <https://doi.org/10.1016/j.neubiorev.2009.12.002>
- Kornmeier, J., & Bach, M. (2006). Bistable perception—Along the processing chain from ambiguous visual input to a stable percept. *International Journal of Psychophysiology*, *62*(2), 345–349. <https://doi.org/10.1016/j.ijpsycho.2006.04.007>

- Kornmeier, J., & Bach, M. (2012). Ambiguous Figures – What Happens in the Brain When Perception Changes But Not the Stimulus. *Frontiers in Human Neuroscience*, 6, 51. <https://doi.org/10.3389/fnhum.2012.00051>
- Kozák, L. R., van Graan, L. A., Chaudhary, U. J., Szabó, Á. G., & Lemieux, L. (2017). ICN_Atlas: Automated description and quantification of functional MRI activation patterns in the framework of intrinsic connectivity networks. *Neuroimage*, 163, 319–341. <https://doi.org/10.1016/j.neuroimage.2017.09.014>
- Margulies, D. S., Ghosh, S. S., Goulas, A., Falkiewicz, M., Huntenburg, J. M., Langs, G., Bezgin, G., Eickhoff, S. B., Castellanos, F. X., Petrides, M., Jefferies, E., & Smallwood, J. (2016). Situating the default-mode network along a principal gradient of macroscale cortical organization. *Proceedings of the National Academy of Sciences*, 113(44), 12574–12579. <https://doi.org/10.1073/pnas.1608282113>
- Mashour, G. A., Roelfsema, P., Changeux, J.-P., & Dehaene, S. (2020). Conscious Processing and the Global Neuronal Workspace Hypothesis. *Neuron*, 105(5), 776–798. <https://doi.org/10.1016/j.neuron.2020.01.026>
- Sandberg, K., Barnes, G. R., Bahrami, B., Kanai, R., Overgaard, M., & Rees, G. (2014). Distinct MEG correlates of conscious experience, perceptual reversals and stabilization during binocular rivalry. *Neuroimage*, 100(100), 161–175. <https://doi.org/10.1016/j.neuroimage.2014.06.023>
- Seghier, M. L. (2013). The angular gyrus: Multiple functions and multiple subdivisions. *The Neuroscientist: A Review Journal Bringing Neurobiology, Neurology and Psychiatry*, 19(1), 43–61. <https://doi.org/10.1177/1073858412440596>
- Spreng, R. N. (2012). The Fallacy of a “Task-Negative” Network. *Frontiers in Psychology*, 3. <https://doi.org/10.3389/fpsyg.2012.00145>
- Theodoni, P., Panagiotaropoulos, T. I., Kapoor, V., Logothetis, N. K., & Deco, G. (2011). Cortical Microcircuit Dynamics Mediating Binocular Rivalry: The Role of Adaptation in Inhibition. *Frontiers in Human Neuroscience*, 5. <https://doi.org/10.3389/fnhum.2011.00145>
- Vidaurre, D., Hunt, L. T., Quinn, A. J., Hunt, B. A. E., Brookes, M. J., Nobre, A. C., & Woolrich, M. W. (2018). Spontaneous cortical activity transiently organises into

frequency specific phase-coupling networks. *Nature Communications*, 9(1), 2987. <https://doi.org/10.1038/s41467-018-05316-z>

Vidaurre, D., Quinn, A. J., Baker, A. P., Dupret, D., Tejero-Cantero, A., & Woolrich, M. W. (2016). Spectrally resolved fast transient brain states in electrophysiological data. *NeuroImage*, 126, 81–95. <https://doi.org/10.1016/j.neuroimage.2015.11.047>

Wang, P., Kong, R., Kong, X., Liégeois, R., Orban, C., Deco, G., Heuvel, M. P. van den, & Yeo, B. T. T. (2019). Inversion of a large-scale circuit model reveals a cortical hierarchy in the dynamic resting human brain. *Science Advances*, 5(1), eaat7854. <https://doi.org/10.1126/sciadv.aat7854>

Wilcke, J. C., O'Shea, R. P., & Watts, R. (2009). Frontoparietal activity and its structural connectivity in binocular rivalry. *Brain Research*, 1305, 96–107. <https://doi.org/10.1016/j.brainres.2009.09.080>

Reviewer #4 (Remarks to the Author):

In their study the authors aim at analyzing the impact of intrinsic brain activity on behavior by using a paradigm of binocular rivalry. The topic itself is of high interest. The authors assemble a set of literature which underlines the relevance of their question and use advanced methods in the analysis pipeline.

My fundamental concern is that the time window in focus does not capture intrinsic brain activity before the percept, but during perception and therefore cannot be labelled predictive. While the results may still be interesting they cannot be interpreted in the way presented in the manuscript. In detail, the authors used the time point of button press response as the time of perception and neglected the reaction times, which typically is a few 100 ms depending on the instruction to the subjects. Most results rely on a time window of 1 sec before the button press response and therefore most likely contain a mixture of pre-percept signal and the sensation of perception. During the 1 sec period before the button press subjects experienced very different conditions, the (actual) change to a dominant color (RPL dominant), the change to a color, which they knew would not last (RPL mixed), the emergence of an ocular dominance (BR dominant) or a state of perceptive uncertainty (BR mixed). Therefore the robust splitting into 4 states seems plausible. The interpretation here needs to be refocused. The button press being a few hundred milliseconds after the percept change is also inline with the strong difference in global field power 400 ms to 300 ms before button press (Fig 3a).

We appreciate the reviewer's concern, which is that the neural states discovered in this study are about perceptual-related differences between conditions thus cannot be interpreted as the endogenous activities preceding a perceptual transition.

We understand the temporal imprecision of using response-locked trials is the major limitation of this study. However, this limitation exists because (1) the original dataset did not provide the stimulus onset in the RPL condition; (2) theoretically it is questionable whether perception happens in discrete moments or evolves as a gradual process (He, 2018; Koivisto & Revonsuo, 2010; Mashour et al., 2020). And we agree with the reviewer that the evoked-response GFP component ([-400 -200] ms) does not necessarily reflect the pre-perceptual differences between the BR and RPL. In the revised manuscript, we have avoided making any such claim:

Manuscript page 8, line 186-189:

Though based on response-locked analyses, this result is comparable with previous literature dominated by stimulus-locked analyses, which suggests that the P300-like ERP component from parietal activity (roughly happening 300 ms before a key press) probably indicates a conscious recognition of a perceptual disambiguation⁴⁰.

SI page 7, line 161-167:

We acknowledge the limitation of the response-locked analyses (as constrained by the experimental design), as the epochs incorporate a mixture of perceptual and motor processes. Admitting RPL is valid control condition in this experiment, any contrasted differences should be about the features that are different between the BR and RPL conditions, rather than the motor-related features or noises that are assumed to be common to both conditions.

However, we would like to point out that the pre-perceptual endogenous states were discovered by the TDE-HMM, which is independent of the evoked response analysis. Despite the limitation of using manual response as a landmark, based on previous literature and our result validation, we are relatively confident that the states were not driven by pure perceptual and motor confounds and we will expand on this below.

Firstly, the TDE-HMM algorithm we adopted to uncover the endogenous dynamic process, does not depend on the temporal precision of the reversal moment. The algorithm is supposed to discover the latent states that persist in all given timepoints and trials. What drives the clustering of states is unlikely to be the fast reversal event (~50-70ms), which is short (10% of all timepoints) and supposedly only happens in BR (dominant) trials, but rather by the endogenous neural states which have a relatively consistent and robust spatiotemporal structure (Cole et al., 2014, 2016; Deco & Jirsa, 2012; Theodoni et al., 2011; Vidaurre et al., 2018; Wang et al., 2019).

Secondly, BR and RPL are perceptually matched conditions, which is a foundation of such experimental paradigm. If the states represent post-perceptual processing alone, we would not have found opposite state characteristics between BR (dominant) and RPL (dominant) (Fig. 5).

Thirdly, we would like to ask the reviewer to consider the HMM result as a support for the notion that the neural states we found actually reflect endogenous neural activities. This can be reflected in our follow-up cross validation results in the section: Hidden neural states within the DMN underlie the upcoming perceptual transitions (page 11, Line 431-470). By knowing the states' FO in the 1 second before a BR transition, we have 91.38% (SD = 7.98%) predictability of the percept type for the upcoming transition; but the accuracy dropped significantly ($t = 2.92$, $p = 0.006$) to 82.40% (SD = 10.77%) for the RPL condition. This could be the case because the perceptual transition in BR is generated endogenously, and the states have captured the pre-onset endogenous neural activities that caused the upcoming perceptual changes. Please also see the evidence of the State 4 time course in the sFig. 7. State 4 captures the neural features temporally leading to the BR (dominant) transition, and a timecourse of its probabilistic presence over trials showed a higher frequency of its presence around [-0.8 -0.2] s, which preceded the window of [-0.4 -0.2] s that we identified from the evoked-response results. Please see relevant report in more details in the manuscript.

Please also see the SI figure 6 and 7:

sFig. 6. Cross validation (CV) score improving and converging across a range of regularisation parameters, respectively for the data in the BR condition (a) and RPL condition (b). The CV score is the accuracy of using hidden states' FO to predict the type (mixed or dominant) of the upcoming transition in each trial. The middle line indicates the mean of accuracy across all shuffled runs. The two dashed lines surrounding the mean mark the range of one standard deviation. The grey dashed line parallel to the x-axis indicates the maximum accuracy the model has reached. (c) Consecutive changes of the CV scores as a function of regularisation parameters. This was used to identify the optimal range of the regularisation parameters, which was marked with the red dashed line. A two-sampled t-test was conducted using the CV scores within the optimal range of the parameter C, for comparing the states' predictability between the BR and RPL conditions.

sFig. 7. The frequency (or proportion) cross trials of the State 4 being present at each time point in the BR (dominant) condition.

Please also see our justification of the time window selection in the revised text:

SI Page 9-10, Line 220-244

Selection of time window and regions of interest for the TDE-HMM dynamic process modelling

Instead of modelling the dynamic process for the whole brain during the whole experiment, we narrowed down the time window and regions of interest (ROIs), in order to increase the method's sensitivity and avoid overfitting¹⁰. Our previous evoked-response analysis helped us to narrow down the time of interest to a window between [-400 -200] ms before a subjective report of perceptual change, which is when the pre-response ERPs showed the biggest GFP difference between BR and RPL. However, the endogenous neural activity triggering an upcoming transition might take effect even earlier than that¹¹. Therefore, we modelled the dynamic process in the 1-second (with 100-ms post-response padding) window before every report of a perceptual change. The 1-s window was chosen as a trade-off between having enough sample points and having enough specificity to the targeted events. It has been suggested in the literature that endogenous perceptual disambiguation is associated with posterior parietal activity 50 ms before the onset of a bistable stimulus¹²; and the upper limit of reaction time to it is about 600 ms^{11,13}, while the upper limit of pure motor execution is about 150 ms^{11,14}. Therefore, the 1-s window should be able to cover the whole dynamic neural process towards a perceptual transition and a little further before. 25767 trials were generated in total, considering all 4 conditions: BR/RPL (dominant) (i.e., transitions from a mixed to a dominant percept in the BR or RPL setting; n=7795/7148), and BR/RPL (mixed) (i.e., transitions from a dominant to a mixed percept in the BR or RPL setting; n=5786/5038). As the algorithm computes the latent states recurring across all timepoints of all trials, the state delineation is assumed to be driven by the endogenous neural states which have a robust spatiotemporal structure recurring through time^{10,15-18}.

Suggested additions:

- One could give an estimate of the subjects' reaction times from conditions 'RPL dominant' and 'RPL mixed'.

We thank the reviewer for this good suggestion. In fact, we also thought of this but there is not enough information in the dataset (such as when the images were alternated in the RPL condition) for us to determine the individual's pure reaction time. We have also contacted the first author of the most recent paper using this dataset (Roy et al., 2017), but he told us that has does not have additional information than available online. We admit this is an inherently limitation for this dataset, but we wanted to reiterate that the dynamic state analysis did not rely on the temporal precision of the events.

- It would be interesting to analyze the transitions through states 1 to 4 across the time interval. Maybe there is a higher prevalence of e.g. state 4 in BR dominant (or mixed) at a certain point in time. If this time point would precede the putative time of percept switch (as estimated from the (individual) reaction times) this would be an interesting information.

We thank the reviewer for this great suggestion. Below we plotted the frequency of State 4's presence at each time point of a trial (BR dominant) (sFig. 7). Apart from the beginning of the trial which likely shows an edging effect, we can see higher prevalence of the state during [-0.7 -0.3] s before a motor response, which we believe is relevant to the endogenous processing prior to a perceptual change. In the paper we did not try to qualify the state's temporal evolution because it may not be sensible to locate the exact onset and offset of the endogenous process (as it might be a gradually accumulating process). Therefore, we took the whole window as being interesting and quantified the accumulated state's lifetime (or averaged FO in a probabilistic form) during this window. In the revision, we also presented this plot in the supplementary material (sFig. 7).

sFig. 7. The frequency (or proportion) cross trials of the State 4 being present at each time point in the BR (dominant) condition.

- There should be a way of quantifying direction of phase difference or causality from the information on the different states themselves. Maybe this is depicted in sFig. 13, and would be of interest, e.g. to clarify if GC (PCU → V1) is a hallmark of state 4.

We appreciate the reviewer's interest in causality modelling; however, causality was not a main focus in our study. In the previous result section: "*Endogenous neural states underlie upcoming perceptual transitions*", we already showed that the endogenous DMN activity can drive the upcoming perceptual change, by using the 4 discovered dynamic states as predictors. The DMN regions' activity can be interpreted as driving the perceptual changes without a causal modelling, because the external stimuli did not change in the BR condition. Hence, the endogenous brain activity is the only source of variance that the perceptual fluctuations are contingent on.

The reviewer made an astute observation on figure sFig. 13. Indeed, the lag information can be identified from sFig. 13, where we can see that [in the BR (dominant) condition] the PCU leads V1 in the delta range, while V1 leads the PCU in the alpha range. It revealed a rather complex pattern such that the relationship between the PCU and V1 seems to be interactive. However, that the PCU and V1 form a causal loop does not invalidate our result that the top-down connection from PCU → V1 is perceptually relevant. We would also like to kindly point out that the GC is calculated as an averaged lagged correlated coefficient across the whole time window, while the states were characterised and estimated on each time point (thus being a dynamic measure). To estimate the dynamic causality within the transient states, it is not impossible (i.e., can be achieved by partial directed coherence), but was not recommended by the toolbox developers, due to technical limitations (such as insufficient timepoints for a robust estimation of causal relationships within transient states).

Minor:

- The overall description of the succession of analyses is quite complicated. It would probably benefit from focusing on the EEG analysis strategies right away. The importance of the concurrent fMRI is not obvious. The overlap of EEG and fMRI localizations stated on p.10 is not obvious to me, especially in the precentral gyrus. The fMRI analysis has its relevance in the EEG source localization, but maybe it would be enough to shift it to the supplementary information. With regards to the source localization in the EEG it never became clear to me which strategy was favored in the end and if fMRI was essential here. Please clarify.

We appreciate the reviewer's suggestion, and we can confirm that we considered similar ideas while writing the article. But we wanted to make the fMRI result to stand on its own because (1) it is a different imaging modality than EEG, separating the fMRI results is naturally expected; (2) we found it obligatory to replicate the previous finding which focused on the fMRI activation using the same dataset (Roy et al., 2017).

To better describe the analyses, we have updated the flowchart of our analysis pipeline, with intuitive explanations to the major statistical analyses (sFig. 1). As to the point that there is limited correspondence between the fMRI and the EEG localisation, we adopted a new visualisation scheme which shows a clearer correspondence. We want to reassure the reviewer that the resemblance is more obvious in the baseline activation (without contrasts between conditions). In fact, the global potential across the brain was generally weaker in BR compared to RPL ([-400 -200] ms before a dominant transition) (Fig. a, and sFig. b); hence a contrast between them did not reserve that correspondence (Fig. 3e, which only showed deactivation).

SI page 16:

sFig. 1. Analysis pipeline for the simultaneous fMRI-EEG data. With the preprocessed fMRI and EEG, we conducted initial data exploration (i.e. activation studies for fMRI and evoked-response analysis for EEG) by following standard neuroimaging analysis

procedures. To ascertain the task-related regions at the network level, we adopted the Intrinsic-Connectivity-Network (ICN) atlas for evaluating the fMRI activation. Since this experiment adopted a continuous BR design where there is no explicit onset of a trial (while the offset is the manual indication of a perceptual transition), we constrained the trial to 1 second (s) by iteratively taking epochs from long to short durations (e.g., from 5 to 3 to 1 s) before a manual response, until we found significant evoked-potential differences between the BR and RPL conditions that were comparable to the previous literature. To initiate the dynamic neural state analysis, we took the source signals (1 s) before every manual report from the DMN local regions (i.e., peak coordinates) shown to be significantly involved in this task by source-level evoked response analysis. With the constrained temporal and spatial range, we set up the time-delay embedded Hidden Markov Model, which is a state-of-art analysis pipeline designed to discover the intrinsic dynamic patterns among the given brain regions. We searched for four common spatiotemporal patterns (i.e., states, with a micro-window of ~100 ms) that persist in all trials (of all conditions and participants). The states were estimated at each timepoints of a trial; hence, despite being transient, they were the most robust patterns recurring across time. To establish the states' perceptual relevance, we conducted ANOVA and cross validation analyses, using the state presence as a predictor to predict the perceptual type of the upcoming transition for each trial. Finally, to understand the states, we extracted the spectral information from the states and conducted permutation tests across states, which revealed the most distinctive neural features of each state. We further focused on certain neural features of the critical state to investigate the possible route of how intrinsic DMN dynamics have an influence on the primary visual cortex.

fMRI event-related activation

Dominant phase in BR

BOLD response associated with dominant percept in BR condition

Dominant phase in RPL

BOLD response associated with dominant percept in RPL condition

EEG source-level evoked response

Pre-dominant phase in BR

source activation during [-200 -400] ms before response to a dominant percept in BR condition

Pre-dominant phase in RPL

source activation during [-200 -400] ms before response to a dominant percept in RPL condition

sFig. 3. (a) The fMRI event-related activation (without contrasts) for the dominant percept in both BR and RPL condition. (b) The EEG source-level evoked activation (without contrasts) during [-200 -400] ms before response to a dominant percept in both BR and RPL conditions. The brain map shows the brain regions with the top 10% of the T-scores.

-
- Fig. 3c: The error bars are asymmetric. Which measures are shown ?

We apologise for our mistake. The plot shows the mean (the point estimator) and its 95% confidence interval for each group. We have generated the plot again with a correct rendering, please see below (changes in the text are highlighted):

Fig. 3. Event-Related Potentials for different perceptual conditions (a) Event-Related Potential contrast between BR and RPL for dominant percept at the sensor level. The global field power (GFP) in the RPL condition was significantly higher than that in **the BR** condition from -400 to -200 ms before the reported change in percept at 0 s. This is indicated in the plot with the black line ($p\text{-corr} < 0.05$ significant clusters, by 1-sample permutation cluster t-test). **The ERP topography for the “BR (dominant) vs. RPL (dominant)” contrast is displayed at the top of the significant time window.** (b) No significant difference was found in the GFP between the BR (mixed) and RPL (mixed) conditions. **(c) GFP difference across sources of perceptual transitions and types of percept.** The interaction effect was not significant ($p\text{-corr} = 0.12$). The plot shows the averaged GFP across the time window between -400 and -200 ms and its 95% confidence intervals for each group. (d) ERP topographies in both of the BR and RPL conditions, **dynamically changing during the critical time window ([−450 −210] ms) before a response to dominant percept.** (e) Source localisation of the evoked responses contrasted between BR and RPL conditions, and for the interaction effect of the two variables, **during the [−400 −200] ms before a subjective report.** Plotted brain regions/voxels survived the significance test with FWE-corrected non-parametric P-values < 0.05 .

- Fig. 3d: Why is the contrast shown for exactly this time window ?

In Fig 3d, we wanted to demonstrate the exact time evolution of topographies that gives rise to the GFP difference across the two perceptual sources during the dominant percept. Hence, we chose a wider, continuous time-window of [-450, -210] ms, and decided to plot voltage map frames at every 80 ms. We note that the 80ms snapshots are chosen arbitrarily for visualization purposes only, in order to fit 4 equidistant voltage topographic frames in time.

To avoid confusion, we have rearranged this figure as shown below. The contrasted ERP topography was shifted to the subplot (a), above the time frame during which there were significant differences between the BR and RPL conditions (please see the above Fig 3.a., changes in text are highlighted).

-
- p. 9: I did not find finer grained results on frequency (and time windows).

We apologise for being inaccurate in our description. We actually presented the description in the SI and the actual results in the online repository: (https://htmlpreview.github.io/?https://github.com/Aubrey-Lyu/BR-project/blob/master/Analysis-3_EEG/results/evokeResponse_result_table_permutationtest.html). Please see the rephrased sentences as below:

Manuscript Page 9, Line 205-209

The activation profiles for finer grained frequency bands and time windows were also explored for a sanity check, with no further attention paid to their differences. A full list of results for all frequency bands, time windows and contrasts was presented in the online repository (https://htmlpreview.github.io/?https://github.com/Aubrey-Lyu/BR-project/blob/master/Analysis-3_EEG/results/evokeResponse_result_table_permutationtest.html)

-
- P. 11: Which figure or table shows the source level EEG activation contrast including the stated regions, e.g. PCC? (caption in Fig. 4 says 'sensor level')

We are very grateful that the reviewer spotted this typo. We actually meant "source-level" peak voxels. In addition, we enhanced the source activation plots with a new scheme of visualisation in the Fig. 3e (see the figure above), where the reviewer can see significant voxels in the DMN regions, such as the PCC.

- P. 11: I don't understand the phrase '(with 0.1-s post-response padding)' in lower page. Please elaborate.

By this phrase we meant that we extended our 1-second (s) epoch to 1.1-s (with 0.1-s post-response timeseries). We adopted the definition of "padding" from the *field-trip* website:

(https://www.fieldtriptoolbox.org/faq/how_can_i_interpret_the_different_types_of_padding_that_i_find_when_dealing_with_artifacts/):

"Padding is an operation that extends a predetermined segment of data (usually referred to as a 'trial') either with zeros or with additional data points."

It can be used for mitigating the edge effects at the beginning and the end of an epoch. Unlike the post-response padding, there is no precise timing of when the endogenous process begins, so we did not add another 0.1-s padding before the 1-s trial. Please also see earlier answer to the reviewer's question and sFig. 7, where the benefit the post-response padding is more obvious. Strictly speaking, the interpretable range of the trial should be slightly shorter than 1 second, because of the edge effect in the begging. Despite the edge effect, the 0.1s in the beginning of the trial constitutes only a small fraction across the time, and we were agnostic as to when exactly the edge effect starts, so we did not discard the first 25 time points for the following analyses, as we believe they do not influence the results.

-
- The caption of Fig. 4 contains methods details which should probably be shifted to the supplements.

We thank the reviewer for suggesting it. We created another section about ROI selection in the methods section in the Supplementary Information. Meanwhile, we also changed the Fig 4. Caption. Please see both changes as below:

SI Page 10-11, Line 245-264:

Given our research interest in the default mode network (DMN) we constrained the subsequent modelling within the following 8 regions: bilateral parahippocampal gyri (HP), bilateral inferior parietal lobules (IPL), ACC, PCC, PCU, and the primary visual cortex (V1). To ensure experimental sensitivity, we extracted signals from the significant peak voxels (confined within the DMN) in the previous source-level evoked-response analysis. The contrasts used for determining the regional involvement in the task were BR (dominant) vs. RPL (dominant), BR (mixed) vs. RPL (mixed) and the interaction effects between the two factors; both contrast directions were considered. Hence, 1-second (+0.1s post-response padding) source signals from these coordinates were

extracted for all trials (trials being the 1-s epoch before a subjective report of a perceptual change). When multiple coordinates were identified within a same ROI (identified by automated anatomical labelling), the average of their signals was used as the representative signal of that ROI. The anatomical labels for the peak coordinates were identified using the Talairach Atlas (<http://www.talairach.org/>), upon a conversion to the Talairach space. The list of the peak voxel coordinates used for this purpose are presented in the Supplementary material Table 2. The full results of all significant voxels/clusters upon all contrasts are available from an online repository:

[https://htmlpreview.github.io/?https://github.com/Aubrey-Lyu/BR-project/blob/master/Analysis-3 EEG/results/evokedResponse_result_table_permutationtest.html](https://htmlpreview.github.io/?https://github.com/Aubrey-Lyu/BR-project/blob/master/Analysis-3%20EEG/results/evokedResponse_result_table_permutationtest.html)

For the Figure and Figure caption in the Manuscript Page 11:

Fig. 4. Schematic illustration of the procedures for HMM. The one-second EEG epochs/trials were taken right before a response of every perceptual change (“dominant” in red or green, “mixed” in black). To ensure experimental sensitivity, we select our regions of interest (ROIs) from the significant results of the **source-level** evoked response in our previous analyses. Also because of our theoretical interest, **we further constrained our selection of ROIs by choosing only the V1 and DMN regions (among all the significant voxels) to construct the HMM.** In the last subplot we presented an example of how the states change across time in a

random trial. The motivation and procedures for selecting the particular time window and ROIs were detailed in the corresponding method section in the SI.

- In the state life times possible ceiling effects should be addressed.

We appreciate the reviewer's concern. The lack of significance of the state-behavioural correspondence at the trial level might indeed be influenced a ceiling effect in some of the subjects (e.g., S01, S03, S04, S08, S14, S16), where the presence of State 4 is too high to provide any variance. However, we do not think the ceiling effect is a big concern for the statistics at the group level: thanks to the prominent individual difference in BR, there is enough variability of State 4 (K4) lifetime among subjects (please see K4-lifetime breakdown as below).

K4 lifetime categories	S01	S02	S03	S04	S05	S06	S07	S08	S09	S10	S11	S12	S13	S14	S15	S16	S17	S18	S19	S20
0.95s < State 4	132	14	44	79	0	2	10	144	0	150	31	0	0	150	15	152	0	37	0	0
0.9s < State 4 <= 0.95s	28	64	63	29	2	40	91	51	13	86	97	61	1	153	27	34	0	123	0	0
0.5s < State 4 <= 0.7s	1	23	4	1	75	60	74	2	47	20	37	116	15	0	72	7	0	39	4	3
State 4 <= 0.5s	2	5	1	1	110	111	20	0	17	26	16	199	292	1	133	0	91	8	114	97

Page 15, Line 361-368:

We did not find a trial-by-trial relationship between the GC (PCU→V1) and the perceptual duration, or an interaction between the GC (PCU→V1) and State 4 lifetime to the perceptual duration. This might be caused by a ceiling effect of the State 4 lifetime from a within-subject analysis: there are 6 participants who showed a dominant presence of State 4 (> 0.9 s) in most of the trials in the BR (dominant) condition, thus providing little variability/statistical power for examining the aforementioned interaction effect. However, a relationship between the GC (PCU→V1) and the perceptual duration was established at the group level.

- Might GC (PCU -> V1) correlate with the presence of state 4?

The Pearson correlation coefficient is 0.137 at the single trial level, and 0.141 at the group level, but the tests were not significant.

- SI: In the caption of sFig. 2 please hint at table s1. Which model was finally chosen and why?

We thank the reviewer for this good suggestion, and we apologise for the omission. Please see the updated figure caption for sFig.2., in which we explained how we chose the model and referred to the sTable 1.

SI page 17:

sFig. 2. Model comparison for different inverse modelling configurations. The variational free energy, which approximates the log model evidence, is used for model comparison. The inverse model (No. 2) with the highest free energy was chosen for further analyses (Henson et al., 2010; López et al., 2014). The model specifications are described in Supplementary table 1.

• SI p. 11: I wonder if the Viterbi path can be properly estimated if subject data as well as event segments are concatenated.

We are sorry that the confusion arose. The Viterbi path (i.e., Markovian process) is estimated solely in the continuous 1-s (250 time points) time window (before a subjective report of a perceptual change). The trials, regardless of what condition/subjects they are from, are concatenated together into another dimension from which the probability of state distribution is extrapolated.

• What does sFig. 13 mean? Does it represent the alpha-band? What is the meaning of the 95%-CI? Why are some regions depicted twice like PCU in sFig. 13b left side?

SFig. 13 represents the phase difference of the PCU and V1 source signals during the 1-s time window before a report of perceptual change. This information might be interesting to people who care about the leading relationships among the DMN regions. Although we have calculated the GC as an approximation of the causal relationship, the GC was conducted on the temporal domain which disregards frequency band

information. Since the causal inference in GC is also based on temporal precedence, the simple measure of phase difference offers a very straightforward picture of the leading relationships among the regions considered. The causal interaction is not the main focus in this study, but we think it is a good supplementary material for facilitating the interpretation of the results that we presented in the main article. For example, we can see that the PCU is leading V1 in the theta band while the V1 is leading PCU in the alpha band, which might be considered together with our result of the phase coherence between of PCU and V1 in the state 4 (alpha range) and the GC (PCU→V1) result. However, the interpretation should be very careful because the state presence is varying during a trial while the phase difference or the GC measure is based on trial average.

The 95% CI is the confidence interval of the null distribution under the hypothesis that the phase difference is zero (1-sample t-test). We have updated the figure caption to make the statistical inference clearer. Please see the highlighted changes in text as below.

The X-axis is frequency, so when some regions show up again, it means they have phase relationships at different frequencies.

SI page 25:

a. Phase difference to PCU

b. Phase difference to V1

sFig. 14. Phase differences between the PCU (or V1) and the other regions of interest on original signals. The represented relationships have exceeded a 95% confidence interval of a null distribution (one-sample t-test, distribution estimated from trial variance for each condition).

Typos:

- SI: p.5 before 'EEG data processing': remove 'been'

We thank the reviewer for pointing this out. We have changed it accordingly.

- P. 11: 'We set the HMM algorithm to extract 4 states from the data, which are the concatenated regional signals of all trials from all conditions and subjects (Fig. 4).' 'Which' is not correct.

We thank the reviewer for pointing out it. Please see the revised sentence:

Manuscript page 10, Line 239:

We set the HMM algorithm to extract 4 states among the regional signals of all trials from all conditions and subjects (Fig. 4).

- First paragraph on p. 14 should be to Figure 5b instead of 4b

We thank the reviewer for spotting this mistake. We have changed it accordingly.

- P.17 after Fig. 5 wrong title insert ?

Yes indeed, we apologise for this mistake. We have reorganised the sections and renamed the section titles to make clearer the progression of analyses.

- SI p. 10, first line after title: ‘used _in_ this study’

We thank the reviewer for spotting this. We are sorry for this occasional failure of text rendering upon a PDF conversion. We have changed it (along with several others) in the revision.

- SI p. 12: ‘resulted state’ is not a proper formulation

We are sorry for this awkward wording. It was changed to “a particular hidden state”. Please see the highlighted change as below:

SI page 15, line 371:

During this investigation, we found that the relationship is dependent on the presence of a particular hidden state.

References for Reviewer 4:

- Britz, J., Pitts, M. A., & Michel, C. M. (2010). Right parietal brain activity precedes perceptual alternation during binocular rivalry. *Human Brain Mapping, 32*(9), 1432–1442. <https://doi.org/10.1002/hbm.21117>
- Cole, M. W., Bassett, D. S., Power, J. D., Braver, T. S., & Petersen, S. E. (2014). Intrinsic and task-evoked network architectures of the human brain. *Neuron, 83*(1), 238–251. <https://doi.org/10.1016/j.neuron.2014.05.014>
- Cole, M. W., Ito, T., Bassett, D. S., & Schultz, D. H. (2016). Activity flow over resting-state networks shapes cognitive task activations. *Nature Neuroscience, 19*(12), 1718–1726. <https://doi.org/10.1038/nn.4406>
- Debecker, J., & Desmedt, J. E. (1970). Maximum capacity for sequential one-bit auditory decisions. *Journal of Experimental Psychology, 83*(3, Pt.1), 366–372. <https://doi.org/10.1037/h0028848>
- Deco, G., & Jirsa, V. K. (2012). Ongoing Cortical Activity at Rest: Criticality, Multistability, and Ghost Attractors. *Journal of Neuroscience, 32*(10), 3366–3375. <https://doi.org/10.1523/JNEUROSCI.2523-11.2012>
- He, B. J. (2018). Robust, Transient Neural Dynamics during Conscious Perception. *Trends in Cognitive Sciences, 22*(7), 563–565. <https://doi.org/10.1016/j.tics.2018.04.005>
- Henson, R. N., Flandin, G., Friston, K. J., & Mattout, J. (2010). A Parametric Empirical Bayesian framework for fMRI-constrained MEG/EEG source reconstruction. *Human Brain Mapping, 31*(10), 1512–1531. <https://doi.org/10.1002/hbm.20956>
- Koivisto, M., & Revonsuo, A. (2010). Event-related brain potential correlates of visual awareness. *Neuroscience & Biobehavioral Reviews, 34*(6), 922–934. <https://doi.org/10.1016/j.neubiorev.2009.12.002>
- Kornmeier, J., & Bach, M. (2006). Bistable perception—Along the processing chain from ambiguous visual input to a stable percept. *International Journal of Psychophysiology, 62*(2), 345–349. <https://doi.org/10.1016/j.ijpsycho.2006.04.007>

- Kornmeier, J., & Bach, M. (2012). Ambiguous Figures – What Happens in the Brain When Perception Changes But Not the Stimulus. *Frontiers in Human Neuroscience*, 6, 51. <https://doi.org/10.3389/fnhum.2012.00051>
- López, J. D., Litvak, V., Espinosa, J. J., Friston, K., & Barnes, G. R. (2014). Algorithmic procedures for Bayesian MEG/EEG source reconstruction in SPM. *NeuroImage*, 84, 476–487. <https://doi.org/10.1016/j.neuroimage.2013.09.002>
- Margulies, D. S., Ghosh, S. S., Goulas, A., Falkiewicz, M., Huntenburg, J. M., Langs, G., Bezgin, G., Eickhoff, S. B., Castellanos, F. X., Petrides, M., Jefferies, E., & Smallwood, J. (2016). Situating the default-mode network along a principal gradient of macroscale cortical organization. *Proceedings of the National Academy of Sciences*, 113(44), 12574–12579. <https://doi.org/10.1073/pnas.1608282113>
- Mashour, G. A., Roelfsema, P., Changeux, J.-P., & Dehaene, S. (2020). Conscious Processing and the Global Neuronal Workspace Hypothesis. *Neuron*, 105(5), 776–798. <https://doi.org/10.1016/j.neuron.2020.01.026>
- Roy, A. V., Jamison, K. W., He, S., Engel, S. A., & He, B. (2017). Deactivation in the posterior mid-cingulate cortex reflects perceptual transitions during binocular rivalry: Evidence from simultaneous EEG-fMRI. *NeuroImage*, 152, 1–11. <https://doi.org/10.1016/j.neuroimage.2017.02.041>
- Theodoni, P., Panagiotaropoulos, T. I., Kapoor, V., Logothetis, N. K., & Deco, G. (2011). Cortical Microcircuit Dynamics Mediating Binocular Rivalry: The Role of Adaptation in Inhibition. *Frontiers in Human Neuroscience*, 5. <https://doi.org/10.3389/fnhum.2011.00145>
- Vidaurre, D., Hunt, L. T., Quinn, A. J., Hunt, B. A. E., Brookes, M. J., Nobre, A. C., & Woolrich, M. W. (2018). Spontaneous cortical activity transiently organises into frequency specific phase-coupling networks. *Nature Communications*, 9(1), 2987. <https://doi.org/10.1038/s41467-018-05316-z>
- Wang, P., Kong, R., Kong, X., Liégeois, R., Orban, C., Deco, G., Heuvel, M. P. van den, & Yeo, B. T. T. (2019). Inversion of a large-scale circuit model reveals a cortical hierarchy in the dynamic resting human brain. *Science Advances*, 5(1), eaat7854. <https://doi.org/10.1126/sciadv.aat7854>

REVIEWER COMMENTS

Reviewer #1 (Remarks to the Author):

The authors have made substantial efforts in trying to address the points raised by the reviewer(s), namely the explicit description of methodological steps, as well as the statistical considerations, applied parameters, and corresponding results.

Medium-level comments:

1. The word 'trigger' on line 28 might be softened to something like, 'indicate'.
2. While I acknowledge that the authors have tried to address the relationship between the 'states', conditions, and dynamics within the tested regions of the network, I strongly recommend refining the first mention of this relationship (esp. where there is focus on the four states – also numbers up to 10 should be spelled out, as in the case of the 'four' states) and making the links between these factors (and any additional, relevant ones) clear (e.g., in the Dynamic neural pattern analyses section). Perhaps, a small schematic or table that can demonstrate these links may also be considered, space and word limits permitting. Clarifying this will help to improve any reader's understanding of the findings.

Minor comments:

1. The 'm' in magnetic on line 45 should not be capitalized, except in the acronym.
2. There are several, new instances where there is no comma after 'i.e.'. Please rectify this for all abbreviations (i.e., "i.e.", "e.g.", etc.) in both the main manuscript and supplementary materials.
3. The 't' in t-test should be lowercase. Please check for this in all instances where this statistical test is mentioned (e.g., line 165). The 't' and 'test' should also be adjoined by a hyphen (e.g., line 166).
4. "1-sample" should be written as 'one-sample' (line 218).
5. Please double-check that the time window indicated on lines 185-186 is written correctly and matches figure 3a (and the corresponding legend).
6. Are the times indicated in square brackets (lines 224 and 226) written as the start and end points of the window (much like coordinates), or rather as the time period to which attention is being drawn? Please indicate, in the case of the former, a distinction by adding a comma to separate these two points or by adding a hyphen if the period is meant to last from -450 ms - -210 ms for example. This is unclear and great care should be taken to communicate this information correctly.
7. Line 299: '... representative of..', instead of 'to'.
8. line 302: 'Within the alpha band, ...' – 'within' might be a better choice.
9. Line 323: Please insert a period after 'conditions'.
10. Line 327: The highlighted section should begin with 'an' if the data reflect one single participant's results.
11. Line 328: When referring to multiple figures, please acknowledge as 'sFigs. 9 and 10'.
12. Line 469: '... hinted at'.

13. Line 470: 'a fundamental component of conscious processing.'
14. SI: As I understand it, the first reference (line 31) is not formatted according to the journal's guidelines (instead it is seen as "(1)"). Please update this.
15. SI: Line 62: Electrocardiogram should not begin with a capitalized letter in this instance.
16. SI: Line 69: 'Gradient' does not need to begin with a capitalized letter.
17. SI: Line 85: Artifact or artefact? The word 'artefact' is used in the main article, but found as 'artifact' in the supplementary materials.
18. SI: Line 118: 'On' should be changed to 'in' MATLAB.
19. SI: Lines 122-126: Please indicate the exact number of regressors used (be it main or nuisance).
20. SI: Line 225: Same comment as before, to clarify carefully the time window.
21. SI: Line 233: Bistable is written without a hyphen here, whereas it is written as 'bi-stable' in the main text. Perhaps, 'bistable' can be adopted throughout both main and supplementary texts.
22. SI: Line 407: The word 'many' should be changed to 'may'.
23. SI: Line 455: There should be no 'd' in 'two-sample'.

Reviewer #2 (Remarks to the Author):

Revision of the revised manuscript „Perturbation or Function? Intrinsic brain dynamics in the Default Mode Network predict involuntary fluctuations of visual awareness“ by Lyu et al.

General:

The manuscript has considerably improved compared to its first version, even though following all the different analysis steps is still a challenge. In the following I list some few points that still need to be addressed before publication.

Line 34 ff: “... Being aware is believed to be a globally “illuminated” inner state when locally encoded information gets propagated through downstream pathways and becomes accessible to other processing streams in the brain!”

I am puzzled by the “downstream” description at this place. According to what I know the propagation of information from early sensory areas to higher level areas is typically described as “upstream”. Is this a mistake or am I wrong?

Line 52: “ ... the two images should be distinguishable but of comparable cognitive load. ... “

Cognitive load may be one factor for a balanced binocular rivalry perception. Others and typically more strongly emphasized factors are contrast, eye dominance, content of the two pictures, etc.. The authors should mention two or three of those factors and then refer to related reviews, like Blake 2001 (or his newer one).

Line 169: ICN is introduced without definition. After a long search I finally found the definition in the supplementary materials. The definition – or at least the decoding of the shortcut – should occur at the first place “ICN” is used in the main manuscript!

Supplementary materials, line 247: Please give the definitions of the shortcuts ACC, PCC, PCU.

Supplementary materials, line 298: “... we randomly selected half of the subjects, ran the HMM analysis ... “

You probably intended to write: “... we randomly selected half of the subjects DATA, ran the HMM analysis ... “

Minor Points:

Lines 100 ff. “For maximally eliciting BR perception, the experiment used stimuli of rotating green and red checkerboard images, which were presented to each eye simultaneously with the order counterbalanced across individuals.”

It is not entirely clear what has been counterbalanced here. Most probably, which color has been presented to which eye. This should be better explained.

Line 112: comparied => compared

Supplementary materials, line 323: "... Granger causality is used this study to establish..."

Replace with: "... Granger causality is used IN this study to establish..."

Reviewer #3 (Remarks to the Author):

I would like to thank the authors for clarifying their results. The manuscript has been significantly improved. The authors argue that their conclusion do not rely on the precise timing of the perceptual transitions, but clearly this is not true.

For example, in the abstract, they conclude “intrinsic EEG oscillations can predict UPCOMING perceptual transitions”, and “intrinsic DMN dynamic PRE-EMPT the content of consciousness”. All these claims emphasize a CAUSAL effect of the hidden DMN dynamic on the perceptual transitions. However, the main finding of the current study shows that the latent neural state can differentiate the dominant or mixed percepts of rivalry and replay conditions. There is no clear evidence that these endogenous dynamics are pre-perceptual.

Selecting 1-second window before the perceptual switch will include more pre-perceptual processes, but it can not remove the contribution of post-perceptual dynamics. The very significant difference in GFP 300ms before the button press from figure 3 clearly indicates that most of the effect from the current study should be post-perceptual, since the delay of button press from the perceptual switch should be around 500ms.

Reviewer #4 (Remarks to the Author):

All my concerns were properly addressed.

In the supplements figure reference in ll.309ff: ‘Cross validation’ should probably go to sFig.6 instead of sFig.15

We thank the reviewers for all their comments which we have addressed below. In our responses, we address each of the reviewers' comments in turn, with reviewer comments in black text, our replies in blue text (following each comment), and with the referred revised text from the article highlighted in green.

REVIEWER COMMENTS

Reviewer #1 (Remarks to the Author):

The authors have made substantial efforts in trying to address the points raised by the reviewer(s), namely the explicit description of methodological steps, as well as the statistical considerations, applied parameters, and corresponding results.

Medium-level comments:

1. The word 'trigger' on line 28 might be softened to something like, 'indicate'.

We thank the reviewer for this suggestion, we have replaced 'trigger' with 'elicit'.

2. While I acknowledge that the authors have tried to address the relationship between the 'states', conditions, and dynamics within the tested regions of the network, **I strongly recommend refining the first mention of this relationship (esp. where there is focus on the four states – also numbers up to 10 should be spelled out, as in the case of the 'four' states) and making the links between these factors (and any additional, relevant ones) clear (e.g., in the Dynamic neural pattern analyses section).**

Perhaps, a small schematic or table that can demonstrate these links may also be considered, space and word limits permitting. Clarifying this will help to improve any reader's understanding of the findings.

We thank the reviewer for their great points. We have now spelled out all the numbers below 10. In addition, we have added a schematic illustration of the relationship between conditions and states. Please see the revision in page 11 as below:

Fig. 4. (a) Schematic illustration of the procedures for HMM. The one-second EEG epochs/trials were taken right before a response of every perceptual change (“dominant” in red or green, “mixed” in black). To ensure experimental sensitivity, we select our regions of interest (ROIs) from the significant results of the source-level evoked response in our previous analyses. Also because of our theoretical interest, we further constrained our selection of ROIs by choosing only V1 and DMN regions (among all the significant voxels) to construct the HMM. In the last subplot (embedded-lag HMM) we presented an example of how the states change across time in a random trial. The motivation and procedures for selecting the particular time window and ROIs were detailed in the corresponding method section in the SI. **(b) Illustration of the relationship between conditions and states.** The timeseries (constructed for demonstration purposes) indicate the EEG source signals of a single trial. The TDE-HMM algorithm clusters the timeseries into four states based on the signals’ inherent spectral features. The inherent states are assumed to recur across time and to be replicable in all trials, conditions and participants, as the endogenous neural activity has been shown to have a robust spatiotemporal structure. However, given the phenomenological differences among the conditions, the composition of the states during the trial are hypothesised to vary.

Minor comments:

1. The 'm' in magnetic on line 45 should not be capitalized, except in the acronym.

We have corrected this typo.

2. There are several, new instances where there is no comma after 'i.e.'. Please rectify this for all abbreviations (i.e., "i.e.," "e.g.," etc.) in both the main manuscript and supplementary materials.

We have changed all instances of "i.e.," accordingly.

3. The 't' in t-test should be lowercase. Please check for this in all instances where this statistical test is mentioned (e.g., line 165). The 't' and 'test' should also be adjoined by a hyphen (e.g., line 166).

We have changed it accordingly. We also italicised the statistical variables such as *t*, *F*, and *p*.

4. "1-sample" should be written as 'one-sample' (line 218).

We have changed it accordingly throughout the manuscript.

5. Please double-check that the time window indicated on lines 185-186 is written correctly and matches figure 3a (and the corresponding legend).

We thank the reviewer for pointing this out. We have removed the confusing statements by adding the individual statistics in the time-window of [-400, -200] ms, where we observed a significantly eventful epoch based on the grand average GFP. As for the [-500, 0] ms time window, it is where we conducted the cluster-based permutation *t*-test. This method allows us to locate the exact time points where the difference becomes significant, while correcting for the alpha inflation from the multiple comparisons that were conducted on all the relevant time points.

To explain the cluster-based permutation test briefly, it is a non-parametric test whereby the *t*-statistic is calculated at each time-point separately. The adjacent significant timepoints are considered as a cluster, and the sum of the *t* score of the largest cluster is taken. Then we permute the label of conditions and repeat the aforementioned steps for constructing a null distribution of the cluster-level *t* scores. If a cluster is significant, its cluster-level *t* score should be greater than what would be observed at random (with 95% confidence interval). Since we already know that the critical cluster should have happen within the [-400, -200] ms window, we selected a slightly wider and narrower-than-1-s window (i.e. [-500, 0 ms]) to conduct the cluster-based permutation *t*-test, which generated a significant cluster of [-388, -350] ms.

To avoid confusion, we have updated the concerned paragraph in the manuscript as below:

Page 8, line 186-191

Global Field Power (GFP) of the perceptual conditions revealed that the most eventful epoch was between [-400, -200] ms, which was when the EEG sensor voltages differed the most between the BR (dominant) and RPL (dominant) conditions (Mean GFP was 0.99 μ V for BR and 1.59 μ V for RPL, with a significant difference: $t = -2.23$, $p = 0.037$). Further, a permutation cluster *t*-test over the [-500, 0] ms time-window confirmed significant time-clusters between [-388, -350] ms, p -corr < 0.05 (Fig. 3a).

6. Are the times indicated in square brackets (lines 224 and 226) written as the start and end points of the window (much like coordinates), or rather as the time period to which attention is being drawn?

Please indicate, in the case of the former, a distinction by adding a comma to separate these two points or by adding a hyphen if the period is meant to last from -450 ms - -210 ms for example. This is unclear and great care should be taken to communicate this information correctly.

We apologise for the confusion. We have adopted the reviewer's suggestion by adding a comma to separate the start and end time points of the time window of interest.

7. Line 299: '... representative of..', instead of 'to'.

We have changed it accordingly.

8. line 302: 'Within the alpha band, ...' – 'within' might be a better choice.

We have changed it accordingly.

9. Line 323: Please insert a period after 'conditions'.

We have changed it accordingly.

10. Line 327: The highlighted section should begin with 'an' if the data reflect one single participant's results.

We have changed it accordingly.

11. Line 328: When referring to multiple figures, please acknowledge as 'sFigs. 9 and 10'.

We have changed it accordingly.

12. Line 469: '... hinted at'.

We have changed it accordingly.

13. Line 470: 'a fundamental component of conscious processing.'

We have changed it accordingly.

14. SI: As I understand it, the first reference (line 31) is not formatted according to the journal's guidelines (instead it is seen as "(1)"). Please update this.

We thank the reviewer for pointing it out. We have changed it.

15. SI: Line 62: Electrocardiogram should not begin with a capitalized letter in this instance.

We have changed it accordingly.

16. SI: Line 69: 'Gradient' does not need to begin with a capitalized letter.

We have changed it accordingly.

17. SI: Line 85: Artifact or artefact? The word 'artefact' is used in the main article, but found as 'artifact' in the supplementary materials.

We changed the "artifact" to "artefact" in the supplementary materials.

18. SI: Line 118: 'On' should be changed to 'in' MATLAB.

We have changed it accordingly.

19. SI: Lines 122-126: Please indicate the exact number of regressors used (be it main or nuisance).

To accommodate the numbers of the regressors used, we updated the paragraph as follows:

SI, page 5, line 121

To fit the fMRI timeseries of each scanning session, the modelled event-related BOLD responses associated with the three main effects (dominant green percept, dominant red percept and mixed percept) were taken as the main regressors, along with the non-neuronal confounds (the first six principal components extracted from WM/CSF and six movement parameters) and the block effects (the 5 task blocks). To increase reliability, each participant was scanned in multiple sessions, with about half (~6) of the sessions under the BR condition and another half under the RPL. Therefore, the first-level GLM for each participant was modelled with a mixed-level factorial design, with percept-generating conditions (BR/RPL) and perceptual types (dominant/mixed) as two factors. The global session effects were specified as nuisance covariates (i.e., separate columns of identity vectors indicating different scanning sessions).

20. SI: Line 225: Same comment as before, to clarify carefully the time window.
We have changed it as before.

21. SI: Line 233: Bistable is written without a hyphen here, whereas it is written as 'bi-stable' in the main text. Perhaps, 'bistable' can be adopted throughout both main and supplementary texts.
We have changed it accordingly.

22. SI: Line 407: The word 'many' should be changed to 'may'.
We have changed it accordingly.

23. SI: Line 455: There should be no 'd' in 'two-sample'.
We have changed it accordingly.

We thank the reviewer for pointing out all the typos in our manuscript

Reviewer #2 (Remarks to the Author):

Dear Authors,

find my comments in the attached pdf.

Revision of the revised manuscript „Perturbation or Function? Intrinsic brain dynamics in the Default Mode Network predict involuntary fluctuations of visual awareness“ by Lyu et al.

General:

The manuscript has considerably improved compared to its first version, even though following all the different analysis steps is still a challenge. In the following I list some few points that still need to be addressed before publication.

Line 34 ff: “... Being aware is believed to be a globally “illuminated” inner state when locally encoded information gets propagated through downstream pathways and becomes accessible to other processing streams in the brain1”

I am puzzled by the “downstream” description at this place. According to what I know the propagation of information from early sensory areas to higher level areas is typically described as “upstream”. Is this a mistake or am I wrong?

We thank the reviewer for this comment, we changed this sentence to the following:

Being aware is believed to be a globally “illuminated” inner state when locally encoded information gets propagated through **subsequent** pathways and becomes accessible to other processing streams in the brain¹.

Line 52: “ ... the two images should be distinguishable but of comparable cognitive load. ... ”

Cognitive load may be one factor for a balanced binocular rivalry perception. Others and typically more strongly emphasized factors are contrast, eye dominance, content of the two pictures, etc.. The authors should mention two or three of those factors and then refer to related reviews, like Blake 2001 (or his newer one).

We thank the reviewer for pointing this out. We have changed it accordingly:

...the two images should be distinguishable but of **comparable features such as image contrast, cognitive load of the content etc.**¹.

Line 169: ICN is introduced without definition. After a long search I finally found the definition in the supplementary materials. The definition – or at least the decoding of the shortcut – should occur at the first place “ICN” is used in the main manuscript!

We apologise for this omission. The very first mention of the ICN was embedded in the figure 2c, that’s where we spelled the ICN out. In the updated version, we expanded the full expression in the main text as requested (page 7 & 10).

Page 7:

Hence, the colour bars of the subplots (a)-(c) indicate the **t** scores from the group-level testing. The circular plot in (c) shows the **Intrinsic Connectivity Network** (ICN) affiliation of the significant clusters for the interaction effect.

Page 10:

In addition, **four** components have often been specified for discovering the EEG microstates in existing literature⁴¹, and were shown to have correspondence with the **Intrinsic Connectivity Network** (ICN) dynamics measured from the BOLD signals⁴².

Supplementary materials, line 247: Please give the definitions of the shortcuts ACC, PCC, PCU.

We apologise for this omission and thank the reviewer for pointing it out. We have added their definitions in the revised version.

Supplementary materials, line 298: “... we randomly selected half of the subjects, ran the HMM analysis ... ”

You probably intended to write: “... we randomly selected half of the subjects DATA, ran the HMM analysis ... ”

We thank the reviewer for pointing this out. We have changed it accordingly.

Minor Points:

Lines 100 ff. “For maximally eliciting BR perception, the experiment used stimuli of rotating green and red checkerboard images, which were presented to each eye simultaneously with the order counterbalanced across individuals.”

It is not entirely clear what has been counterbalanced here. Most probably, which color has been presented to which eye. This should be better explained.

We apologise for this confusion. We now describe it with more words in the SI (Page 2, line 43):
“The order of green and red visual stimuli for two eyes in the rivalry condition, was counterbalanced between the left and right eyes across participants and their multiple experimental sessions”. We adopted this phrase in the revised manuscript.

Line 112: compared => compared

We thank the reviewer for pointing this out. We have changed it accordingly.

Supplementary materials, line 323: “... Granger causality is used this study to establish...” Replace with: “... Granger causality is used IN this study to establish...”

We thank the reviewer for pointing this out. We have changed it accordingly.

We thank the reviewer for pointing out all the typos in our manuscript

Reviewer #3 (Remarks to the Author):

I would like to thank the authors for clarifying their results. The manuscript has been significantly improved. The authors argue that their conclusion do not reply on the precise timing of the perceptual transitions, but clearly this is not true.

For example, in the abstract, they conclude “intrinsic EEG oscillations can predict UPCOMING perceptual transitions”, and “intrinsic DMN dynamic PRE-EMPT the content of consciousness”. All these claims emphasize a CAUSAL effect of the hidden DMN dynamic on the perceptual transitions. However, the main finding of the current study shows that the latent neural state can differentiate the dominate or mixed percepts of rivalry and replay conditions. There is no clear evidence that these endogenous dynamics are pre-perceptual.

Selecting 1-second window before the perceptual switch will include more pre-perceptual processes, but it can not remove the contribution of post-perceptual dynamics. The very significant difference in GFP 300ms before the button press from figure 3 clearly indicates that most of the effect from the current study should be post-perceptual, since the delay of button press from the perceptual switch should be around 500ms.

We appreciate the reviewer’s concerns about our selection of the 1-second time window for the analysis: as argued by the reviewer, the GFP result around the 300ms before a motor response, might be related to post-perceptual events. We agree with the reviewer that the GFP result alone cannot support

our claim about the perceptual endogenous activities. In fact, we have discussed in the paper that the GFP result might not be pre-perceptual but rather indicates conscious recognition of a perceptual disambiguation (manuscript page 8), and the pre-perceptual endogenous activities should be earlier than that (SI page 9). These statements were made based on our results as well as the previous BR literature (Kornmeier & Bach, 2012).

Here we will focus on the evidence we have relied on for supporting our claims. In our previous response to the reviewer, we argued that the *precise* timing of perceptual transition is contentious both empirically and theoretically, hence we opt for modelling the perceptual event as a dynamically evolving process with the newly developed method TDE-HMM. We believe our TDE-HMM results captured the ongoing endogenous activities preceding a perceptual transition for the following reasons.

Firstly, we claimed in the paper that State 4 captures the ongoing endogenous neural activities preceding a perceptual transition. We have presented the timecourse of its probabilistic presence over the pre-motor trial before a BR (dominant) transition (sFig. 7). It showed a higher presence of State 4 around [-0.8, -0.2] s, which precedes and overlaps with the [-0.4, -0.2] time window we identified from the evoked-response analysis. This suggests that the critical state became active before the visual reversal.

sFig. 7. The frequency (or proportion) cross trials of the State 4 being present at each time point in the BR (dominant) condition.

Secondly, by considering the overall State presence (i.e., averaged fractional occupancy/FO) during the 1-s time window, we found the states can predict the upcoming transition types (page 11). This analysis was conducted with a machine learning algorithm (cross validation), which offers a standardised protocol for inferences about feature predictability. It is true that we are agnostic of the moments of perceptual change; however, the neural states of that particular moment should not dominate the entire 1-s window as would the endogenous neural activities, which are supposed to have a relatively robust spatiotemporal structure (Cole et al., 2014, 2016; Margulies et al., 2016; Theodoni et al., 2011; Vidaurre et al., 2018).

Finally, we would like to politely disagree with the reviewer's claim that "the delay of button press from the perceptual switch should be around 500 ms". According to the literature, the upper limit of pure motor execution is about 150 ms; while the reversal period (related to perceptual disambiguation)

following a bistable visual stimulus is estimated to last 50-70 ms (Debecker & Desmedt, 1970; Kornmeier & Bach, 2012). Therefore, the 1-s time window before a motor response should mostly cover the pre-perceptual neural activities. We did not truncate the trial up to the -300 ms before a response because (1) the HMM-TDE is a data-driven method which will compute the states at all given timepoints of all conditions anyway; and variation in the dataset would benefit the state clustering. (2) The [-400, -200] ms time window for visual disambiguation is an estimate at the grand-average level which would surely vary in individual trials. Truncating uniformly this window from each trial is equally imprecise. In addition, it may cause failure to state detection, as we have considered that the whole process up to an explicit motor response might be a continuous entity.

Reviewer #4 (Remarks to the Author):

All my concerns were properly addressed.

In the supplements figure reference in ll.309ff: ‚Cross validation‘ should probably go to sFig.6 instead of sFig.15

Unfortunately, we cannot locate this mistake

REVIEWER COMMENTS

Reviewer #3 (Remarks to the Author):

I would like to thank the authors for their clarifications. The high presence of state 4 from 800ms before the button press is interesting, but still it is not a strong evidence of pre-perceptual activity. For the flickering BR stimuli in this study, perception is unlikely to be exclusive and the perceptual transition could take a relatively long time. Yes, you can use 1 second or even longer as the window for HMM analysis, that window contains the HMM states doesn't mean it is pre-perceptual. Overall I think the study is interesting and important that showing the hidden state from the DMN can indicate perceptual transitions, but the conclusion of a causal influence need to be careful.

REVIEWERS' COMMENTS

Reviewer #3 (Remarks to the Author):

I would like to thank the authors for their clarifications. The high presence of state 4 from 800ms before the button press is interesting, but still it is not a strong evidence of pre-perceptual activity. For the flickering BR stimuli in this study, perception is unlikely to be exclusive and the perceptual transition could take a relatively long time. Yes, you can use 1 second or even longer as the window for HMM analysis, that window contains the HMM states doesn't mean it is pre-perceptual. Overall I think the study is interesting and important that showing the hidden state from the DMN can indicate perceptual transitions, but the conclusion of a causal influence need to be careful.

We thank the reviewer for their concern. In the final version, we have read through the whole manuscript carefully and made sure that we did not make an overstated inference about DMN's causal influence. We changed one sentence (following the cross-validation result) that may be considered as such to the following version:

Original:

“This suggests that these neural states actually encoded the endogenous neural activities that triggered, and hence could be used to predict, the perceptual change during BR.”

Revised:

“This suggests that these neural states have registered the endogenous neural activities that could predict the upcoming perceptual change during BR. (page 12, Line 292-293)”